# Continual Model Routing in Evolving Model Hubs

**Jack Bell**[1]   **Giacomo Carfi**[1]   **Gerlando Gramaglia**[1]   **Vincenzo Lomonaco**[2]

## Abstract

AI model hubs provide access to a rapidly growing collection of powerful pre-trained models, enabling off-the-shelf mixture-of-experts systems with different routing strategies. However, this rapid growth poses two fundamental challenges: scaling model selection across thousands of experts and continually updating routing mechanisms as new models and tasks are introduced. In this paper, we formalise this setting as Continual Model Routing (CMR) and propose *CMRBench*, a new large-scale benchmark simulating realistic hub expansion and including over 2,000 candidate models. Finally, we introduce *CARvE*, a contrastive embedding approach for efficient continual model routing via checkpoint-based anchoring and structured replay. Extensive empirical results and ablations show that CARvE significantly outperforms zero-shot retrieval, fine-tuning, and adapter-merging baselines in model, family, and domain-level accuracy.

## 1. Introduction

AI model hubs now host *millions* of pre-trained foundation models spanning modalities, domains, and levels of specialisation. As a result, the central question has shifted from "can we build a model?" to "which model should we run?". In many real-world deployments, this choice must be made *dynamically* and *before inference*, under strict latency and cost constraints: given an input query, the system must select a single model that is likely to perform well, without executing multiple candidates to compare outputs (Patil et al., 2024; Wang et al., 2025; Lin et al., 2025). Model routing thus emerges as a first-class problem, tightly coupled to practical efficiency and reliability rather than model capacity alone.[1]

[1]Department of Computer Science, University of Pisa, Pisa, Italy [2]LUISS University, Rome, Italy. Correspondence to: Jack Bell <jack.bell@phd.unipi.it>.

*Proceedings of the $43^{rd}$ International Conference on Machine Learning*, Seoul, South Korea. PMLR 306, 2026. Copyright 2026 by the author(s).

[1]Our code is available here: ⚙ Github

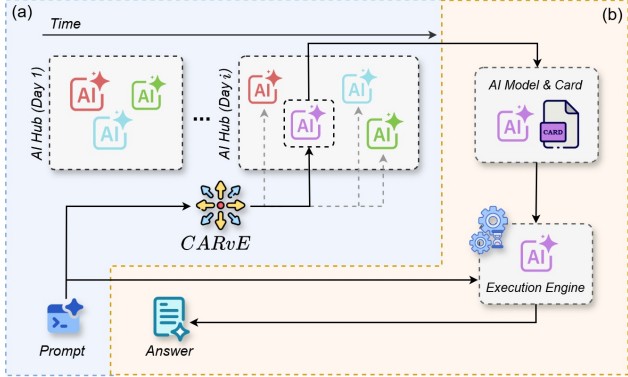

*Figure 1.* Conceptual framework for Continual Model Routing. (a) Given an evolving collection of AI models, our adaptive router *CARvE* learns continually from selection samples how to dynamically route a prompt query to the most appropriate model. (b) Actual execution of the model (not the focus of this paper).

This shift reflects a broader trend toward scaling by specialisation rather than by monolithic models. End-to-end trained Mixture-of-Experts (MoE) architectures (Shazeer et al., 2017) have demonstrated that activating a small subset of specialised components can outperform a single dense model at comparable cost, while improving coverage and robustness (Su et al., 2025). Beyond internal MoE layers, modern systems increasingly compose *heterogeneous*, independently pretrained experts—often dozens of models—into tool-use or orchestration pipelines (Ong et al., 2024; Chen et al., 2024b). Model hubs effectively externalise this paradigm, acting as continuously expanding repositories of experts with diverse inductive biases, training data, and capabilities. Consequently, system performance is increasingly determined by the quality of expert selection rather than by individual model improvements.

However, this emerging scenario introduces fundamental challenges that are not addressed by existing approaches. First, routing must scale to *thousands* of candidate models, far beyond the regime where exhaustive post-inference evaluation or naive ranking remains feasible. Second, model hubs are inherently non-stationary: new experts are added, existing ones are updated, and others become obsolete. While routing can be framed as a pure retrieval problem—ranking model descriptions or embeddings by similarity to a query—static retrieval methods have shown to be

brittle in large expert spaces (Zhang et al., 2023). Moreover, Retrieval-Augmented Training (RAT) (Patil et al., 2024) and Generation (RAG) (Lewis et al., 2020), though effective for out-of-distribution generalisation, do not ensure stable routing performance as the expert pool evolves and the effective target space shifts. In this setting, the goal is no longer to train a router once, but to *efficiently maintain* high-quality routing decisions as the model hub itself changes over time.

In this work, we argue that *model routing must be treated as a continual learning problem.* As model hubs expand and evolve, a practical router must simultaneously (i) scale to thousands of candidate experts and (ii) adapt efficiently to non-stationary expert pools without retraining from scratch (Figure 1). Hence, we formalise pre-inference model selection as a *continual classification* task, where the label space grows over time as new models are introduced. This perspective enables routers that can be updated incrementally based on users' selection preferences and operate under realistic data and compute constraints. Building on this formulation, we introduce CMRBench, the first benchmark designed specifically for this setting. Prior routing benchmarks assume a fixed candidate pool; CMRBench instead treats hub evolution as a first-class property, organising evaluation into four temporally grounded experiences that together span more than 2,000 candidate models. We regard it as both the evaluation substrate for this work and a reproducible reference point for the community going forward. On this basis, we also propose CARvE, a scalable embedding-based router built for continual adaptation as the model hub grows.

Our contributions can be summarised as follows:

- We introduce *Continual Model Routing (CMR)* and formalise it as a continual learning problem where a router predicts the most suitable model for a given prompt and incrementally adapts over evolving model hubs (Section 2).

- We introduce *CMRBench*, that simulates realistic hub expansion across four sequential experiences, encompassing over 2,000 candidate models and multiple domains (Section 3.1).

- We propose *model family accuracy*, an intermediate-granularity evaluation metric that captures semantic routing quality beyond exact model-identifier matching, while remaining more discriminative than coarse domain-level accuracy (Section 3.2).

- We present *CARvE*, a contrastive embedding approach for CMR, enabling efficient scaling to thousands of models and fast adaptation while minimizing storage of private user selection preferences (Section 2).

- We provide extensive empirical results showing that *CARvE* substantially outperforms strong pre-inference

routing baselines—including retrieval-based methods, model merging, and naive replay—under realistic hub expansion (Section 3.3).

To the best of our knowledge, this is the first work to empirically demonstrate—through large-scale benchmarks, ablations, and analysis—the central role of continual learning in model routing, and to show how treating routing as a continual classification problem enables scalable expert selection and robust adaptation in evolving model hubs. All code and benchmarks will be open-sourced upon publication.

## 2. CARvE: Continual Anchored Router with Contrastive Embeddings

We study *pre-inference* model routing in an evolving hub. As in classic class-incremental settings, data arrive as experiences $\{E_t\}_{t=1}^T$ (van de Ven et al., 2022), where each $E_t$ provides supervised samples $(q_i, m_i, d_i)$ comprising a query $q_i$, a target model identifier (ID) $m_i \in \mathcal{M}_t$, and an optional domain label $d_i$. The candidate pool grows cumulatively as $\mathcal{M}_{\leq t} = \bigcup_{k \leq t} \mathcal{M}_k$. At experience $t$, the objective is to learn a router $g_t$ that predicts an appropriate model-ID for any query seen so far,

$$g_t : q \mapsto m, \qquad \forall q \in \bigcup_{k \leq t} E_k,$$

whilst mitigating forgetting under an expanding label space. Routing is *selection-only*: the router chooses a model-ID without executing candidate models.

**Model cards (used by baselines).** Many pre-inference routing approaches use *model cards*—natural-language descriptions of each model's intended use, capabilities, and limitations—as a retrieval corpus or as additional context for a router (e.g., retrieval-only and retrieval-augmented training). We denote the card for model $m$ as $c_m$ and the set available in an experience as $\mathcal{C} = \{c_m\}_{m \in \mathcal{M}}$. In CMR-Bench, model cards are baseline-specific side information used to retrieve and rank candidate or condition an LLM controller. In contrast, **CARvE does not consume model-card text**: it routes by embedding the user query and comparing it against a learned, continually updated registry of model embeddings, trained from supervised prompt–model pairs.

As shown in Figure 2, CARvE routes by embedding similarity and is trained continually using candidate sets, replay, periodic hard-negative mining, and checkpoint-based anchoring.

**Design rationale.** While replay buffers, embedding-based retrieval, and parameter regularisation have each been explored in prior work, continual model routing imposes a combination of constraints that existing approaches are not designed to jointly satisfy. In CARvE, routing is performed

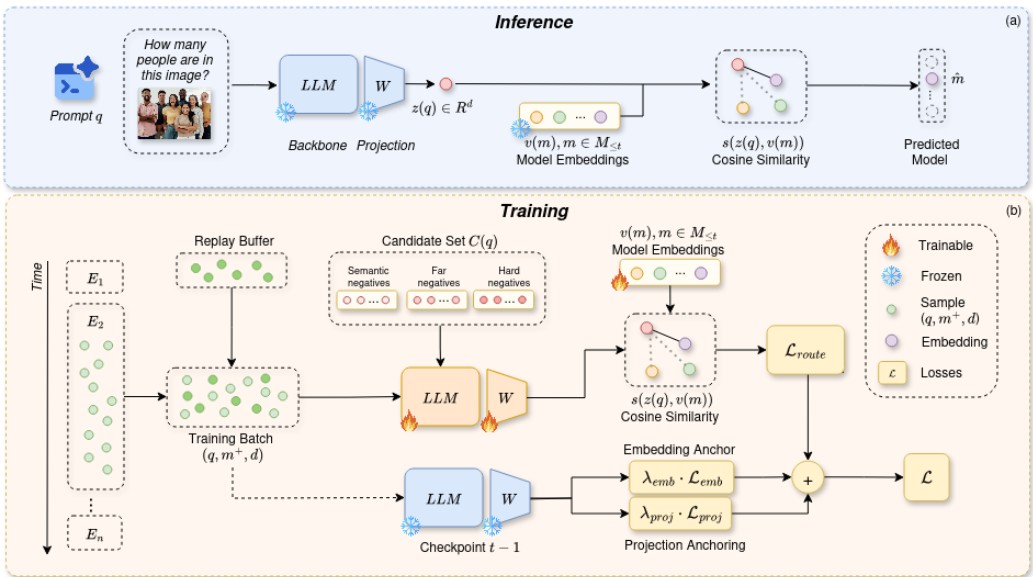

*Figure 2.* Continual model routing in evolving hubs. (a): pre-inference routing selects a model by embedding similarity without executing candidate models. (b): continual training with candidate sets combines routing loss with checkpoint-based embedding and projection anchoring, while periodic hard-negative mining maintains discriminative candidates as the model hub expands.

by comparing query embeddings against a continuously expanding registry of model embeddings, rather than through a fixed classifier head. As a result, embeddings for previously observed model-IDs must remain geometrically stable across experiences, while the router must still incorporate newly introduced models efficiently. At the same time, routing must scale to thousands of candidates without executing any candidate model, ruling out post-inference comparison strategies and making full output-space optimisation increasingly costly.

The design choices of CARvE follow from these requirements. Asymmetric anchoring preserves the geometry of previously learned model embeddings and the query projection, while leaving newly added model embeddings free to adapt. Fixed-size candidate-set training avoids full softmax over the expanding model registry and keeps per-example scoring proportional to $\mathcal{O}(Kd)$. Structured negative sampling, including hard, semantic, and far negatives, maintains discriminative separation as the label space grows.

**Model Embedding Initialisation.** Model embeddings are initialised randomly rather than warm-started from model-card representations. Across four card-based variants covering global and within-domain initialisation at two hyperparameter settings each, every variant underperforms random initialisation by 3–5pp D-Acc with roughly double the forgetting (Table 24). Card embeddings encode linguistic similarity between descriptions rather than routing discriminability, so the contrastive objective must first undo that geometry before it can organise the space usefully.

**Embedding-based router.** A backbone language model produces final-layer hidden states $H \in \mathbb{R}^{L \times D}$ for the tokenised input. We pool over prompt tokens to obtain a query representation $h(q) \in \mathbb{R}^D$ (we use the final prompt-token state). A learned projection $W \in \mathbb{R}^{D \times d}$ yields a query embedding, which we $\ell_2$-normalise:

$$u(q) = h(q)W, \qquad z(q) = \frac{u(q)}{\|u(q)\|_2} \in \mathbb{R}^d.$$

Each model-ID $m$ has a learned embedding $v(m) \in \mathbb{R}^d$; we similarly normalise $e(m) = v(m)/\|v(m)\|_2$. Given a candidate set $\mathcal{C}(q) \subseteq \mathcal{M}_{\leq t}$, we score candidates using temperature-scaled cosine similarity,

$$s(q, m) = \frac{z(q)^\top e(m)}{\tau}, \qquad m \in \mathcal{C}(q),$$

and predict $\hat{m} = \arg\max_{m \in \mathcal{C}(q)} s(q, m)$.

In CARvE, the backbone LM is frozen throughout; LoRA adapters are trained at every experience. The projection head $W$ and model embedding table $\{v(m)\}$ are also updated at each experience, along with the registry and mining caches.

**Candidate-set training.** Training with a full softmax over $|\mathcal{M}_{\leq t}|$ is costly when the hub contains thousands of models. We therefore train with fixed-size candidate sets $\mathcal{C}(q)$ of size $K$ that always include the positive model (the ground-truth label $m_i$, denoted $m^+$ within $\mathcal{C}(q)$) and a mixture of negatives: cached *hard* confusers mined under the current router, *semantic* negatives from the same (or related) domain when available, and *far* negatives from other domains, with

random fill to avoid duplicates. Hard negatives are refreshed periodically by scoring within-domain pools and retaining the top-scoring incorrect models.

**Domain–model coreset replay.** Random replay can be highly redundant under heavy-tailed hubs (many models with few examples). We therefore construct a replay buffer as a *domain–model coreset*: given a replay budget $B$, we allocate per-domain quotas approximately proportional to domain frequency, with a minimum per domain and an optional cap. Within each domain we additionally cap the number of stored examples per model-ID, and select diverse examples via farthest-point sampling (FPS) in a fixed embedding space using cosine distance. Concretely, FPS greedily selects examples that maximise their minimum distance to the already selected set, promoting coverage over near-duplicates. Full details (quotas, caps and embeddings) are deferred to Appendix C.3.

**Objectives and anchoring.** Given $(q, m^+)$ and its candidate set $\mathcal{C}(q)$, we optimise cross-entropy over candidate logits:

$$\mathcal{L}_{\text{route}}(q, m^+) = -\log \frac{\exp(s(q, m^+))}{\sum_{m \in \mathcal{C}(q)} \exp(s(q, m))}.$$

To stabilise routing behaviour across experiences, at the start of experience $t$ we take a reference snapshot of the router parameters from the end of experience $t{-}1$, and penalise drift relative to this snapshot whilst learning on $E_t$. Let $\mathcal{M}_{\leq t-1}$ denote the set of model-IDs observed up to experience $t{-}1$. Let $v_{t-1}(m)$ and $\Theta_{t-1}$ denote the model-embedding rows and prompt-projection parameters at the end of experience $t{-}1$. During training on $E_t$, anchoring is applied only to previously seen IDs $m \in \mathcal{M}_{\leq t-1}$; embeddings for newly introduced IDs are left unanchored.

We penalise embedding drift using cosine distance:

$$\mathcal{L}_{\text{emb}} = \frac{1}{|\mathcal{M}_{\leq t-1}|} \sum_{m \in \mathcal{M}_{\leq t-1}} \left( 1 - \frac{v_t(m)^\top v_{t-1}(m)}{\|v_t(m)\|_2 \|v_{t-1}(m)\|_2} \right)$$

(with an $\ell_2$ variant in the Appendix). Let $\Theta_t$ denote the router projection parameters (in our implementation, a single matrix $W$). We anchor the projection by mean-squared error:

$$\mathcal{L}_{\text{proj}} = \frac{1}{|\Theta_t|} \sum_{\theta \in \Theta_t} \frac{1}{|\theta|} \|\theta - \theta_{t-1}\|_2^2.$$

The final objective is:

$$\mathcal{L} = \mathcal{L}_{\text{route}} + \lambda_{\text{emb}} \mathcal{L}_{\text{emb}} + \lambda_{\text{proj}} \mathcal{L}_{\text{proj}}.$$

We maintain a persistent registry of model-ID with stable indices; when new models appear, we append new rows to

---

**Algorithm 1** Continual model routing (training)

1: **Input:** experiences $\{E_t\}_{t=1}^T$, candidate size $K$, weights $\lambda_{\text{emb}}, \lambda_{\text{proj}}$
2: **State:** registry $\mathcal{R}$, projection $W$, embeddings $\{v(m)\}$, hard-negative cache $\mathcal{H}$
3: **for** $t \leftarrow 1$ **to** $T$ **do**
4:     **if** $t > 1$ **then**
5:         Save snapshot from end of $t{-}1$ for anchoring
6:     **end if**
7:     Add new IDs in $E_t$ to $\mathcal{R}$; expand embedding table
8:     **for all** batches $\mathcal{B} \subseteq E_t$ **do**
9:         Build candidates for $\mathcal{B}$     $\triangleright$ $m^+$ + hard/semantic/far
10:         Compute $\mathcal{L}_{\text{route}}$ on candidates
11:         Add $\lambda_{\text{emb}} \mathcal{L}_{\text{emb}} + \lambda_{\text{proj}} \mathcal{L}_{\text{proj}}$
12:         Gradient step
13:     **end for**
14:     Refresh $\mathcal{H}$ by hard-negative mining (periodic)
15: **end for**

---

the embedding table (copying existing rows and initialising new ones) so that training proceeds without reindexing prior labels. Anchoring is applied only to router parameters ($W$ and embedding rows for previously seen model-IDs), and does not regularise backbone LM parameters or LoRA weights used by generative baselines.

**Complexity.** Per example, scoring costs O(Kd) once z(q) is computed, and does not require executing candidate models. Hard-negative mining adds occasional scoring over a larger within-domain pool and is amortised over training. At larger hub scales, the model registry can be indexed with approximate nearest-neighbour search (e.g. FAISS), reducing retrieval cost to $O(log|M|)$ without retraining.

**Training Details.** We report all CARvE hyperparameters (candidate-set composition, hard-negative mining, anchoring coefficients, and optimisation settings) in Appendix C.1, Table 10.

## 3. Empirical Evaluation

### 3.1. Datasets

We introduce CMRBench, a continual model routing benchmark composed of four sequential experiences. It integrates two existing benchmarks—APIBench (Patil et al., 2024) and ToolMMBench (Wang et al., 2025)—together with **HuggingBench**, which we introduce for model routing over *Hugging Face* models and structured into two experiences. All datasets are mapped to a schema for continual learning. We treat APIBench as reference format and adapt the others to match it. Each instance contains an *instruction*, a *model_identifier* in Hugging Face form (*owner/repo_id*), and the associated *api_data* (model card JSON format in the APIBench specification).

**APIBench.** APIBench contains 1,645 machine learning

*Table 1.* Summary statistics of the CMRBench in temporal order. Experiences 1–2 correspond to APIBench and ToolMMBench, while Experiences 3–4 correspond to HuggingBench. Dates denote the upper bound on model release time for each experience.

|  | $E_1$ | $E_2$ | $E_3$ | $E_4$ |
|---|---|---|---|---|
| Samples | 8693 | 5391 | 9831 | 10330 |
| Models | 852 | 481 | 520 | 547 |
| Domains | 40 | 35 | 49 | 45 |
| Prev. Models | 0 | 120 | 70 | 83 |
| Date | $\leq 2023$ | $\leq 2024$ | $\leq 2025.05$ | $\leq 2026$ |

API calls scraped from TorchHub, TensorFlow Hub, and Hugging Face Hub (HF) across 61 domains. To avoid systematic inconsistencies across hubs, we restrict to `HF` entries, canonicalise model-IDs to `owner/repo_id` and drop non-conforming datapoints. We keep the original train/eval split.

**ToolMMBench.** ToolMMBench is a Hugging Face multimodal benchmark with 932 models across multiple domains. Since queries can map to multiple valid models, we enforce a one-to-one instruction–model mapping. We verify availability and extract domain from model metadata, discarding inaccessible or untagged models, resulting in 481 models.

**HuggingBench.** We introduce *HuggingBench*, a temporally structured benchmark for *Continual Model Routing* on the evolving Hugging Face hub. The benchmark is defined as a sequence of two routing experiences. Each experience consists of a persistent subset of *legacy models* and a set of *newly introduced models*, capturing both long-term availability and continuous expansion of the candidate pool. Queries in $Q_t$ are generated via a self-instruct procedure (Wang et al., 2023), conditioned on model cards and inspired by prior benchmarks (Patil et al., 2024; Wang et al., 2025). Table 1 reports statistics for all four experiences, including model overlap with past experiences; full construction, preprocessing, and human evaluation details are in Appendix A.

### 3.2. Metrics

To comprehensively evaluate the performance of the model router, we report multiple metrics capturing different aspects of prediction quality.

**Model-ID accuracy (M)**: fraction of predictions that exactly match the ground-truth model identifier.

**Domain accuracy (D)**: fraction of predictions belonging to the same domain as the ground-truth model.

**Model family accuracy (F)**: intermediate-granularity metric between $M$ (overly fine at $> 2,000$ models) and $D$ (overly coarse). We group functionally similar variants (e.g., yolov8**m/n/s**) into families via name normalisation and clustering; details are in Appendix B.1.

**Label Snapping**: Invalid SFT-generated model-IDs are re-placed with the closest valid name using Levenshtein similarity with a **0.8** threshold. Metrics $M$, $D$, and $F$ are computed after snapping. This is not used in CARvE.

**Forgetting**: We measure *forgetting* (FGT) via backward transfer (BWT) (Rodríguez et al., 2018), reporting FGT as |BWT| for $M$, $D$, and $F$ to assess knowledge retention across sequential experiences.

### 3.3. Experiments and Results

We report preliminary experiments to narrow the configuration space, followed by main experiments comparing continual routing strategies and *CARvE*. All experiments follow the sequential experience framework of (Lomonaco et al., 2021) with experiences $E_1, \ldots, E_T$. At each experience, we perform different experiments including supervised fine-tuning (SFT) with LoRA adapters (Hu et al., 2022) on a LLaMA2-7B backbone (Touvron et al., 2023). All experiments are run on one NVIDIA H100 GPU.

**Retrieval-Only.** Following Gorilla (Patil et al., 2024), we evaluate retrieval-based routers using a cumulative corpus of unique model cards. We compare **BM25** (Robertson & Zaragoza, 2009) and open-source retrievers: **SentenceTransformers** (Reimers & Gurevych, 2019) with `all-mpnet-base-v2`, **SPLADE** (Formal et al., 2021), and **BGE-M3** (Chen et al., 2024a) to cover lexical, dense, and modern sparse/reranking approaches. Extended results (Appendix, Table 13) show **BGE-M3** achieves the highest average model-ID accuracy (13.8%), and is therefore used as the default retriever in subsequent experiments.

**Zero-Shot Fine-Tuning VS RAT.** We compare Gorilla *zero-shot* fine-tuning (Patil et al., 2024) (predict model-ID from query only) with *RAT*, which augments the query with the top-1 retrieved model card (BGE-M3 over the cumulative corpus). Both use replay, where at experience $t$ we include 10% of data from each prior experience. Results (Appendix, Tables 14 and 15) show zero-shot consistently outperforms RAT, suggesting retrieval often adds misleading context; we therefore adopt zero-shot in subsequent experiments (templates in Appendix C.2).

**Model Merging.** We train one LoRA adapter per experience and merge them using averaging, *TIES* (Yadav et al., 2023) and *DARE* (Yu et al., 2024) (using equal weights). TIES performs best (mean M-acc 8.9 across three experiences and three runs), so we report only TIES in the main tables. Full settings and before and after label snapping results are in the Appendix (Tables 16 and 17).

**Main Experiments.** Our main evaluation compares *CARvE* to baselines spanning common continual learning strategies: *sequential fine-tuning*, *random replay* with 5%, 10%, and 20% past data, *joint training* (zero-shot and RAT), *cumulative* training where a single LoRA adapter is incrementally

updated on $\bigcup_{k \leq t} E_k$ and checkpointed after each experience; and *from scratch* training, where a fresh LoRA adapter is trained at each experience on the same cumulative data. We also test a *retrieval-only update* baseline that freezes the adapter after $E_1$ and updates only the retriever corpus (i.e., a Gorilla RAG-style system), and a *HuggingGPT*-like controller that routes via metadata and prompting rather than a learned router. For each experiment, we performed three runs with different random seeds. This is to account for variability due to stochastic optimisation and data ordering, and to report statistically more reliable results.

Table 2 reports mean accuracy and FGT over four experiences for model-ID (M), model-family (F), and domain (D). *CARvE* consistently outperforms continual baselines and reduces FGT, particularly at the domain level. Sequential fine-tuning exhibits severe catastrophic forgetting - while current experience accuracy is competitive, performance on earlier experience collapses (Table 18), leading to the largest forgetting across all three metrics. Replay mitigates this degradation and improves with larger budgets (5-10%), confirming rehearsal is necessary at hub scale.

At matched replay budgets, CARvE improves average domain accuracy by large margins: at 10% replay, CARvE achieves 80.7% D-acc versus 75.9% for standard replay, while cutting domain FGT from 13.1% to 5.9%. It even surpasses LoRA joint training (79.3% D-acc). With 20% replay, CARvE reaches 82.9% D-acc with 3.0% domain FGT, compared to 78.1% and 7.8% for standard LoRA replay. Model-ID accuracy also improves with CARvE as replay increases (e.g. 46.4% M-acc at 20% replay), indicating that anchoring and structured replay help preserve fine-grained decision boundaries rather than only coarse domain cues.

**Per-experience behaviour under hub growth.** Per-experience results show standard replay degrades sharply when HuggingBench is introduced (Exp. 3–4), notably on Exp. 3 after training on Exp. 4: at 15% replay, D-Acc drops to 54%(Table 19), whereas CARvE maintains 69.7% (Table 20). A similar gap appears on Exp. 1 after Exp. 4 (60.8% vs. 74.5%). This supports anchoring: stabilising prompt and model embeddings prevents drift toward newly added model-IDs.

Figure 3a compares domain accuracy over four experiences across learning strategies. The left line plot tracks accuracy after training up to Exp. $X$, while the right histogram reports the average accuracy on Exp. $X$ over all subsequent training stages. LoRA and CARvE joint training are upper-bound baselines, with CARvE performing best. Sequential fine-tuning, TIES, and Gorilla-RAG degrade sharply, while replay mitigates forgetting: Replay 10% helps, and CARvE Replay 10% nearly matches LoRA joint training. HuggingGPT-style is intermediate.

**Retrieval and Joint-training baselines.** Retrieval-only routing is comparatively stable but has significantly lower accuracy at hub scale, indicating that model card similarity alone is not sufficient to resolve fine-grained routing decisions among thousands of candidates. Joint training on all experiences simultaneously provides an upper bound for how well a router can perform when the full data distribution is available: it yields strong average accuracy, where notably CARvE with joint training outperforming LoRA. CARvE closes the gap to joint training and, in some cases, surpasses it on domain accuracy, while still operating under the continual constraint where data arrive sequentially.

**Regularization-based methods.** Elastic Weight Consolidation (EWC) (Kirkpatrick et al., 2017) improves over Learning without Forgetting (LwF) (Li & Hoiem, 2018) in average routing accuracy, but both methods remain substantially below CARvE and exhibit significantly higher forgetting. In particular, EWC reaches 66.2% D-Acc with 31.4% D-Fgt, compared to 80.7% D-Acc and 5.9% D-Fgt for CARvE at the same replay budget. Moreover, CARvE-EWC performs comparably to standard CARvE, suggesting that Fisher-style regularisation alone is insufficient to explain CARvE's gains.

**Backbone sensitivity.** CARvE and Random Replay are also evaluated with Qwen2.5-7B and Qwen3-4B (Table 2). Results show that routing performance scales with backbone capacity and remains stable across comparable architectures. CARvE with Qwen2.5-7B reaches 81.5% D-Acc, close to LLaMA2-7B (80.7%), indicating that continual routing generalises beyond one decoder family. Smaller backbones such as Qwen3-4B show higher forgetting and lower fine-grained routing accuracy, suggesting that representation quality remains a bottleneck under continual hub expansion.

### 3.4. Ablation

We ablate CARvE over four sequential experiences to isolate the contribution of replay design and anchoring. All ablations use the same backbone model, random seeds, training budgets, and evaluation protocol as the default configuration, and differ only in the factor under study. Table 3 reports M-acc, D-acc and D-forgetting averaged over experiences.

**Replay percentage and replay strategy.** At low replay budgets, domain-stratified coreset replay provides stronger performance than random replay, improving both M-Acc and D-Acc at 5% replay (40.5% vs. 37.3% M-Acc; 78.5% vs. 77.4% D-Acc). At larger replay budgets (10–20%), the gap between replay strategies becomes substantially smaller, likely because CMRBench experiences exhibit significant model overlap across time (as shown in Figure 1).

**Compute efficiency.** Figure 3b quantifies this trade-off concretely. CARvE at 10% domain replay requires roughly

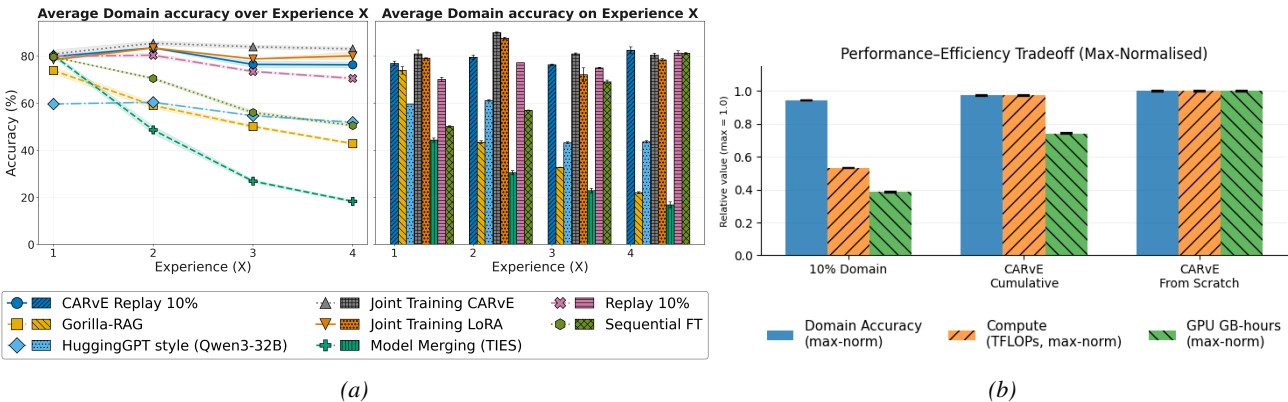

*(a)*  *(b)*

*Figure 3.* **(a)**: The line plot shows performance when the model is trained in a continual learning setting up to Experience X, while the histogram reports the average accuracy on Experience X across all subsequent training stages. **(b)**: Max-normalised domain accuracy, compute (TFLOPs), and GPU GB-hours for CARvE 10% Domain replay vs. CARvE cumulative and from-scratch baselines. Error bars show standard deviation over three runs.

45% fewer TFLOPs and 48% fewer GPU GB-hours than cumulative training, and 47% fewer TFLOPs and 61% fewer GPU GB-hours than from-scratch retraining. At a production deployment 50 times larger than our benchmark scale, the annual saving against from-scratch retraining exceeds $170,000, at which point continual operation is a necessity rather than a marginal efficiency choice. The accuracy gap relative to from-scratch training (80.7% vs. 85.7% D-Acc) should be read in light of label-space size: with over 2,000 candidate models, top-1 accuracy is a strict criterion. CARvE at 10% domain replay achieves top-3 domain accuracy of 94.8% (Table 22), so a re-ranking over three candidates yields near-perfect domain routing in practice.

**Cumulative and from scratch training.** With cumulative data, LoRA reaches 81% D-Acc with near-zero forgetting, while CARvE remains higher at 83% with similarly low forgetting, reinforcing that CARvE's advantage is not just rehearsal but improved stability under registry expansion.

**Anchoring mechanisms.** Anchoring is necessary for stability (Table 3). Disabling embedding anchoring reduces accuracy and increases forgetting ($43.1\% \rightarrow 41.7\%$ M-Acc and $5.9\% \rightarrow 7.4\%$ D-Fgt), while disabling projection anchoring causes a larger drop ($43.1\% \rightarrow 40.1\%$ M-Acc and $5.9\% \rightarrow 9.4\%$ D-Fgt). The stronger effect of projection anchoring suggests that constraining drift in the projection is critical for maintaining the embedding space over time.

**Candidate set size and label noise.** The choice of candidate set size K has little effect on routing quality: as shown in Table 3, D-Acc varies by less than one point across the range K = 32 to K = 512, which supports the use of K = 64 as a computationally convenient default. With respect to label noise, performance decreases as expected when 10% of training labels are corrupted, with D-Acc falling from 80.7% to 72.1%, though forgetting remains contained (5.9%

to 8.0%). At 20% corruption D-Acc falls further to 63.1%, consistent with the view that the contrastive objective and checkpoint-based anchoring together limit the damage noisy supervision causes to the learned embedding geometry.

## 4. Related Works

Model routing has been studied across tool invocation, expert selection, and model orchestration. Most existing work assumes either post-inference selection, a fixed and bounded candidate set, or jointly trained experts. In contrast, our work focuses on pre-inference routing in evolving model hubs, where the candidate set grows over time and routing decisions must remain reliable under continual updates. Below, we review the lines of work most closely related to this setting and clarify their limitations with respect to scalability, temporal adaptation, and deployment assumptions.

**Pre-inference model routing** Early approaches exploit the zero-shot reasoning of general-purpose LLMs, such as HuggingGPT (Shen et al., 2023), to interpret task descriptions and coordinate expert models via metadata. However, routing is treated as an emergent behaviour, leaving scalability and robustness unaddressed as the candidate set grows. Subsequent approaches introduce explicit supervision or architectural structure: Gorilla (Patil et al., 2024) finetunes LLMs to emit valid API calls given known schemas, MLLM-Tool (Wang et al., 2025) extends this setting to multimodal inputs, and Olympus (Lin et al., 2025) maps visual inputs to a predefined task taxonomy. Routoo (Mohammadshahi et al., 2024) incorporates performance and cost to select among a fixed pool of LLMs, while training-free approaches such as Eagle (Zhao et al., 2024b) rely on pairwise comparisons and ELO-style ratings. Overall, these methods assume a centralised router operating over a static or moderate candidate set, and do not address continual expansion or long-term

*Table 2.* Averaged results over all four experiences. We report mean accuracy (%) and mean forgetting (%) across experiences for Model-ID (M), Model-family (F), and Domain (D). Values after $\pm$ denote *standard error* (computed over three separate seeds across experiences). Forgetting is averaged over $t \in \{2, \ldots, T\}$. Unless specified, all methods use LLaMA2-7B as the backbone model.

| Setting | Accuracy (%) ($\uparrow$) | | | Forgetting (%) ($\downarrow$) | | |
| --- | --- | --- | --- | --- | --- | --- |
| | **M-Acc** | **F-Acc** | **D-Acc** | **M-Fgt** | **F-Fgt** | **D-Fgt** |
| **Retrieval / LLM-controller baselines** | | | | | | |
| BGE-M3 (Chen et al., 2024a) | **13.6 $\pm$ 0.0** | **16.2 $\pm$ 0.0** | 44.0 $\pm$ 0.0 | 2.5 $\pm$ 0.0 | 3.9 $\pm$ 0.0 | 3.3 $\pm$ 0.0 |
| Gorilla RAG (Patil et al., 2024) | 6.7 $\pm$ 0.2 | 10.4 $\pm$ 0.2 | 43.0 $\pm$ 0.5 | **0.0 $\pm$ 0.2** | **0.8 $\pm$ 0.2** | **0.1 $\pm$ 0.2** |
| HuggingGPT Qwen3-32B (Shen et al., 2023) | – | – | **51.7 $\pm$ 0.2** | – | – | – |
| **Continual learning experiments** | | | | | | |
| Random Replay (5%) | 36.5 $\pm$ 0.4 | 44.2 $\pm$ 0.4 | 70.2 $\pm$ 4.2 | 19.5 $\pm$ 0.5 | 23.2 $\pm$ 0.5 | 24.0 $\pm$ 4.4 |
| Random Replay (10%) | 39.1 $\pm$ 0.5 | 47.3 $\pm$ 0.7 | 75.9 $\pm$ 0.2 | 14.0 $\pm$ 0.1 | 16.8 $\pm$ 0.3 | 13.1 $\pm$ 0.2 |
| Random Replay (20%) | 41.3 $\pm$ 0.2 | 49.8 $\pm$ 0.2 | 78.1 $\pm$ 0.1 | 8.6 $\pm$ 0.4 | 11.1 $\pm$ 0.3 | 7.8 $\pm$ 0.1 |
| Random Replay Qwen2.5-7B (10%) | 40.7 $\pm$ 0.5 | 49.2 $\pm$ 0.4 | 78.2 $\pm$ 0.2 | 14.2 $\pm$ 0.6 | 16.5 $\pm$ 0.6 | 10.3 $\pm$ 0.5 |
| Random Replay Qwen3-4B (10%) | 39.6 $\pm$ 0.2 | 47.9 $\pm$ 0.1 | 77.2 $\pm$ 0.2 | 14.9 $\pm$ 0.2 | 18.2 $\pm$ 0.4 | 11.2 $\pm$ 0.4 |
| Sequential Finetuning | 28.0 $\pm$ 0.1 | 34.8 $\pm$ 0.1 | 64.3 $\pm$ 0.2 | 34.6 $\pm$ 0.2 | 41.8 $\pm$ 0.2 | 37.2 $\pm$ 0.2 |
| Model Merging (TIES) (Yadav et al., 2023) | 7.6 $\pm$ 0.3 | 10.9 $\pm$ 0.2 | 28.6 $\pm$ 0.5 | 15.0 $\pm$ 0.2 | 19.7 $\pm$ 0.1 | 32.6 $\pm$ 0.2 |
| LwF (Li & Hoiem, 2018) | 28.8 $\pm$ 0.3 | 35.9 $\pm$ 0.5 | 56.4 $\pm$ 0.2 | 20.8 $\pm$ 0.3 | 25.7 $\pm$ 0.2 | 39.5 $\pm$ 0.2 |
| EWC (Kirkpatrick et al., 2017) | 31.3 $\pm$ 0.4 | 38.4 $\pm$ 0.4 | 66.2 $\pm$ 0.1 | 26.6 $\pm$ 0.3 | 32.7 $\pm$ 0.2 | 31.4 $\pm$ 0.5 |
| CARvE EWC (10% replay) | 42.2 $\pm$ 2.4 | 47.7 $\pm$ 2.3 | 80.5 $\pm$ 2.2 | 9.1 $\pm$ 1.9 | 9.8 $\pm$ 3.3 | 6.1 $\pm$ 4.8 |
| CARvE (5% replay) | 40.5 $\pm$ 0.7 | 45.4 $\pm$ 0.6 | 78.5 $\pm$ 0.4 | 15.9 $\pm$ 1.7 | 17.4 $\pm$ 1.1 | 10.1 $\pm$ 0.7 |
| CARvE (10% replay) | 43.1 $\pm$ 0.2 | 48.5 $\pm$ 0.3 | 80.7 $\pm$ 0.0 | 11.4 $\pm$ 0.4 | 12.5 $\pm$ 0.3 | 5.9 $\pm$ 0.3 |
| CARvE Qwen3-4B (10% replay) | 36.7 $\pm$ 3.4 | 42.1 $\pm$ 3.9 | 78.6 $\pm$ 3.3 | 13.5 $\pm$ 3.3 | 14.3 $\pm$ 4.5 | 9.1 $\pm$ 5.0 |
| CARvE Qwen2.5-7B (10% replay) | 42.4 $\pm$ 0.1 | 48.1 $\pm$ 0.3 | 81.5 $\pm$ 0.3 | 11.2 $\pm$ 0.2 | 12.3 $\pm$ 0.1 | 6.2 $\pm$ 0.6 |
| CARvE (20% replay) | **46.4 $\pm$ 0.1** | **51.9 $\pm$ 0.2** | **82.9 $\pm$ 0.2** | **6.4 $\pm$ 0.8** | **7.3 $\pm$ 0.8** | **3.0 $\pm$ 0.3** |
| **Upper bounds (non-continual)** | | | | | | |
| Joint Training Zero-Shot | 43.1 $\pm$ 0.6 | 51.6 $\pm$ 0.4 | 79.3 $\pm$ 0.9 | – | – | – |
| Cumulative Zero-Shot | 45.9 $\pm$ 0.4 | 54.6 $\pm$ 0.1 | 81.9 $\pm$ 0.5 | 0.6 $\pm$ 0.3 | 1.2 $\pm$ 0.2 | 0.8 $\pm$ 0.2 |
| From Scratch Zero-Shot | 43.9 $\pm$ 0.5 | 53.0 $\pm$ 0.6 | 80.1 $\pm$ 0.3 | **0.2 $\pm$ 0.5** | 1.1 $\pm$ 0.5 | 1.9 $\pm$ 0.1 |
| Joint Training RAT | 39.9 $\pm$ 0.2 | 48.4 $\pm$ 0.3 | 77.3 $\pm$ 0.3 | – | – | – |
| Joint Training CARvE | 49.5 $\pm$ 0.2 | 55.4 $\pm$ 0.2 | 84.6 $\pm$ 0.5 | – | – | – |
| Cumulative CARvE | 48.1 $\pm$ 1.2 | 54.2 $\pm$ 0.9 | 83.5 $\pm$ 0.3 | 0.8 $\pm$ 0.1 | 1.0 $\pm$ 0.1 | **0.5 $\pm$ 0.1** |
| From Scratch CARvE | **50.4 $\pm$ 1.2** | **56.4 $\pm$ 1.4** | **85.7 $\pm$ 0.4** | 0.8 $\pm$ 0.2 | **0.8 $\pm$ 1.0** | 0.5 $\pm$ 0.3 |

retention across thousands of heterogeneous models.

**Post-inference model routing.** A complementary line of work routes *after* executing multiple candidates and then ranking, fusing or orchestrating their outputs. Methods such as LLM-Blender (Jiang et al., 2023), LLM-TOPLA (Tekin et al., 2024), and Smoothie (Guha et al., 2024) improve output quality via learned ranking, diversity-aware ensembling, or per-sample quality estimation. In parameter-efficient settings, AdapterSoup (Chronopoulou et al., 2023), LoraHub (Huang et al., 2024), and LoraRetriever (Zhao et al., 2024a) retrieve and combine adapters or LoRA modules conditioned on the input. Other work optimises orchestration under additional constraints (e.g. ToolOrchestra (Su et al., 2025)) or supports limited continual tool usage (e.g., COLT (Liu et al., 2025)). While effective for quality improvement, these approaches incur substantial inference cost and are orthogonal to our pre-inference setting.

**Mixture-of-Experts and expert routing.** Routing is central to Mixture-of-Experts (MoE) models (Jacobs et al., 1991; Shazeer et al., 2017; Fedus et al., 2022): a gate selects a sparse subset of experts per input, enabling capacity to scale with the number of experts while keeping per-token compute bounded. This scaling-by-experts paradigm increasingly appears at the *system level* as well, where applications route each query to one model among a pool of specialised LLMs to trade off cost and quality (Chen et al., 2024b; Ong et al., 2024; Hu et al., 2024). However, MoE assumes joint training of experts and gate, and most system-level routing work assumes a largely fixed candidate pool; model hubs instead evolve continuously as new experts are added and others change. Recent work begins to consider routing under dynamic model pools (Shu et al., 2026), and our work targets hub-scale pre-inference routing under continual expansion.

**API-routing benchmarks.** Several benchmarks target tool-augmented LLMs rather than hub-scale model routing. API-Bank (Li et al., 2023) and API-BLEND (Basu et al., 2024) provide annotated datasets for API detection and invocation,

*Table 3.* CARvE ablations over four experiences. Cells show **M-Acc**, **D-Acc**, **D-Fgt** (in %). Values are mean $\pm$ SD over three separate seeds.

| Setting | EmbA | ProjA | M-Acc | D-Acc | D-Fgt |
|---|---|---|---|---|---|
| *Replay type & ratio (all components enabled)* | | | | | |
| Domain (5%) | ✓ | ✓ | 40.5±0.7 | 78.5±0.4 | 10.1±0.7 |
| Domain (10%) | ✓ | ✓ | 43.1±0.2 | 80.7±0.0 | 5.9±0.3 |
| Domain (20%) | ✓ | ✓ | 46.4±0.1 | 82.9±0.2 | 3.0±0.3 |
| Random (5%) | ✓ | ✓ | 37.3±0.5 | 77.4±1.0 | 10.1±0.7 |
| Random (10%) | ✓ | ✓ | 43.0±0.1 | 80.6±0.2 | 6.1±0.7 |
| Random (20%) | ✓ | ✓ | 45.8±0.4 | 82.8±0.3 | 3.2±0.4 |
| *Component ablations (dom replay, 10%)* | | | | | |
| No emb. anchor | × | ✓ | 41.7±0.3 | 77.9±0.6 | 7.4±0.5 |
| No proj. anchor | ✓ | × | 40.1±1.3 | 78.6±0.6 | 9.4±1.0 |
| *Candidate set size K (dom replay, 10%, both anchors)* | | | | | |
| $K = 32$ | ✓ | ✓ | 40.5±0.5 | 80.1±0.2 | 6.0±0.6 |
| $K = 48$ | ✓ | ✓ | 40.7±0.1 | 80.0±0.3 | 6.7±0.5 |
| $K = 96$ | ✓ | ✓ | 42.3±0.2 | 80.5±0.3 | 5.9±0.7 |
| $K = 192$ | ✓ | ✓ | 43.2±0.3 | 80.6±0.1 | 6.6±0.2 |
| $K = 384$ | ✓ | ✓ | 43.2±0.7 | 80.8±0.4 | 6.3±0.6 |
| $K = 512$ | ✓ | ✓ | 43.4±0.3 | 81.0±0.2 | 6.2±0.5 |
| *Label Noise (dom replay, 10%)* | | | | | |
| 10% noise | ✓ | ✓ | 38.7±0.6 | 72.1±0.7 | 8.0±0.7 |
| 20% noise | ✓ | ✓ | 36.1±0.7 | 63.1±1.7 | 8.4±1.2 |
| *Cumulative Training* | | | | | |
| Cumulative | ✓ | ✓ | 48.1±1.2 | 83.5±0.3 | 0.5±0.1 |
| From Scratch | × | × | 50.4±1.2 | 85.7±0.4 | 0.5±0.3 |

while TaskBench (Shen et al., 2024) and ToolBench (Xu et al., 2023) evaluate structured tool selection and parameters. These benchmarks focus on APIs/tools and do not capture continual emergence of new models or routing over thousands of hub-hosted candidates.

**Continual learning for routing.** Our setting also relates to continual learning (Parisi et al., 2019), which studies how models acquire knowledge over time while mitigating catastrophic forgetting (Kirkpatrick et al., 2017). Recent work argues that continual learning remains critical even with foundation models (Bell et al., 2025), supporting continual pre-training, fine-tuning, and compositional systems. This suggests progress will be driven less by a single static model and more by ecosystems of evolving, interacting models—making CL increasingly relevant for routing.

## 5. Discussion and Conclusion

We introduced *Continual Model Routing (CMR)* as a class-incremental formulation of pre-inference model selection in large, evolving model hubs, where a router must continually incorporate newly released models while preserving competence on previously seen ones. To support rigorous evaluation, we constructed a four-experience benchmark spanning APIBench, ToolMMBench, and HuggingBench, and we pro-

posed *CARvE*, an anchored vector-embedding router that scales to hub-sized label spaces via fixed-size candidate-set training and a persistent model registry. We also introduced *model family accuracy* as an intermediate-granularity metric that better reflects practical routing quality than exact model-ID matching alone.

CARvE delivers consistent gains in accuracy and retention relative to strong continual baselines. In the main setting, CARvE substantially improves D-Acc while reducing forgetting: at 10% replay, CARvE reaches 80.7% D-Acc with 5.9% D-Fgt, compared to 75.9% and 13.1% for standard replay; at 20% replay, CARvE reaches 82.9% D-Acc with 3.0% forgetting, compared to 78.1% and 7.8% for replay. These gains extend to fine-grained routing, improving model-ID accuracy (46.4% vs. 41.3% at 20% replay) and model-family accuracy (51.9% vs. 59.8%). Ablations confirm that CARvE's stability mechanisms are necessary under registry expansion: domain-stratified coreset replay reduces long-tail forgetting at higher replay budgets, and anchoring is critical—disabling projection anchoring causes the largest degradation in both M-Acc and D-Fgt.

**Limitations and future work.** A key limitation of CARvE is that it does not support *zero-shot* routing for newly added models: when new model-IDs appear, CARvE requires an additional continual learning step and examples of how those models are used in order to learn reliable prompt–model associations. A further limitation concerns representational capacity: while LoRA adapters are updated each experience, the backbone LM is frozen throughout, so its hidden representations may become insufficient if later experiences introduce queries or domains that are substantially out-of-distribution relative to its initial training distribution. For significantly longer experience sequences, periodic backbone updates would be necessary.More broadly, an important extension is to reduce dependence on labelled prompt–model pairs by incorporating self-supervised objectives that exploit prompt, model, and domain similarity. In preliminary attempts, we explored using model-card similarity as a soft target for contrastive learning, but this did not yield reliable gains, highlighting that model-card quality and consistency must improve for such signals to be effective.

Finally, while this work focuses on *pre-inference* routing, future systems can benefit from a hybrid *pre-* and *post-inference* strategy: first predict the domain (or a small domain-consistent subset of models), then execute a small number of top candidates within that domain and perform lightweight ranking to improve final routing accuracy while controlling cost. Taken together, we hope this work encourages the community to treat routing in evolving model hubs as a first-class continual learning problem, and to develop selectors that remain accurate, stable, and scalable as the underlying model ecosystem continues to grow.

## Acknowledgements

This research was partially supported by the FIS2 Grant from the Italian Ministry of University and Research (MUR), Grant No. FIS2023-03382, under the project "Continual, Decentralized Compositionality for Sustainable Artificial Intelligence" with Prof. Vincenzo Lomonaco serving as Principal Investigator (PI).

## Impact Statement

Our work targets a practical systems bottleneck: selecting an appropriate specialist model before inference as model hubs grow and change over time. By improving hub-scale routing accuracy and reducing catastrophic forgetting under continual updates, CMRBench and CARvE can lower latency, cost, and energy relative to post-inference orchestration that runs multiple models per query, and can make specialised capabilities more accessible in resource-constrained deployments. However, better routing also increases the effectiveness of any downstream model, including potentially unsafe or biased experts; misrouting can select models with weaker safety properties or amplify domain-specific biases. Deployment should therefore pair routing with safety policies (e.g., model allowlists, capability audits, and monitoring), and treat user preference data with care; CARvE's embedding-based design reduces reliance on storing raw user–model histories, but does not eliminate privacy risk.

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

# A. Dataset and Preprocessing

## A.1. APIBench

Data are originally distributed across multiple files, corresponding to Hugging Face, TensorFlow Hub, and TorchHub models. Due to systematic inconsistencies across hubs—such as differing model-naming conventions, different model-IDs distributions between training and evaluation sets—we did not merge these files into a single unified dataset. Instead, we restrict our analysis to Hugging Face models only, discarding TensorFlow Hub and TorchHub entries to maintain a coherent canonical schema and avoid extensive hub-specific normalization.

APIBench data include an `api_data` field containing the model card information, which in turn includes an `api_name` field representing the model identifier. However, in most cases this field does not conform to the expected `owner/repo_id` format, which leads to *not found* errors when retrieving model information from the hub. To address this issue, we implemented rules to extract the correct model-ID from the `api_call` field. Nevertheless, due to data heterogeneity, some model-IDs could not be recovered, and the corresponding samples were discarded. Instructions were extracted from the `code` field by identifying the relevant system prompt and actual user input. Finally, we removed the `api_arguments`, `example_code`, and `python_environment_requirements`, fields from `api_data`, as they are not particularly relevant to our analysis, may introduce noise into the model card information, and would be difficult to reconstruct consistently across different datasets.

We now provide a concrete example illustrating a single APIBench entry before and after preprocessing.

---

**APIBench entry before preprocessing**

```
{
    "code":
        "###Instruction: Design a feature for a social media website...
        ###Output:
            <<<domain>>>: Natural Language Processing Sentence Similarity\n
            <<<api_call>>>: AutoModel.from_pretrained('princeton-nl...
            <<<api_provider>>>: Hugging Face Transformers\n
            <<<explanation>>>:1. We first import the necessary classes and...
            <<>>: from transformers import AutoTokenizer..."
    "api_call": "AutoModel.from_pretrained('princeton-nlp/unsup-simcse-roberta-base')"
    "provider": "Hugging Face Transformers"
    "api_data": {
        "domain": "Natural Language Processing Sentence Similarity",
        "framework": "Hugging Face Transformers",
        "functionality": "Feature Extraction",
        "api_name": "princeton-nlp/unsup-simcse-roberta-base",
        "api_call": "AutoModel.from_pretrained('princeton-nl...",
        "api_arguments": null,
        "python_environment_requirements": ["transformers"],
        "example_code": null,
        "performance": {"dataset": null, "accuracy": null},
        "description": "An unsupervised sentence embedding model..."
    }
}
```

---

**APIBench entry after preprocessing**

```
{
    "model_name":"princeton-nlp/unsup-simcse-roberta-base",
    "model_family":"unsup-simcse-roberta",
    "created_at":1646263745000,
    "instruction":"###Instruction: Design a feature for...,
    "domain":"Natural Language Processing Sentence Similarity",
    "original_dataset":"APIBench",
    "model_source":"HuggingFace",
    "api_data":{
        "domain":"Natural Language Processing Sentence Similarity",
```

```
        "framework":"Hugging Face Transformers",
        "functionality":"Feature Extraction",
        "api_name":"princeton-nlp/unsup-simcse-roberta-base"
        "api_call":AutoModel.from_pretrained('princeton-nlp/unsup-s...',
        "performance":{"dataset":null,"accuracy":null},
        "description":"An unsupervised sentence embedding...,
        "explanation":"1. We first import...
    }
}
```

## A.2. ToolMMBench

To process ToolMMBench and adapt it to our schema, the `model_name` field was extracted from the `conversations` field, which contains the Hugging Face model-ID in the `owner/repo_id` format. Therefore, no additional preprocessing was required. Instructions were also derived from the `conversations` field by isolating the user messages. ToolMM-Bench spans multiple modalities, including text-only (e.g., Text-to-Text) as well as multimodal settings such as text+image, text+audio, and text+video. In this work, we consider *all* modalities provided by ToolMMBench and treat them uniformly at the schema level, relying on the natural-language instruction as the primary query representation. To ensure consistency with the APIBench format, we prepended the string `###Instruction:` to each extracted instruction. A key characteristic of ToolMMBench is that a single query can be resolved by multiple models. To enforce a one-to-one mapping between each instruction and a model name, we counted the occurrence frequency of all models and assigned to each instruction the least frequent model among those capable of solving it. This strategy increases the heterogeneity of the resulting dataset. Furthermore, we filtered out all examples where the model frequency was lower than 7, ensuring a sufficient number of samples per model. Since ToolMMBench does not provide model card information, we reused the model cards from APIBench whenever the model appeared in both datasets. For models not present in APIBench, we reconstructed their model cards by querying the Hugging Face Hub and extracting the same fields used in APIBench, such as `domain`, `performance`, and `description`. Models that were no longer available on Hugging Face were discarded. To compute model domains in a format consistent with APIBench, we downloaded Hugging Face model card and tags for models that are not present in APIBench and matched Hugging Face model tags to a predefined set of domain categories using normalized edit distance and assigned each model to the most similar category. These categories are provided below. Finally, unlike APIBench, ToolMMBench does not provide an official train/eval split; therefore, we constructed one by allocating 15% of the data to the evaluation set, stratified by `model_name` to preserve the distribution of models across splits. We now provide a representative example of raw ToolMMBench data prior to preprocessing; we omit the corresponding processed example, as the final representation follows the same unified format used for APIBench.

---

**ToolMMBench entry before preprocessing**

```
{
        "id": 1,
        "image_name": "",
        "video_name": "",
        "audio_name": "",
        "input_modality": "text",
        "output_modality": "text",
        "conversations": [
            {
                "from": "human",
                "value": "I came across...",
                "input_modality": "text"
            },
            {
                "from": "gpt",
                "value": "BM-K/KoSimCSE-roberta",
                "ambiguity": "-1",
                "output_modality": "text"
            }
        ]
```

```
    },
```

## Model Categories

```
categories = {
    "Multimodal": [
        "Audio-Text-to-Text","Image-Text-to-Text","Image-to-Text",
        "Visual Question Answering","Document Question Answering",
        "Video-Text-to-Text","Visual Document Retrieval","Any-to-Any",
        "Feature Extraction", "Text-to-Video",
        "Text-to-Image", "Zero-Shot Image Classification",
        "Graph Machine Learning"
    ],
    "Computer Vision":[
        "Depth Estimation", "Image Classification", "Object Detection",
        "Image Segmentation", "Text-to-Image", "Image-to-Text", "Image-to-Image",
        "Image-to-Video", "Unconditional Image Generation","Video Classification",
        "Text-to-Video","Zero-Shot Image Classification", "Mask Generation",
        "Zero-Shot Object Detection", "Text-to-3D", "Image-to-3D",
        "Image Feature Extraction", "Keypoint Detection", "Video-to-Video"
    ],
    "Natural Language Processing": [
        "Text Classification","Token Classification",
        "Table Question Answering","Question Answering",
        "Zero-Shot Classification","Translation","Summarization",
        "Feature Extraction","Text Generation","Fill-Mask",
        "Sentence Similarity","Text Ranking",
        "Text2Text Generation", "Conversational"
    ],
    "Audio": [
        "Text-to-Speech", "Text-to-Audio", "Automatic Speech Recognition",
        "Audio-to-Audio", "Audio Classification", "Voice Activity Detection",
    ],
    "Tabular": [
        "Tabular Classification", "Tabular Regression","Time Series Forecasting",
    ],
    "Reinforcement Learning": [
        "Reinforcement Learning","Robotics",
    ],
    "Other": ["Graph Machine Learning"]
}
```

### A.3. HuggingBench

HuggingBench is a large-scale, temporally structured benchmark designed to study Continual Model Routing under realistic and evolving model ecosystems. It extends prior static benchmarks for API selection (Patil et al., 2024; Wang et al., 2025) by explicitly modeling both the continuous arrival of new models and the long-term persistence of widely adopted ones, reflecting the dynamics of modern model hubs such as Hugging Face. The benchmark supports incremental routing scenarios in which a router must adapt to newly introduced candidates while retaining effective routing strategies for previously observed models.

Model collection follows a dual-track strategy. Newly released models are systematically harvested from Hugging Face across all supported domains. Models are first filtered to retain only those providing a complete model card; among these, the top-25 most downloaded models are selected within each domain ensuring coverage of models demonstrably used in practice. This is consistent with APIBench and ToolMMBench, neither of which applies post-hoc capability evaluation.

The model cards are automatically normalized and cleaned using an LLM, preserving information relevant to routing decisions, including task specification, domain, input-output behavior, and usage constraints, while removing boilerplate or redundant content. Models whose processed documentation does not meet a minimum semantic completeness criterion are discarded. In parallel, legacy models are selected based on sustained popularity, measured via download counts, and the

presence of comprehensive model cards. This design choice ensures that routing decisions are evaluated not only on novelty but also on long-term utility, a key requirement in continual routing settings.

The final hub contains 1,067 models with validated documentation, comprising 100 legacy models in APIBench or ToolMMBench, across 31 domains, and 914 newer models spanning 50 domains. This composition intentionally induces heterogeneity in capability overlap, domain granularity, and temporal availability, all of which are critical factors for studying routing under non-stationary model pools. The overall composition and the allocation of models and queries per temporal experience are summarized in Table 4.

*Table 4.* HuggingBench statistics for Continual Model Routing. Number of models, domains, queries, and overlap per temporal experience.

| Metric | Experience 3 | Experience 4 |
|---|---|---|
| #Models | 520 | 547 |
| #Legacy models | 70 | 83 |
| #New models | 450 | 464 |
| #Domains | 49 | 45 |
| #Queries | 9,831 | 10,330 |

Routing queries are derived via a self-instruct procedure adapted from prior works (Patil et al., 2024; Wang et al., 2025). For each model, an LLM conditions on the model card and domain description to generate 20 naturalistic user queries that implicitly require the model's capabilities without explicitly naming the model itself. The objective is to produce diverse user intents that are ambiguous at the surface level but discriminative at the capability level, inducing non-trivial routing decisions. The prompt enforces structured output, lexical constraints prohibiting explicit references to models or tools, and diversity across intent, phrasing, and context. A lightweight manual verification step removes near-duplicate queries and ensures strict adherence to prohibited lexical items. The high-level specification of the self-instruct prompt is provided in Figure 4. To enable continual evaluation, HuggingBench is organized into temporally ordered experiences aligned with the continual model routing setting. Queries associated with newly released models are partitioned according to model release timestamps: using the median release date as a split point, queries corresponding to earlier releases are assigned to Experience 3, while those associated with later releases form Experience 4. At the end, Experience 3 contains legacy models of APIBench and ToolMMBench, while Experience 4 contains legacy models also from Experience 3. The overall domain-level query distribution for both experiences is summarised in Table 6. All model card cleaning and query generation is performed using Qwen3-32B, selected for its open-source availability and performance comparable to gpt4o-mini (Yang et al., 2025).

To further assess query quality, two independent annotators each evaluated 50 non-overlapping prompt–model pairs sampled from HuggingBench, for a total of 100 prompts. Annotators rated: (i) domain fit, (ii) model fit, and (iii) prompt naturalness on a 1–5 Likert scale. As shown in Table 5, results indicate strong domain alignment, reasonable model specificity, and adequate prompt naturalness.

| **Criterion** | **Mean score** ↑ | **% ≥ 4** ↑ |
|---|---|---|
| Domain fit | 4.12 | 71% |
| Model fit | 3.76 | 64% |
| Prompt naturalness | 3.61 | 57% |

*Table 5.* Human evaluation of HuggingBench queries.

---

**Self-Instruct Prompt for Query Generation**

You are an expert in natural language query generation and API utilization. Your task is to generate 20 diverse and creative user queries that can interact with a given API function, based on its description, while strictly following these rules:

1. Output a single JSON object with key "queries" containing 20 query strings. 2. Queries must be natural, human-like, and clearly indicate use of the described capability. 3. Do NOT include the API's name or prohibited words such as `"API"`, `"model"`, `"tool"`, `"tools"`, `"use"`, `"function"`, or `"method"`. 4. Ensure diversity in context, style, and phrasing. 5. Each query must be self-contained and understandable without external context. 6. Avoid repetition, generic queries, or programming references unless explicitly required.

**Input format:**
```
{ "API Name":  "[API Name]",
"Description":  "[Detailed API description]",
"ProhibitedWords":  ["API", "model", "tool", "tools", "use", "function",
"method", "model_id"]
}
```
**Output format:**
```
{ "queries":  [
"Query 1:  ...",
"Query 2:  ...",
"...",
"Query 20:  ..."
]
}
```
**Input example:**
```
{ "API Name":  "model_id",
"Description":  "modelcard",
"Prohibited Words":  ["API", "model", "tool", "tools", "use", "function",
"method", "model_id"]
}
```

---

*Figure 4.* Self-Instruct Prompt for Query Generation that involves the usage of a given model of Hugging Face

## A.4. Dataset Statistics

In this section, we report summary statistics of the processed datasets used in our experiments. These statistics provide an overview of the scale, diversity, and temporal structure of the data across learning experiences. Table 6 reports the number of queries per domain across the four experiences. Domains correspond to the Hugging Face domains, which are currently defined as Tasks on the Hugging Face models webpage, and reflect the functional categorization of models used in practice. Table 7 summarizes the number of queries per model and the number of unique models per domain across experiences, and additionally reports the number of distinct model families computed over all four experiences.

*Table 6.* Number of queries per domain across experiences, where domains correspond to the Hugging Face model task categories. NLP stands for Natural Language Processing, CV for Computer Vision, QA for Question Answering, RL for Reinforcement Learning

| Domain | APIBench | ToolMMBench | HuggingBench-1 | HuggingBench-2 |
|---|---|---|---|---|
| Audio Audio Classification | 230 | 78 | 370 | 150 |
| Audio Audio-to-Audio | 239 | 70 | 120 | 300 |
| Audio Automatic Speech Recognition | 299 | 32 | 360 | 320 |
| Audio Classification | 60 | 0 | 20 | 0 |
| Audio Text-to-Audio | 0 | 16 | 0 | 0 |
| Audio Text-to-Speech | 288 | 47 | 230 | 330 |
| Audio Voice Activity Detection | 119 | 14 | 20 | 0 |
| CV Depth Estimation | 280 | 32 | 209 | 249 |
| CV Image Classification | 319 | 123 | 259 | 120 |
| CV Image Feature Extraction | 0 | 0 | 120 | 377 |
| CV Image Segmentation | 290 | 64 | 456 | 199 |
| CV Image-to-3D | 0 | 0 | 210 | 330 |
| CV Image-to-Image | 260 | 276 | 139 | 540 |
| CV Image-to-Text | 0 | 8 | 122 | 279 |
| CV Image-to-Video | 0 | 0 | 240 | 260 |
| CV Keypoint Detection | 0 | 0 | 230 | 249 |
| CV Mask Generation | 0 | 8 | 150 | 170 |
| CV Object Detection | 290 | 71 | 370 | 210 |
| CV Text-to-3D | 0 | 16 | 110 | 230 |
| CV Text-to-Image | 0 | 311 | 179 | 320 |
| CV Text-to-Video | 0 | 24 | 210 | 289 |
| CV Unconditional Image Generation | 290 | 0 | 80 | 100 |
| CV Video Classification | 297 | 37 | 209 | 240 |
| CV Video-to-Video | 0 | 0 | 130 | 350 |
| CV Zero-Shot Image Classification | 219 | 71 | 338 | 159 |
| CV Zero-Shot Object Detection | 0 | 0 | 300 | 180 |
| Multimodal Audio-Text-to-Text | 0 | 0 | 60 | 170 |
| Multimodal Doc. Question Answer | 270 | 20 | 0 | 0 |
| Multimodal Doc. Question Answering | 10 | 0 | 259 | 60 |
| Multimodal Feature Extraction | 89 | 50 | 20 | 0 |
| Multimodal Graph Machine Learning | 29 | 0 | 0 | 0 |
| Multimodal Image-Text-to-Text | 0 | 0 | 221 | 593 |
| Multimodal Image-to-Text | 287 | 126 | 52 | 40 |
| Multimodal Text-to-Image | 289 | 131 | 20 | 0 |
| Multimodal Text-to-Video | 90 | 24 | 0 | 0 |
| Multimodal Video-Text-to-Text | 0 | 0 | 259 | 179 |
| Multimodal Visual Document Retrieval | 0 | 0 | 289 | 210 |
| Multimodal Visual Question Answering | 168 | 67 | 239 | 80 |
| Multimodal Zero-Shot Image Classification | 20 | 0 | 20 | 0 |
| NLP Conversational | 180 | 0 | 0 | 0 |

Table 6 – continued from previous page

| | APIBench | ToolMMBench | HuggingBench-1 | HuggingBench-2 |
|---|---|---|---|---|
| NLP Feature Extraction | 59 | 221 | 80 | 140 |
| NLP Fill-Mask | 289 | 516 | 0 | 40 |
| NLP Question Answering | 290 | 144 | 100 | 120 |
| NLP Sentence Similarity | 329 | 168 | 529 | 250 |
| NLP Summarization | 228 | 156 | 120 | 120 |
| NLP Table Question Answering | 300 | 0 | 509 | 70 |
| NLP Text Classification | 289 | 523 | 440 | 167 |
| NLP Text Generation | 370 | 502 | 434 | 362 |
| NLP Text Ranking | 0 | 8 | 180 | 380 |
| NLP Text2Text Generation | 398 | 383 | 0 | 0 |
| NLP Token Classification | 299 | 0 | 80 | 518 |
| NLP Translation | 247 | 990 | 219 | 260 |
| NLP Zero-Shot Classification | 308 | 64 | 220 | 100 |
| RL | 188 | 0 | 0 | 0 |
| RL Reinforcement Learning | 0 | 0 | 80 | 20 |
| RL Robotics | 20 | 0 | 0 | 0 |
| Tabular Tabular Classification | 89 | 0 | 60 | 200 |
| Tabular Tabular Regression | 78 | 0 | 160 | 300 |

*Table 7.* Summary statistics of dataset composition across experiences. The table reports the number of queries per model, and models per domain observed in each experience, together with statistics on model families. *Prev. Families* denotes the number of model families in the current experience that were already present in previous experiences, capturing family-level overlap across time.

| **Statistic** | | **APIBench** | **ToolMMBench** | **HuggingBench-1** | **HuggingBench-2** |
|---|---|---|---|---|---|
| | min | 9 | 7 | 10 | 10 |
| Queries per model | max | 29 | 52 | 20 | 20 |
| | mean | 10.2 | 11.21 | 18.91 | 18.88 |
| | min | 1 | 1 | 1 | 1 |
| Unique models per domain | max | 40 | 101 | 28 | 31 |
| | mean | 21.4 | 13.7 | 10.65 | 12.16 |
| | Families | 501 | 392 | 206 | 217 |
| Model families | Prev. Families | 0 | 107 | 31 | 46 |

To further investigate the diversity and redundancy of models included in the benchmark, we analysed download count distributions across benchmarks. Table 8 reports the median, maximum, and minimum number of downloads per experience, while Figure 5 shows the histogram of download distributions across experiences. Median downloads per predicted model range from 6,945 (ToolMMBench) to 14,054 (HuggingBench-1), confirming that the benchmark covers well-used and functionally differentiated models rather than a long tail of rarely-accessed submissions. To further assess the redundancy, we compute pairwise cosine similarity between model cards within each domain using BGE-M3 embeddings. In HuggingBench-1, only 3 domains exhibit substantial redundancy ($> 30\%$ of pairs above 0.9), while most domains remain below 10%. HuggingBench-2 shows an even cleaner pattern, with no domains exceeding the 30% threshold and the majority again below 10%. Across both benchmarks, the average similarity ($0.59 \pm 0.15$) indicates moderate but non-trivial overlap, well below near-duplicate regimes. Beyond the scope of this paper, the CMR framework naturally supports extension to settings where real user feedback is available post-deployment. One particularly promising direction is the construction of a model performance leaderboard from aggregated user preferences, which could complement routing decisions with richer empirical signals.

**Human Evaluation of Dataset Quality.** A potential concern in model routing benchmarks is whether routing labels simply reflect predefined dataset assignments rather than meaningful model-task alignment. However, CARvE and baselines

*Table 8.* Download statistics across benchmarks.

| Benchmark | Median | Max | Min |
|---|---|---|---|
| APIBench | 9,394.5 | 23,269,454 | 1 |
| ToolMMBench | 6,945 | 23,269,454 | 2 |
| HuggingBench-1 | 14,054 | 23,269,454 | 1 |
| HuggingBench-2 | 10,425 | 206,907,723 | 1 |

Distribution of Unique Selected Models by Download Count

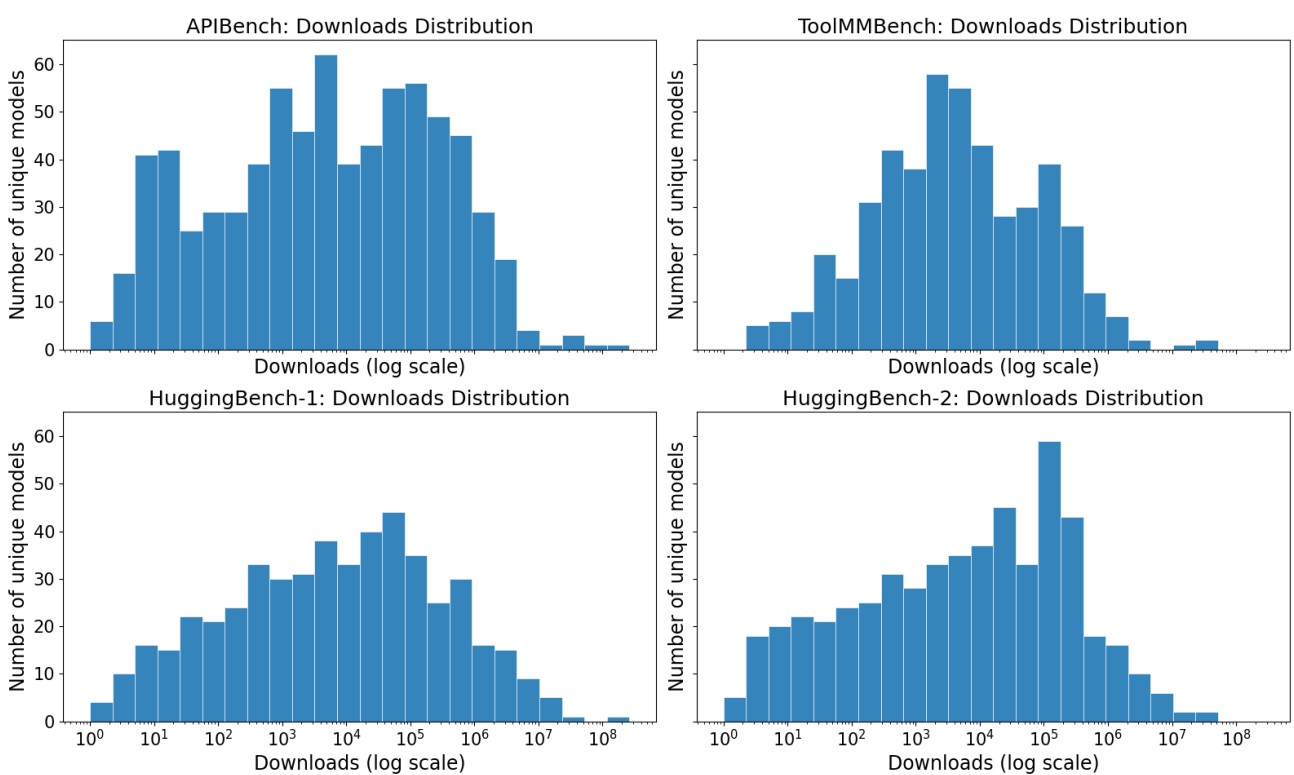

*Figure 5.* Distribution of model downloads across benchmark experiences.

we tested, operates strictly in a pre-inference setting, where routing decisions must be made before executing candidate models. Consequently, post-execution quality signals are unavailable at routing time. Our evaluation protocol therefore follows prior routing benchmarks such as APIBench and ToolMMBench, which similarly formulate routing as predicting the most appropriate model conditioned only on the input query. To assess whether the dataset labels correspond to meaningful model-query alignment, we additionally conducted a human evaluation study. Two independent reviewers evaluated 50 non-overlapping prompt–model pairs across three dimensions: *domain fit*, *model fit*, and *prompt naturalness*, using a 1–5 Likert scale. Results show strong domain alignment ($\mu = 4.12$, with 71% of samples rated $\geq 4$), reasonable model fit ($\mu = 3.76$, 64% rated $\geq 4$), and adequate prompt naturalness ($\mu = 3.61$, 57% rated $\geq 4$). The comparatively lower naturalness score reflects the inherent difficulty of generating fully naturalistic queries via self-instruct methods, a limitation shared by prior benchmarks such as APIBench and ToolMMBench. Overall, these results support that the benchmark captures meaningful functional distinctions between models rather than arbitrary label assignments.

# B. Metrics

## B.1. Model Family Metric

With over 2,000 models, each experience includes very few queries per model, making exact matches rare. As shown in Table 7, we have about **15** queries per model on average, across all experiences. So we found that *model-ID accuracy* can be overly strict. Conversely, *domain accuracy* can be too coarse, as multiple models within the same domain may differ substantially in their capabilities and may not actually solve a given query. To address this, we introduce *model family accuracy*, which provides a middle ground between these two extremes. A model family is defined as a group of closely related models that share the same core architecture and are released in multiple variants (e.g., different quantization levels, backends, or deployment formats). To compute model families, we first normalize model-IDs by stripping out size indicators (e.g., *small*, *medium*, *large*, *xl*) and other minor adjectives. Next, we encode the normalized names into vector representations using a TF-IDF approach. We then compute a pairwise cosine similarity matrix between all model-ID vectors and apply an agglomerative clustering procedure to group together similar normalized names. Each resulting cluster defines a model family, and *model family accuracy* measures whether the predicted model belongs to the same family as the ground-truth model. This metric captures fine-grained functional similarity across models while remaining robust to superficial differences in naming, providing a more meaningful assessment of the router's ability to select an appropriate model in practice. Statistics on model families are reported in Table 7. We now present illustrative examples of model families. In each example, the first identifier denotes the name assigned to the family, while the subsequent list enumerates all individual models belonging to that family.

```
Model Families example

glm-4.6v-flash (9 models):
  - cyankiwi/GLM-4.6V-AWQ-4bit
  - cyankiwi/GLM-4.6V-Flash-AWQ-4bit
  - cyankiwi/GLM-4.6V-Flash-AWQ-8bit
  - lmstudio-community/GLM-4.6V-Flash-MLX-4bit
  - lmstudio-community/GLM-4.6V-Flash-MLX-6bit
  - lmstudio-community/GLM-4.6V-Flash-MLX-8bit
  - unsloth/GLM-4.6V-Flash
  - unsloth/GLM-4.6V-Flash-GGUF
  - zai-org/GLM-4.6V-Flash

glpn-kitti (3 models):
  - sayakpaul/glpn-kitti-finetuned-diode
  - sayakpaul/glpn-kitti-finetuned-diode-221214-123047
  - vinvino02/glpn-kitti

gemma-3n-e4b-it-less-heretic-i1-gguf (3 models):
  - MuXodious/gemma-3n-E4B-it-absolute-heresy-GGUF
  - MuXodious/gemma-3n-E4B-it-less-heretic
  - mradermacher/gemma-3n-E4B-it-less-heretic-i1-GGUF

gemmamed-cardio (1 models):
  - uaritm/gemmamed_cardio

finbert-esg-9-categories (1 models):
  - yiyanghkust/finbert-esg-9-categories

flan-t5 (6 models):
  - google/flan-t5-base
  - google/flan-t5-large
  - google/flan-t5-small
  - google/flan-t5-xl
  - google/flan-t5-xxl
  - pszemraj/flan-t5-large-grammar-synthesis
```

# C. Experimental Setup

## C.1. Training and Evaluation Configuration

In this section, we describe the training and evaluation configuration and hyperparameters used across all experiments. Our CL setup consists of a sequence of four experiences, on each of which a LoRA adapter is fine-tuned while keeping the backbone model frozen.

To limit computational cost, hyperparameter tuning is performed only on the first experience, or alternatively the first time the specific hyperparameters are introduced, for example in Experience 2. The selected hyperparameters are then fixed and reused for training on subsequent experiences. This design choice significantly reduces the expense of repeated hyperparameter searches in later stages, but it may result in **suboptimal performance on experiences 2–4**, as the fixed configuration may not be optimal for all data distributions encountered over time. All hyperparameters used throughout the experiments are reported in Table 9a. For CARvE, Table 10 additionally reports the routing specific settings required for reproducibility, including hard-negative mining frequency and pool sizes, soft-target parameters and the anchoring coefficients used to stabilise model and projection embeddings under hub expansion.

We analyse the token length distribution for each dataset used in the experiments, as shown in Figure 6. Due to differing input characteristics, we adopt separate configurations for zero-shot and RAT experiments, summarised in Table 9b, for maximum sequence length, batch sizes and gradient accumulation steps.

*(a)* Training hyperparameters used across all experiences. While most settings are self-explanatory, Target modules denote the transformer sub-modules to which LoRA adapters are applied. In our setup, LoRA is injected into all major projection layers of the backbone model, including attention and feed-forward components.

| Hyperparameter | Value |
| --- | --- |
| Epochs | 5 |
| Learning rate | 5e-4 |
| Max gradient norm | 1.0 |
| Weight decay | 0.001 |
| Warmup steps | 10 |
| Learning rate scheduler | Linear |
| Optimizer | AdamW |
| Packing | False |
| Group by length | True |
| Completion-only loss | True |
| Label smoothing | 0.05 |
| Metric for best model | Eval loss |
| LoRA rank ($r$) | 32 |
| LoRA alpha | 64 |
| LoRA dropout | 0.05 |
| Target modules | q_proj, k_proj, v_proj, o_proj, gate_proj, down_proj, up_proj |

*(b)* Zero-shot vs. RAT configuration. Comparison of training hyperparameters used for the zero-shot and Retrieval-Augmented Training (RAT) settings. Different batch sizes, gradient accumulation steps, and maximum sequence lengths are adopted to account for the substantially longer prompts in RAT, where the retrieved model card is appended to the user query.

| Parameter | Zero-Shot | RAT |
| --- | --- | --- |
| Batch size | 64 | 32 |
| Gradient accumulation | 2 | 4 |
| Max sequence length | 256 | 1200 |

*(c)* Evaluation decoding and sampling hyperparameters.

| Hyperparameter | Value |
| --- | --- |
| Max new tokens | 128 |
| Temperature | 0.35 |
| Do sampling | True |
| Top-$k$ | 10 |
| Top-$p$ | 0.7 |

*(d)* Hyperparameters for continual learning regularization.

| Method | Hyperparameter | Value |
| --- | --- | --- |
| LoRA + EWC | $\lambda$ | 20,000 |
| CARvE + EWC | $\lambda$ | 20,000 |
| LoRA + LwF | $T$ | 10.0 |
| LoRA + LwF | $\alpha$ | 4.0 |

We now describe the evaluation configuration used across all experiments. During inference, the maximum input sequence length is kept consistent with training, using a maximum length of 256 tokens for zeroshot experiments and 1200 tokens for RAT experiments. This ensures that the evaluation setting matches the context length constraints learned during fine-tuning. All remaining decoding and sampling hyperparameters are fixed across experiments and are reported in Table 9c.

Hyperparameters for continual learning regularization are reported in Table 9d. For EWC-based methods, the Fisher

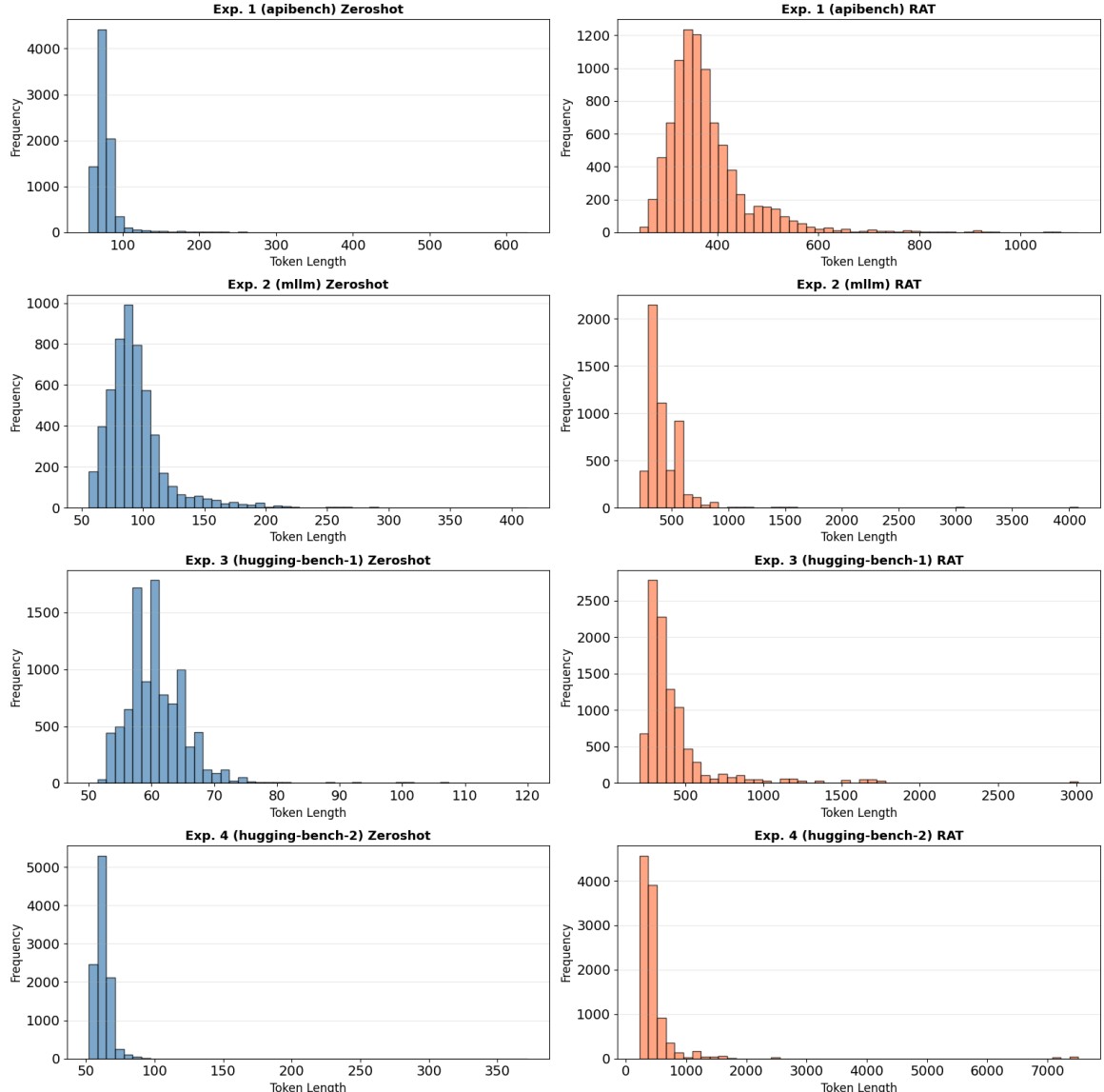

*Figure 6.* Token length distribution across learning experiences. Input lengths for the zero-shot configuration are shown in blue, while those for the RAT configuration are shown in orange. As observed across experiences, most zero-shot prompts are concentrated around 100 tokens. In contrast, RAT inputs are substantially longer, with the majority of prompts falling in the 400–600 token range due to the inclusion of model cards. A small number of outliers with input lengths exceeding 7,000 tokens are also present; these anomalies are primarily caused by poorly formatted or excessively verbose model cards included in the retrieved content.

information matrix is recomputed at the end of each experience. For LwF, distillation targets are generated from the checkpoint obtained after the previous experience. Hyperparameters were selected through preliminary tuning.

### C.2. Prompt Templates

To ensure consistency with the Gorilla routing paradigm, we adopt two prompt templates corresponding to the zero-shot and retrieval-augmented training settings. Both templates are designed to instruct the model to act as an expert model router and to return only a single valid model-ID as output.

In the zero-shot setting, the model receives only the user instruction and is required to predict the most appropriate model without access to any external retrieved information.

| Group | Hyperparameter | Value |
|---|---|---|
| *Global (all runs unless noted)* | | |
| Architecture | Embedding dimension | 1024 |
| Architecture | Pooling | last_token |
| Routing | Temperature ($\tau$) | 0.08 |
| Optimization | Projection LR | $3 \times 10^{-4}$ |
| Optimization | Embedding LR | $5 \times 10^{-5}$ |
| Candidate sets | $K_{\text{total}}$ | 64 (total candidates per example) |
| Candidate sets | $K_{\text{semantic}}$ | 38 (in-domain negatives) |
| Candidate sets | $K_{\text{far}}$ | 10 (out-of-domain negatives) |
| Candidate sets | $K_{\text{hard}}$ | 15 (mined hard negatives) |
| Hard-neg mining | Mining frequency | every 100 steps |
| Hard-neg mining | Hard pool size | 50 |
| Hard-neg mining | Semantic pool size | 1024 |
| Hard-neg mining | Max pool size | 2048 |
| Soft targets | Enabled | true |
| Soft targets | $\epsilon$ | 0.02 |
| Soft targets | $k$ (neighbors) | 10 |
| Batching | Semantic batching | enabled |
| Batching | Domains per batch | 2 |
| *Regime-specific settings* | | |
| APIBench | Loss mode | supervised+router |
| APIBench | Two-phase training | disabled |
| APIBench | Anchoring | disabled |
| mllm_onwards | Loss mode | router |
| mllm_onwards | Two-phase training | enabled; Phase 1 fraction = 0.4; Phase 1 loss = router |
| mllm_onwards | Phase 1 LRs | projection LR $1 \times 10^{-4}$; embedding LR $5 \times 10^{-5}$ |
| mllm_onwards | LM freezing | true |
| mllm_onwards | Replay loss multiplier | 5.0 |
| mllm_onwards | Embedding anchoring | enabled (phase 1); normalised; $\lambda = 10{,}000$ |
| mllm_onwards | Projection anchoring | enabled (phase 1); $\lambda = 50{,}000$ |

*Table 10.* CARVE hyperparameters used in all experiments. "mllm_onwards" denotes the multi-modal experiences where two-phase training and anchoring are enabled.

---

**Zero-shot prompt template**

You are Gorilla, an expert API model router. Read the ###Instruction and ###Input below and return **ONLY** a single model name. Do not invent model names. Do not return anything else.

---

In the retrieval-augmented training setting, the user instruction is augmented with a retrieved model card. The model is explicitly informed that the retrieved information may be noisy or irrelevant and must decide whether to leverage it when predicting the target model-ID.

---

**Retrieval-augmented training (RAT) prompt template**

You are Gorilla, an expert API model router. Read the ###Instruction and ###Input below and return **ONLY** a single model name. Do not invent model names. Do not return anything else.
In this task, you have access to suggested API information retrieved from a knowledge base. Not all retrieved information is relevant or accurate. Use your judgment to decide whether to incorporate it into your response. Retrieved API information is provided after the instruction under the <Reference API> tag.

## C.3. Domain–Model Coreset Replay and Farthest-Point Sampling Details

This appendix provides the hyperparameters and procedure used to construct the *domain–model coreset replay* buffer, including the quotas, caps, and embedding space used for *farthest-point sampling (FPS)* with cosine distance.

### C.3.1. REPLAY BUDGET, DOMAIN QUOTAS, AND MODEL CAPS

Let $\mathcal{D}$ denote the set of domains seen so far, and let $\mathcal{S}_t$ denote the union of all training examples observed up to experience $t$. We construct a replay buffer $\mathcal{B}_t \subseteq \mathcal{S}_{t-1}$ whose total size is determined either by a replay ratio $\rho \in (0, 1]$ or a fixed replay budget $B$:

$$|\mathcal{B}_t| = \begin{cases} \lfloor \rho \, |\mathcal{S}_{t-1}| \rfloor & \text{if replay ratio is used,} \\ B & \text{if a fixed number of replay samples is used.} \end{cases} \tag{1}$$

We allocate this budget across domains via per-domain quotas. Let $n_d$ be the number of examples from domain $d$ in $\mathcal{S}_{t-1}$. We first compute a proportional allocation $q_d \propto n_d$, then clamp it using a minimum and (optionally) a maximum:

$$q_d \leftarrow \min \big( q_d^{\max}, \, \max(q^{\min}, \, q_d) \big), \tag{2}$$

where $q^{\min} = $ `replay_min_per_domain` is a floor ensuring each domain is represented (default: 5), and $q_d^{\max} = $ `replay_max_per_domain` is an optional cap (default: none). If a domain contains fewer than $q_d$ available examples, we select all its examples and redistribute any leftover budget to other domains.

Within each domain, we additionally cap the number of stored examples per model-ID to avoid over-representing high-frequency models. Let $n_{d,m}$ denote the number of examples for model $m$ within domain $d$. We apply a per-model cap $k_m = \min($ `replay_max_per_model`$, n_{d,m})$ (default: `replay_max_per_model = 3`). When a domain quota exceeds the sum of its capped per-model selections, we fill the remaining slots using FPS over the domain pool while enforcing diversity relative to already selected points.

### C.3.2. EMBEDDING SPACE FOR FPS

FPS is performed in a fixed embedding space using cosine distance. For each example $x_i$ (a training prompt), we compute an embedding $\mathbf{e}_i \in \mathbb{R}^p$ with an external sentence-embedding model and $\ell_2$-normalise it, so cosine distance is equivalent to Euclidean distance in the normalised space. We use one of two embedding sources:

- **Sentence-transformer** (default): `all-mpnet-base-v2`.

- **FlagEmbedding**: BGE-M3, matching the retriever used in retrieval-only baselines.

The choice is controlled by `replay_embedding_source`. Embeddings are computed for all candidates in $\mathcal{S}_{t-1}$ and cached for reuse during coreset construction.

### C.3.3. FARTHEST-POINT SAMPLING PROCEDURE

Given a set of candidate indices $I$ and their embeddings $\{\mathbf{e}_i\}_{i \in I}$, FPS greedily selects a subset $S$ of size $k$ that maximises coverage by repeatedly adding the point with the largest minimum distance to the current selected set:

$$S \leftarrow \{i_0\} \quad \text{(random initial point)} \tag{3}$$

$$i^* \leftarrow \arg \max_{i \in I \setminus S} \min_{j \in S} \big( 1 - \langle \mathbf{e}_i, \mathbf{e}_j \rangle \big) \tag{4}$$

$$S \leftarrow S \cup \{i^*\} \quad \text{until } |S| = k. \tag{5}$$

The initial point $i_0$ is sampled using a fixed random seed (set from `train_config.seed`) to ensure reproducibility. When filling remaining quota within a domain, we optionally pass an `existing_selected` set to ensure new selections remain diverse relative to already selected examples in that domain.

### C.3.4. HYPERPARAMETERS

Table 11 lists the hyperparameters controlling coreset construction and FPS.

*Table 11.* Hyperparameters controlling domain–model coreset replay and FPS.

| Hyperparameter | Default | Role |
|---|---|---|
| `replay_percentage` ($\rho$) | varies | Sets total replay budget as a fraction of past data. |
| `replay_num_samples` ($B$) | None | Optional fixed replay budget (overrides `replay_percentage`). |
| `replay_min_per_domain` ($q^{\min}$) | 5 | Minimum quota per domain. |
| `replay_max_per_domain` ($q_d^{\max}$) | None | Optional maximum quota per domain. |
| `replay_max_per_model` | 3 | Per-model cap within each domain; sets $k$ for per-model FPS selection. |
| `replay_embedding_source` | `sentence_transformer` | Embedding space for FPS (`all-mpnet-base-v2` or BGE-M3). |
| `seed` | config-dependent | Controls FPS initial point selection for determinism. |

## C.4. Extending the router embedding space under registry expansion

CARvE maintains a *persistent model registry* whose size grows as new models are introduced in each experience. The router operates in a fixed-dimensional embedding space $\mathbb{R}^k$ and consists of two learned components: (i) a model embedding table and (ii) a prompt-projection head.

**Router architecture.** Let $k$ denote the router embedding dimension and let $M_t = |\mathcal{M}_t|$ be the number of registered models after experience $t$. The router comprises:

- **Model embedding table** $\mathbf{E}_t \in \mathbb{R}^{M_t \times k}$, implemented as `nn.Embedding(M_t, k)`, containing one embedding vector per model-ID.

- **Prompt projection** $\mathbf{W}_t \in \mathbb{R}^{h \times k}$, implemented as `nn.Linear(h, k, bias=False)`, which projects backbone LM hidden states of dimension $h$ into the router embedding space.

At inference, a prompt embedding $\mathbf{z} \in \mathbb{R}^k$ is obtained by projecting the LM representation, and routing is performed by temperature-scaled cosine similarity between $\mathbf{z}$ and the rows of $\mathbf{E}_t$.

**Initialisation.** When a router is first instantiated, both $\mathbf{E}_0$ and $\mathbf{W}_0$ are initialised with Xavier-uniform initialisation.

**Registry expansion when starting a new experience.** When moving from experience $t-1$ to $t$, the registry expands from $M_{t-1}$ to $M_t$ to incorporate newly added model-IDs. The embedding *dimension* $k$ is held fixed; only the number of rows in the model embedding table changes. We construct a new router instance with an expanded embedding table $\mathbf{E}_t \in \mathbb{R}^{M_t \times k}$ and update parameters as follows:

1. **Copy overlapping rows (preserve old embeddings).** Let $M_{\text{old}} = M_{t-1}$ and $M_{\text{new}} = M_t$. For the overlap $1{:}M_{\text{old}}$, we copy the learned embeddings:
$$\mathbf{E}_t[1{:}M_{\text{old}}] \leftarrow \mathbf{E}_{t-1}[1{:}M_{\text{old}}].$$

2. **Initialise new rows.** For newly added models $m \in \{M_{\text{old}}{+}1, \ldots, M_{\text{new}}\}$, we initialise their embedding rows with Xavier-uniform:
$$\mathbf{E}_t[M_{\text{old}}{+}1{:}M_{\text{new}}] \sim \text{XavierUniform}.$$

3. **Projection head continuity.** The prompt-projection matrix is loaded from the previous checkpoint to preserve compatibility with the existing embedding space:
$$\mathbf{W}_t \leftarrow \mathbf{W}_{t-1}.$$

(When projection anchoring is enabled during training, $\mathbf{W}_t$ is additionally regularised to remain close to $\mathbf{W}_{t-1}$; see Section 2)

# D. Results

In this section, we provide an in-depth analysis of all the experiments conducted in this work. We first summarize the experimental settings in Table 12. In particular, *label snapping* refers to the post-processing step introduced in Section 3.2, where the string generated by a LoRA-based method is mapped to the closest valid model-ID in the registry using a Levenshtein similarity metric with a threshold of 0.8. Since label snapping is not applied uniformly across all methods, Table 12 explicitly reports whether this mechanism is used, together with the use of domain labels and model cards during training.

In the following subsections, we present extended results for the *retrieval-only* experiment, the comparison between *zero-shot* and *RAT* settings, the *model merging* experiment, and our main evaluation, where we compare different continual learning baselines against our proposed method, *CARvE*.

*Table 12.* Experimental Setting Summary. Label Snapping denotes the use of label snapping as described above at test time. Model cards denotes the use of model cards during training. Domain label represents the use of domain labels during training. Domain list at test time denotes the use of a list of possible domains in the training data passed to the model at test time, applying only to the zero-shot method, HuggingGPT. For CARvE and Replay experiments, results are reported under multiple replay ratios (5%, 10%, and 20%)

| Experiment | Label Snapping | Model Cards | Domain Label | Domain list at test time |
|---|---|---|---|---|
| Retrieval-Only | No | Yes | No | No |
| Gorilla with RAG | Yes | Yes | No | No |
| HuggingGPT | No | No | No | Yes |
| Zero-Shot w/ Random Replay | Yes | No | No | No |
| RAT w/ Random Replay | Yes | Yes | No | No |
| Model Merging | Yes | No | No | No |
| Sequential Fine-tuning | Yes | No | No | No |
| Joint Training Zero-Shot | Yes | No | No | No |
| Joint Training RAT | Yes | Yes | No | No |
| **CARvE** | **No** | **No** | **Yes** | **No** |

## D.1. Retrieval-Only Extended Results

In Table 13, we provide extended results for the retrieval-only preliminary experiments across all evaluated experiences. As can be observed from the results, BGE-M3 consistently outperforms all the other retrievers, achieving the best performance across the considered settings in model-ID accuracy, domain accuracy and model-family accuracy. In contrast, BM25, due to its simplicity and reliance on sparse lexical matching, represents the weakest baseline and exhibits the lowest performance on average across all experiences.

*Table 13.* Retrieval-Only results across Experience 1–4. M: Model-ID accuracy, F: Model family accuracy, D: domain accuracy (%). Results are computed directly from the retrieved model predictions, without applying label snapping. *Average* denotes row-wise averages over seen experiences; *Mean* denotes column-wise averages within each setting.

| Retrievers | Experience 1 | | | Experience 2 | | | Experience 3 | | | Experience 4 | | | Average | | |
|---|---|---|---|---|---|---|---|---|---|---|---|---|---|---|---|
| | M | F | D | M | F | D | M | F | D | M | F | D | M | F | D |
| *BM25* | | | | | | | | | | | | | | | |
| Indexed(Exp 1) | 5.3 | 6.6 | 17.5 | – | – | – | – | – | – | – | – | – | 5.3 | 6.6 | 17.5 |
| Indexed(Exp 1-2) | 4.7 | 5.5 | 17.8 | 1.7 | 2.0 | 15.5 | – | – | – | – | – | – | 3.2 | 3.8 | 16.6 |
| Indexed(Exp 1-3) | 4.6 | 5.4 | 19.0 | 1.5 | 1.6 | 13.5 | 3.6 | 4.4 | 11.7 | – | – | – | 3.2 | 3.8 | 14.7 |
| Indexed(Exp 1-4) | 3.6 | 4.6 | 17.0 | 1.4 | 1.5 | 13.2 | 2.9 | 3.5 | 11.2 | 4.8 | 5.5 | 15.2 | 3.2 | 3.8 | 14.1 |
| Mean | 4.6 | 5.5 | 17.8 | 1.5 | 1.7 | 14.1 | 3.2 | 4.0 | 11.4 | 4.8 | 5.5 | 15.2 | 3.5 | 4.2 | 14.6 |
| *SentenceTransformer* | | | | | | | | | | | | | | | |
| Indexed(Exp 1) | 15.0 | 20.6 | 56.9 | – | – | – | – | – | – | – | – | – | 15.0 | 20.6 | 56.9 |
| Indexed(Exp 1-2) | 12.5 | 17.0 | 54.6 | 11.4 | 13.0 | 47.2 | – | – | – | – | – | – | 11.9 | 15.0 | 50.9 |
| Indexed(Exp 1-3) | 11.0 | 14.3 | 52.1 | 9.8 | 11.0 | 45.7 | 9.9 | 11.5 | 36.4 | – | – | – | 10.2 | 12.3 | 44.7 |
| Indexed(Exp 1-4) | 10.4 | 13.6 | 50.2 | 8.9 | 10.1 | 42.5 | 8.1 | 9.5 | 34.6 | 11.8 | 14.3 | 37.5 | 9.8 | 11.9 | 41.2 |
| Mean | 12.2 | 16.4 | 53.4 | 10.0 | 11.4 | 45.1 | 9.0 | 10.5 | 35.5 | 11.8 | 14.3 | 37.5 | 10.8 | 13.1 | 42.9 |
| *SPLADE* | | | | | | | | | | | | | | | |
| Indexed(Exp 1) | 5.0 | 8.0 | 30.0 | – | – | – | – | – | – | – | – | – | 5.0 | 8.0 | 30.0 |
| Indexed(Exp 1-2) | 2.5 | 4.5 | 27.0 | 6.4 | 7.2 | 29.9 | – | – | – | – | – | – | 4.5 | 5.8 | 28.4 |
| Indexed(Exp 1-3) | 1.7 | 2.8 | 26.4 | 4.6 | 5.1 | 26.0 | 6.0 | 7.5 | 25.9 | – | – | – | 4.1 | 5.1 | 26.1 |
| Indexed(Exp 1-4) | 1.6 | 2.4 | 25.6 | 4.2 | 4.6 | 22.6 | 4.8 | 5.8 | 24.9 | 8.8 | 11.0 | 28.7 | 4.9 | 6.0 | 25.4 |
| Mean | 2.7 | 4.4 | 27.2 | 5.1 | 5.6 | 26.2 | 5.4 | 6.6 | 25.4 | 8.8 | 11.0 | 28.7 | 5.5 | 6.9 | 26.9 |
| *BGE-M3* | | | | | | | | | | | | | | | |
| Indexed(Exp 1) | 22.6 | 27.7 | 60.0 | – | – | – | – | – | – | – | – | – | 22.6 | 27.7 | 60.0 |
| Indexed(Exp 1-2) | 20.8 | 24.0 | 58.2 | 12.4 | 14.2 | 47.2 | – | – | – | – | – | – | 16.6 | 19.1 | 52.7 |
| Indexed(Exp 1-3) | 18.8 | 21.6 | 54.8 | 11.0 | 12.6 | 44.1 | 11.3 | 13.3 | 34.8 | – | – | – | 13.7 | 15.8 | 44.6 |
| Indexed(Exp 1-4) | 17.4 | 20.1 | 52.6 | 10.3 | 11.7 | 43.8 | 9.6 | 10.8 | 34.3 | 13.0 | 16.7 | 39.8 | 12.6 | 14.8 | 42.6 |
| Mean | 19.9 | 23.4 | 56.4 | 11.2 | 12.8 | 45.0 | 10.4 | 12.0 | 34.6 | 13.0 | 16.7 | 39.8 | **13.6** | **16.2** | **44.0** |

## D.2. Zero-Shot vs RAT Extended Results

In this experiment, we compare a zero-shot routing strategy against a Retrieval-Augmented Training (RAT) approach under a continual learning setup with 10% replay. In the RAT configuration, the BGE-M3 retriever is used to provide candidate model information to the routing model.

Results for all four experiences are reported in Table 14, while Table 15 reports the corresponding results before applying *label snapping* with a threshold of 0.8.

Each row in the tables corresponds to an *LoRA adapter* trained up to a given experience (e.g., Trained(Exp 1–3) denotes an adapter trained incrementally from Experience 1 to 3). The *Average* columns report, for each adapter, the mean performance across the experiences it has been trained on, computed as row-wise averages over the available experiences and excluding missing entries.

The *Mean* rows report column-wise averages across all adapters within the same setting (Zero-shot or RAT) for a given experience, providing an aggregate view of performance at each stage of continual learning. The bold values highlight the best-performing setting across Zero-shot and RAT, based on the average of the *Mean* scores over all experiences. Overall, the zero-shot configuration achieves higher mean performance than RAT for model-ID accuracy (M), model family accuracy (F), and domain accuracy (D). This trend suggests that incorporating retrieved candidate models may introduce additional noise in some cases, which can negatively affect routing decisions. Based on these results, we adopt the zero-shot configuration as the default setting for all subsequent experiments.

*Table 14.* Zero-Shot versus RAT results across Experiences 1–4 with label snapping. M: model-ID accuracy, F: model family accuracy, D: domain accuracy (%). Rows correspond to LoRA adapters trained up to the indicated experience. *Average* denotes row-wise averages over seen experiences; *Mean* denotes column-wise averages within each setting. Results are reported for a single seed after label snapping (threshold 0.8). Bold indicates the best-performing method based on the average performance across all experiences. RAT uses BGE-M3.

| Setting | Experience 1 | | | Experience 2 | | | Experience 3 | | | Experience 4 | | | Average | | |
|---|---|---|---|---|---|---|---|---|---|---|---|---|---|---|---|
| | M | F | D | M | F | D | M | F | D | M | F | D | M | F | D |
| *Zero-shot with Replay 10%* | | | | | | | | | | | | | | | |
| Indexed(Exp 1) | 25.7 | 36.2 | 79.9 | – | – | – | – | – | – | – | – | – | 25.7 | 36.2 | 79.9 |
| Indexed(Exp 1-2) | 19.8 | 29.3 | 72.1 | 69.5 | 71.1 | 88.8 | – | – | – | – | – | – | 44.6 | 50.2 | 80.4 |
| Indexed(Exp 1-3) | 17.0 | 25.0 | 65.9 | 48.9 | 51.4 | 75.6 | 45.0 | 58.9 | 82.0 | – | – | – | 37.0 | 45.1 | 74.5 |
| Indexed(Exp 1-4) | 15.9 | 22.8 | 63.2 | 44.0 | 46.2 | 70.0 | 18.8 | 27.6 | 68.0 | 49.6 | 60.6 | 81.7 | 32.1 | 39.3 | 70.7 |
| Mean | 19.6 | 28.3 | 70.3 | 54.1 | 56.2 | 78.1 | 31.9 | 43.2 | 75.0 | 49.6 | 60.6 | 81.7 | **38.8** | **47.1** | **76.3** |
| *RAT with Replay 10%* | | | | | | | | | | | | | | | |
| Indexed(Exp 1) | 18.8 | 30.2 | 74.4 | – | – | – | – | – | – | – | – | – | 18.8 | 30.2 | 74.4 |
| Indexed(Exp 1-2) | 13.8 | 21.2 | 67.2 | 65.1 | 67.0 | 87.9 | – | – | – | – | – | – | 39.4 | 44.1 | 77.6 |
| Indexed(Exp 1-3) | 11.1 | 18.8 | 61.8 | 46.0 | 48.1 | 72.8 | 46.1 | 60.1 | 81.4 | – | – | – | 34.4 | 42.3 | 72.0 |
| Indexed(Exp 1-4) | 10.3 | 17.1 | 59.7 | 42.3 | 44.0 | 68.0 | 16.3 | 25.0 | 62.9 | 45.0 | 55.8 | 78.5 | 28.5 | 35.5 | 67.3 |
| Mean | 13.5 | 21.8 | 65.8 | 51.1 | 53.0 | 76.2 | 31.2 | 42.6 | 72.2 | 45.0 | 55.8 | 78.5 | 35.2 | 43.3 | 73.2 |

*Table 15.* Zero-Shot versus RAT results across Experiences 1–4 **without label snapping**. M: model-ID accuracy, F: model family accuracy, D: domain accuracy (%). Rows correspond to adapters trained up to the indicated experience. *Average* denotes row-wise averages over seen experiences; *Mean* denotes column-wise averages within each setting. Here, we report a single run per experiment. Bold indicates the best-performing method based on the average performance across all experiences. RAT uses BGE-M3.

| Setting | Experience 1 | | | Experience 2 | | | Experience 3 | | | Experience 4 | | | Average | | |
|---|---|---|---|---|---|---|---|---|---|---|---|---|---|---|---|
| | M | F | D | M | F | D | M | F | D | M | F | D | M | F | D |
| *Zero-shot with Replay 10%* | | | | | | | | | | | | | | | |
| Indexed(Exp 1) | 25.7 | 36.1 | 79.6 | – | – | – | – | – | – | – | – | – | 25.7 | 36.1 | 79.6 |
| Indexed(Exp 1-2) | 19.8 | 28.8 | 70.5 | 69.5 | 71.1 | 88.5 | – | – | – | – | – | – | 44.6 | 49.9 | 79.5 |
| Indexed(Exp 1-3) | 16.7 | 24.7 | 64.7 | 48.0 | 50.4 | 74.3 | 45.0 | 58.9 | 81.9 | – | – | – | 36.6 | 44.7 | 73.6 |
| Indexed(Exp 1-4) | 15.7 | 22.5 | 62.2 | 43.6 | 45.7 | 68.7 | 18.7 | 27.3 | 67.5 | 49.4 | 60.4 | 81.5 | 31.9 | 39.0 | 70.0 |
| Mean | 19.5 | 28.0 | 69.2 | 53.7 | 55.7 | 77.2 | 31.8 | 43.1 | 74.7 | 49.4 | 60.4 | 81.5 | **38.6** | **46.8** | **75.7** |
| *RAT with Replay 10%* | | | | | | | | | | | | | | | |
| Indexed(Exp 1) | 18.3 | 29.6 | 73.4 | – | – | – | – | – | – | – | – | – | 18.3 | 29.6 | 73.4 |
| Indexed(Exp 1-2) | 13.5 | 20.5 | 65.2 | 65.1 | 66.9 | 87.5 | – | – | – | – | – | – | 39.3 | 43.7 | 76.3 |
| Indexed(Exp 1-3) | 10.6 | 18.0 | 60.1 | 45.1 | 47.1 | 70.7 | 46.1 | 60.1 | 81.4 | – | – | – | 33.9 | 41.7 | 70.7 |
| Indexed(Exp 1-4) | 9.8 | 16.4 | 58.2 | 41.4 | 43.1 | 66.1 | 16.0 | 24.7 | 62.3 | 44.9 | 55.7 | 78.3 | 28.0 | 35.0 | 66.2 |
| Mean | 13.0 | 21.1 | 64.2 | 50.5 | 52.4 | 74.8 | 31.0 | 42.4 | 71.8 | 44.9 | 55.7 | 78.3 | 34.9 | 42.9 | 72.3 |

## D.3. Model Merging Extended Results

In this experiment, we study strategies for merging multiple LoRA adapters, each trained independently on distinct experiences, into a single consolidated model. We consider three representative merging approaches: TIES, DARE, and simple arithmetic averaging. For both TIES and DARE, all adapters are assigned equal weights of 1.0, with TIES using a sparsity density of 0.3 and DARE a density of 0.8. The arithmetic mean performs element-wise averaging of the adapter weights. Extended results for all merging strategies, both with and without label snapping, are reported in Tables 16 and 17. Each row corresponds to a merged model obtained from adapters trained up to the indicated experience, with the *Average* columns reporting row-wise averages over the experiences included in the merged model and the *Mean* rows reporting column-wise averages across merged models for a given experience.

Overall, the three merging strategies show a clear trend in which performance decreases as more experiences are merged. TIES consistently achieves slightly higher mean accuracies across model, family, and domain metrics compared to DARE

and arithmetic averaging, and based on this observation, we select TIES as the preferred merging strategy for subsequent experiments.

*Table 16.* Comparison of model merging strategies across Experiences 1–4 with label snapping. M: model-ID accuracy, F: model family accuracy, D: domain accuracy (%). Rows correspond to merged models trained up to the indicated experience. *Average* and *Mean* denote row-wise and column-wise averages, respectively. Bold indicates the best-performing method based on the average performance across all experiences.

| Setting | Experience 1 | | | Experience 2 | | | Experience 3 | | | Experience 4 | | | Average | | |
|---|---|---|---|---|---|---|---|---|---|---|---|---|---|---|---|
| | M | F | D | M | F | D | M | F | D | M | F | D | M | F | D |
| *TIES* | | | | | | | | | | | | | | | |
| Indexed(Exp 1) | 26.4 | 37.5 | 80.2 | – | – | – | – | – | – | – | – | – | 26.4 | 37.5 | 80.2 |
| Indexed(Exp 1-2) | 13.0 | 20.0 | 57.4 | 23.2 | 25.5 | 53.4 | – | – | – | – | – | – | 18.1 | 22.8 | 55.4 |
| Indexed(Exp 1-3) | 5.4 | 9.5 | 37.7 | 11.0 | 12.4 | 36.2 | 7.7 | 12.7 | 31.3 | – | – | – | 8.0 | 11.5 | 35.1 |
| Indexed(Exp 1-4) | 3.0 | 6.3 | 24.8 | 5.3 | 5.9 | 22.4 | 4.3 | 11.3 | 33.7 | 3.9 | 6.5 | 27.5 | 4.1 | 7.5 | 27.1 |
| Mean | 12.0 | 18.3 | 50.0 | 13.2 | 14.6 | 37.3 | 6.0 | 12.0 | 32.5 | 3.9 | 6.5 | 27.5 | **8.8** | **12.8** | **36.8** |
| *DARE* | | | | | | | | | | | | | | | |
| Indexed(Exp 1) | 26.9 | 38.1 | 81.0 | – | – | – | – | – | – | – | – | – | 26.9 | 38.1 | 81.0 |
| Indexed(Exp 1-2) | 11.0 | 15.9 | 52.4 | 27.9 | 29.5 | 55.3 | – | – | – | – | – | – | 19.4 | 22.7 | 53.8 |
| Indexed(Exp 1-3) | 5.3 | 8.8 | 34.6 | 15.0 | 16.7 | 36.2 | 5.5 | 9.5 | 26.6 | – | – | – | 8.6 | 11.7 | 32.5 |
| Indexed(Exp 1-4) | 2.0 | 4.4 | 14.0 | 8.9 | 9.5 | 21.6 | 2.9 | 8.1 | 24.4 | 1.9 | 4.0 | 16.8 | 3.9 | 6.5 | 19.2 |
| Mean | 11.3 | 16.8 | 45.5 | 17.3 | 18.6 | 37.7 | 4.2 | 8.8 | 25.5 | 1.9 | 4.0 | 16.8 | 8.7 | 12.1 | 31.4 |
| *Arithmetic Mean* | | | | | | | | | | | | | | | |
| Indexed(Exp 1) | 26.4 | 37.5 | 80.7 | – | – | – | – | – | – | – | – | – | 26.4 | 37.5 | 80.7 |
| Indexed(Exp 1-2) | 11.8 | 17.1 | 51.9 | 26.5 | 28.2 | 55.7 | – | – | – | – | – | – | 19.1 | 22.6 | 53.8 |
| Indexed(Exp 1-3) | 5.8 | 10.8 | 39.3 | 14.6 | 16.2 | 41.3 | 4.2 | 8.1 | 28.7 | – | – | – | 8.2 | 11.7 | 36.4 |
| Indexed(Exp 1-4) | 3.1 | 6.6 | 23.4 | 7.8 | 8.5 | 23.1 | 3.3 | 9.8 | 28.7 | 1.8 | 3.3 | 21.3 | 4.0 | 7.0 | 24.1 |
| Mean | 11.8 | 18.0 | 48.8 | 16.3 | 17.6 | 40.0 | 3.8 | 9.0 | 28.7 | 1.8 | 3.3 | 21.3 | 8.4 | 12.0 | 34.7 |

*Table 17.* Comparison of model merging strategies across Experiences 1–4 **without label snapping**. M: model-ID accuracy, F: model family accuracy, D: domain accuracy (%). Rows correspond to merged models trained up to the indicated experience. *Average* and *Mean* denote row-wise and column-wise averages, respectively. Bold indicates the best-performing method based on the average performance across all experiences.

| Setting | Experience 1 | | | Experience 2 | | | Experience 3 | | | Experience 4 | | | Average | | |
|---|---|---|---|---|---|---|---|---|---|---|---|---|---|---|---|
| | M | F | D | M | F | D | M | F | D | M | F | D | M | F | D |
| *TIES* | | | | | | | | | | | | | | | |
| Indexed(Exp 1) | 26.4 | 37.5 | 80.2 | – | – | – | – | – | – | – | – | – | 26.4 | 37.5 | 80.2 |
| Indexed(Exp 1-2) | 12.2 | 18.5 | 51.3 | 23.0 | 25.1 | 49.1 | – | – | – | – | – | – | 17.6 | 21.8 | 50.2 |
| Indexed(Exp 1-3) | 5.0 | 8.4 | 30.2 | 10.4 | 11.1 | 30.2 | 6.0 | 10.1 | 22.6 | – | – | – | 7.1 | 9.9 | 27.7 |
| Indexed(Exp 1-4) | 2.7 | 5.2 | 20.1 | 5.2 | 5.7 | 15.2 | 3.0 | 7.9 | 21.6 | 2.7 | 4.3 | 15.9 | 3.4 | 5.8 | 18.2 |
| Mean | 11.6 | 17.4 | 45.4 | 12.9 | 14.0 | 31.5 | 4.5 | 9.0 | 22.1 | 2.7 | 4.3 | 15.9 | **7.9** | **11.2** | **28.7** |
| *DARE* | | | | | | | | | | | | | | | |
| Indexed(Exp 1) | 26.9 | 37.9 | 80.9 | – | – | – | – | – | – | – | – | – | 26.9 | 37.9 | 80.9 |
| Indexed(Exp 1-2) | 10.5 | 14.4 | 47.3 | 26.8 | 28.2 | 50.3 | – | – | – | – | – | – | 18.6 | 21.3 | 48.8 |
| Indexed(Exp 1-3) | 4.8 | 8.0 | 25.8 | 14.6 | 15.6 | 29.9 | 3.9 | 7.2 | 18.7 | – | – | – | 7.8 | 10.3 | 24.8 |
| Indexed(Exp 1-4) | 1.6 | 3.3 | 9.6 | 8.2 | 8.4 | 15.7 | 1.7 | 5.7 | 14.3 | 1.4 | 2.5 | 8.1 | 3.2 | 5.0 | 11.9 |
| Mean | 11.0 | 15.9 | 40.9 | 16.5 | 17.4 | 32.0 | 2.8 | 6.4 | 16.5 | 1.4 | 2.5 | 8.1 | 7.9 | 10.5 | 24.4 |
| *Arithmetic Mean* | | | | | | | | | | | | | | | |
| Indexed(Exp 1) | 26.4 | 37.4 | 80.6 | – | – | – | – | – | – | – | – | – | 26.4 | 37.4 | 80.6 |
| Indexed(Exp 1-2) | 11.1 | 15.9 | 48.1 | 25.5 | 27.1 | 50.2 | – | – | – | – | – | – | 18.3 | 21.5 | 49.2 |
| Indexed(Exp 1-3) | 5.0 | 8.7 | 30.8 | 13.6 | 15.0 | 34.4 | 3.4 | 6.8 | 20.5 | – | – | – | 7.3 | 10.2 | 28.6 |
| Indexed(Exp 1-4) | 2.8 | 5.0 | 18.8 | 7.7 | 8.0 | 18.0 | 2.0 | 6.0 | 17.7 | 1.1 | 2.1 | 12.9 | 3.4 | 5.3 | 16.9 |
| Mean | 11.3 | 16.8 | 44.6 | 15.6 | 16.7 | 34.2 | 2.7 | 6.4 | 19.1 | 1.1 | 2.1 | 12.9 | 7.7 | 10.5 | 27.7 |

### D.4. Baseline Experiments Extended Results

In Tables 18 and 19 we report the extended results for Experiences 1–4, comparing different setups. We present accuracy at three levels: model (M), family (F), and domain (D), along with forgetting metrics (%) computed across all experiences. Results are shown both with and without label snapping to highlight the effect of label alignment on continual learning performance.

**Replay-based and Sequential Finetuning strategies.** Increasing the replay buffer size from 5% to 20% consistently improves accuracy across M, F, and D metrics while reducing forgetting. Domain-level accuracy benefits particularly from larger replay percentages, indicating that retaining samples from previous experiences mitigates catastrophic forgetting at finer-grained levels. Sequential finetuning, in contrast, exhibits severe forgetting, especially in later experiences.

**Merging with TIES.** The TIES merging strategy shows lower retention of previously learned experiences than replay-based strategies. Accuracy on each experience decreases progressively as new experiences are merged, resulting in substantial drops in performance. This trend indicates that TIES is less effective at preserving information from multiple sequential experiences in a continual learning setting.

**Upper Bound Settings: Cumulative, From Scratch, and Joint Training.** The *Cumulative* setting incrementally trains LoRA adapters across experiences. We first train an adapter on Experience 1, then reuse it to train on the combined data from Experiences 1 and 2, subsequently extending to Experiences 1–3, and so on. The *From Scratch* setting trains each adapter independently from random initialization on the combined experiences, without leveraging previously learned adapters. The two *Joint Training* variants train a single adapter on a unified dataset containing all experiences simultaneously, either with *RAT* or zero-shot. All four settings serve as *upper bounds* in the continual learning scenario.

**Retrieval-Based Methods.** We consider two retrieval-based approaches. The *Retrieval-Only* method uses the BGE-M3 (Chen et al., 2024a) model to search a growing dataset of experiences. This approach is evaluated only in the *without label snapping* setting, since label alignment is not required. In contrast, *Gorilla (Patil et al., 2024) with RAG* trains Gorilla on the

first experience only and uses BGE to retrieve the most relevant model for a given query, which is then passed to Gorilla to predict the correct output. Results indicate that *Retrieval-Only* maintains low accuracy across all experiences, including domain-level predictions, reflecting its inability to adapt to newly added experiences. Similarly, *Gorilla with RAG* does not benefit from retrieval: as experiences accumulate, Gorilla fails to return the new models, resulting in declining performance on later experiences.

**HuggingGPT-Style Upper Bounded by Retrieval** *HuggingGPT-Style* (Shen et al., 2023) routes each query by selecting the most appropriate domain from a predefined set of available domains, without predicting a model-ID directly. We evaluate only this domain-selection step, reporting domain accuracy alone. Were one to complete the full HuggingGPT pipeline by pairing domain predictions with a retrieval model to produce final model-ID predictions, the resulting model-ID accuracy would be upper bounded by BGE-M3 retrieval with a ground-truth domain oracle—the performance ceiling reported in Table 19. This method is evaluated in the *without label snapping* setting.

**Impact of Label Snapping.** Label snapping has a limited impact on performance. For replay 10%, averaging over the four experiences, model accuracy increases from 39.05% to 39.25%, corresponding to a gain of only 0.2 percentage points. Model family accuracy increases from 47.35% to 47.65%, with a gain of 0.3 points, while domain accuracy increases from 75.90% to 76.43%, with a gain of 0.53 points. For the merging setting, where TIES is reported, label snapping improves model accuracy from 7.9% to 8.8%, model family accuracy from 11.2% to 12.8%, and domain accuracy from 28.7% to 36.8%, corresponding to gains of 0.9, 1.6, and 8.1 percentage points, respectively. Overall, these results suggest that label snapping provides only marginal improvements in most cases. The larger gain observed for domain-level accuracy in TIES is expected: when model accuracy is very low, even small improvements in correctly identifying individual models can translate into larger changes at the domain level, since the number of domains is much smaller than the number of model names.

*Table 18.* Extended results for Experiments 1–4 with label snapping. For each experience, we report accuracy for Model-ID (M), Model-family (F), and Domain (D) (%). The final block reports forgetting (%) for the same metrics, computed across sequential experiences. Unless otherwise specified, all methods use LLaMA2-7B as the backbone model.

| Setting | Experience 1 | | | Experience 2 | | | Experience 3 | | | Experience 4 | | | Forgetting | | |
|---|---|---|---|---|---|---|---|---|---|---|---|---|---|---|---|
| | M | F | D | M | F | D | M | F | D | M | F | D | M | F | D |
| *Replay (5%)* | | | | | | | | | | | | | | | |
| Trained(Exp 1) | 25.8 | 37.6 | 80.1 | – | – | – | – | – | – | – | – | – | – | – | – |
| Trained(Exp 1-2) | 17.2 | 24.9 | 67.0 | 69.6 | 71.1 | 89.0 | – | – | – | – | – | – | 8.5 | 12.8 | 13.1 |
| Trained(Exp 1-3) | 11.1 | 18.0 | 52.8 | 41.1 | 42.9 | 67.1 | 47.0 | 61.7 | 82.4 | – | – | – | 21.6 | 23.9 | 24.6 |
| Trained(Exp 1-4) | 10.1 | 16.7 | 50.0 | 36.6 | 38.0 | 59.0 | 16.0 | 23.5 | 56.2 | 50.7 | 61.1 | 81.6 | 26.5 | 30.8 | 28.8 |
| Mean | 16.1 | 24.3 | 62.5 | 49.1 | 50.7 | 71.7 | 31.5 | 42.6 | 69.3 | 50.7 | 61.1 | 81.6 | 18.9 | 22.5 | 22.2 |
| *Replay (10%)* | | | | | | | | | | | | | | | |
| Trained(Exp 1) | 25.4 | 37.5 | 80.3 | – | – | – | – | – | – | – | – | – | – | – | – |
| Trained(Exp 1-2) | 19.8 | 28.3 | 73.4 | 68.8 | 70.4 | 88.8 | – | – | – | – | – | – | 5.6 | 9.2 | 6.9 |
| Trained(Exp 1-3) | 15.3 | 23.3 | 65.2 | 49.6 | 51.7 | 75.7 | 47.3 | 61.1 | 82.1 | – | – | – | 14.7 | 16.5 | 14.1 |
| Trained(Exp 1-4) | 14.5 | 22.4 | 65.0 | 45.4 | 47.2 | 70.0 | 19.7 | 28.5 | 68.2 | 50.2 | 61.5 | 81.4 | 20.6 | 23.6 | 16.0 |
| Mean | 18.7 | 27.9 | 71.0 | 54.6 | 56.4 | 78.2 | 33.5 | 44.8 | 75.1 | 50.2 | 61.5 | 81.4 | 13.6 | 16.4 | 12.3 |
| *Replay (20%)* | | | | | | | | | | | | | | | |
| Trained(Exp 1) | 25.7 | 38.3 | 80.2 | – | – | – | – | – | – | – | – | – | – | – | – |
| Trained(Exp 1-2) | 22.7 | 31.5 | 75.9 | 68.9 | 70.1 | 89.2 | – | – | – | – | – | – | 2.9 | 6.8 | 4.3 |
| Trained(Exp 1-3) | 20.3 | 29.3 | 72.8 | 56.5 | 58.7 | 80.4 | 46.4 | 60.4 | 81.6 | – | – | – | 8.9 | 10.2 | 8.1 |
| Trained(Exp 1-4) | 19.9 | 28.3 | 70.8 | 54.8 | 56.4 | 78.5 | 26.0 | 36.7 | 72.6 | 47.5 | 58.1 | 79.8 | 13.4 | 15.8 | 9.7 |
| Mean | 22.2 | 31.8 | 74.9 | 60.1 | 61.7 | 82.7 | 36.2 | 48.5 | 77.1 | 47.5 | 58.1 | 79.8 | 8.4 | 10.9 | 7.4 |
| *Replay (10%) Qwen2.5-7B* | | | | | | | | | | | | | | | |
| Trained(Exp 1) | 26.2 | 38.2 | 80.2 | – | – | – | – | – | – | – | – | – | – | – | – |
| Trained(Exp 1-2) | 20.9 | 29.4 | 76.2 | 71.2 | 72.7 | 89.8 | – | – | – | – | – | – | 5.3 | 8.8 | 4.0 |
| Trained(Exp 1-3) | 14.8 | 23.8 | 69.2 | 51.4 | 53.3 | 77.0 | 49.7 | 63.8 | 83.6 | – | – | – | 15.6 | 16.9 | 11.9 |
| Trained(Exp 1-4) | 15.6 | 24.7 | 70.6 | 47.5 | 49.4 | 74.0 | 22.6 | 32.2 | 71.2 | 51.5 | 62.5 | 83.3 | 20.5 | 22.8 | 12.6 |
| Mean | 19.4 | 29.0 | 74.1 | 56.7 | 58.5 | 80.3 | 36.1 | 48.0 | 77.4 | 51.5 | 62.5 | 83.3 | 13.8 | 16.2 | 9.5 |

**Table 18** (continued)

| Setting | Experience 1 | | | Experience 2 | | | Experience 3 | | | Experience 4 | | | Forgetting | | |
|---|---|---|---|---|---|---|---|---|---|---|---|---|---|---|---|
| | M | F | D | M | F | D | M | F | D | M | F | D | M | F | D |
| *Replay (10%) Qwen3-4B* | | | | | | | | | | | | | | | |
| Trained(Exp 1) | 26.7 | 39.4 | 80.3 | – | – | – | – | – | – | – | – | – | – | – | – |
| Trained(Exp 1-2) | 19.3 | 27.6 | 74.9 | 69.3 | 70.7 | 88.8 | – | – | – | – | – | – | 7.4 | 11.8 | 5.4 |
| Trained(Exp 1-3) | 15.2 | 22.4 | 66.7 | 49.4 | 51.3 | 76.9 | 47.9 | 62.6 | 83.3 | – | – | – | 15.7 | 18.2 | 12.8 |
| Trained(Exp 1-4) | 15.2 | 23.5 | 69.1 | 45.7 | 47.9 | 73.5 | 20.8 | 29.7 | 69.7 | 51.0 | 61.5 | 82.1 | 20.7 | 23.9 | 13.3 |
| Mean | 19.1 | 28.2 | 72.8 | 54.8 | 56.6 | 79.7 | 34.4 | 46.2 | 76.5 | 51.0 | 61.5 | 82.1 | 14.6 | 18.0 | 10.5 |
| *Sequential Finetuning* | | | | | | | | | | | | | | | |
| Trained(Exp 1) | 25.4 | 38.0 | 79.9 | – | – | – | – | – | – | – | – | – | – | – | – |
| Trained(Exp 1-2) | 9.9 | 13.4 | 54.7 | 70.2 | 71.9 | 89.0 | – | – | – | – | – | – | 15.5 | 24.6 | 25.2 |
| Trained(Exp 1-3) | 2.3 | 4.7 | 40.2 | 9.4 | 9.8 | 47.1 | 47.1 | 61.1 | 82.4 | – | – | – | 42.0 | 47.7 | 40.8 |
| Trained(Exp 1-4) | 0.7 | 2.2 | 29.0 | 1.5 | 1.7 | 36.6 | 5.7 | 12.0 | 56.8 | 50.1 | 61.4 | 81.3 | 45.0 | 51.7 | 43.0 |
| Mean | 9.6 | 14.6 | 50.9 | 27.0 | 27.8 | 57.5 | 26.4 | 36.6 | 69.6 | 50.1 | 61.4 | 81.3 | 34.2 | 41.3 | 36.3 |
| *LwF* | | | | | | | | | | | | | | | |
| Trained(Exp 1) | 25.3 | 37.8 | 80.4 | – | – | – | – | – | – | – | – | – | – | – | – |
| Trained(Exp 1-2) | 21.2 | 29.5 | 67.9 | 61.9 | 63.3 | 84.0 | – | – | – | – | – | – | 4.1 | 8.3 | 12.6 |
| Trained(Exp 1-3) | 10.8 | 16.2 | 36.4 | 31.1 | 32.7 | 49.9 | 39.6 | 53.6 | 77.1 | – | – | – | 22.6 | 26.1 | 39.1 |
| Trained(Exp 1-4) | 4.6 | 8.5 | 17.7 | 11.0 | 11.7 | 18.7 | 14.3 | 20.9 | 38.0 | 41.3 | 52.2 | 75.6 | 32.3 | 37.8 | 55.8 |
| Mean | 15.5 | 23.0 | 50.6 | 34.7 | 35.9 | 50.9 | 26.9 | 37.3 | 57.5 | 41.3 | 52.2 | 75.6 | 19.7 | 24.1 | 35.8 |
| *EWC* | | | | | | | | | | | | | | | |
| Trained(Exp 1) | 26.1 | 38.1 | 80.6 | – | – | – | – | – | – | – | – | – | – | – | – |
| Trained(Exp 1-2) | 15.2 | 20.0 | 63.1 | 69.2 | 70.9 | 88.6 | – | – | – | – | – | – | 10.8 | 18.1 | 17.5 |
| Trained(Exp 1-3) | 4.6 | 7.8 | 43.8 | 31.7 | 33.1 | 62.9 | 45.4 | 59.8 | 80.6 | – | – | – | 29.5 | 34.1 | 31.3 |
| Trained(Exp 1-4) | 1.5 | 3.6 | 29.8 | 16.5 | 17.1 | 42.8 | 14.2 | 21.9 | 62.5 | 47.0 | 58.3 | 79.4 | 36.2 | 42.1 | 38.2 |
| Mean | 11.9 | 17.3 | 54.3 | 39.1 | 40.4 | 64.8 | 29.8 | 40.8 | 71.5 | 47.0 | 58.3 | 79.4 | 25.5 | 31.4 | 29.0 |
| *Merging (TIES)* | | | | | | | | | | | | | | | |
| Trained(Exp 1) | 25.6 | 37.4 | 80.6 | – | – | – | – | – | – | – | – | – | – | – | – |
| Merged(Exp 1-2) | 12.7 | 19.0 | 55.1 | 23.6 | 25.4 | 52.7 | – | – | – | – | – | – | 12.9 | 18.4 | 25.5 |
| Merged(Exp 1-3) | 5.7 | 9.9 | 36.9 | 10.3 | 11.6 | 34.7 | 7.6 | 13.4 | 31.8 | – | – | – | 16.6 | 20.7 | 30.9 |
| Merged(Exp 1-4) | 2.7 | 5.8 | 22.7 | 5.1 | 6.0 | 21.5 | 4.5 | 11.8 | 32.1 | 3.1 | 5.4 | 25.5 | 14.8 | 17.5 | 29.6 |
| Mean | 11.7 | 18.0 | 48.8 | 13.0 | 14.3 | 36.3 | 6.1 | 12.6 | 32.0 | 3.1 | 5.4 | 25.5 | 14.8 | 18.9 | 28.7 |
| *Gorilla with RAG* | | | | | | | | | | | | | | | |
| Trained(Exp 1) | 18.7 | 31.6 | 74.5 | – | – | – | – | – | – | – | – | – | – | – | – |
| Indexed(Exp 1-2) | 18.3 | 30.2 | 74.2 | 7.0 | 9.1 | 44.6 | – | – | – | – | – | – | 0.4 | 1.4 | 0.3 |
| Indexed(Exp 1-3) | 17.9 | 28.4 | 74.4 | 6.9 | 8.8 | 45.5 | 0.9 | 2.2 | 33.0 | – | – | – | 0.5 | 1.8 | 0.0 |
| Indexed(Exp 1-4) | 18.8 | 28.6 | 74.2 | 7.0 | 8.7 | 44.4 | 0.9 | 3.2 | 33.3 | 1.0 | 1.7 | 23.1 | 0.0 | 0.8 | 0.1 |
| Mean | 18.4 | 29.7 | 74.3 | 7.0 | 8.9 | 44.8 | 0.9 | 2.7 | 33.2 | 1.0 | 1.7 | 23.1 | 0.5 | 1.3 | 0.2 |
| *BGE-M3 (Retriever-only)* | | | | | | | | | | | | | | | |
| Indexed(Exp 1) | 22.6 | 27.7 | 60.0 | – | – | – | – | – | – | – | – | – | – | – | – |
| Indexed(Exp 1-2) | 20.8 | 24.0 | 58.2 | 12.4 | 14.2 | 47.2 | – | – | – | – | – | – | 1.8 | 3.7 | 1.8 |
| Indexed(Exp 1-3) | 18.8 | 21.6 | 54.8 | 11.0 | 12.6 | 44.1 | 11.3 | 13.3 | 34.8 | – | – | – | 2.6 | 3.8 | 4.2 |
| Indexed(Exp 1-4) | 17.4 | 20.1 | 52.6 | 10.3 | 11.7 | 43.8 | 9.6 | 10.8 | 34.3 | 13.0 | 16.7 | 39.8 | 3.0 | 4.2 | 3.8 |
| Mean | 19.9 | 23.4 | 56.4 | 11.2 | 12.8 | 45.0 | 10.4 | 12.0 | 34.6 | 13.0 | 16.7 | 39.8 | 2.5 | 3.9 | 3.3 |
| *Cumulative* | | | | | | | | | | | | | | | |
| Trained(Exp 1) | 25.9 | 38.2 | 80.7 | – | – | – | – | – | – | – | – | – | – | – | – |
| Trained(Exp 1-2) | 27.5 | 37.6 | 80.6 | 68.6 | 70.3 | 89.4 | – | – | – | – | – | – | 0.0 | 0.6 | 0.1 |
| Trained(Exp 1-3) | 26.8 | 36.5 | 79.3 | 67.9 | 69.3 | 88.1 | 46.8 | 61.3 | 82.5 | – | – | – | 0.0 | 1.3 | 1.3 |
| Trained(Exp 1-4) | 27.4 | 37.2 | 80.0 | 69.4 | 70.7 | 88.5 | 44.0 | 56.0 | 80.2 | 42.8 | 52.5 | 78.3 | 0.1 | 2.0 | 1.3 |
| Mean | 26.9 | 37.4 | 80.1 | 68.7 | 70.1 | 88.7 | 45.4 | 58.7 | 81.4 | 42.8 | 52.5 | 78.3 | 0.1 | 1.3 | 0.9 |
| *From scratch* | | | | | | | | | | | | | | | |

**Table 18 (continued)**

| Setting | Experience 1 | | | Experience 2 | | | Experience 3 | | | Experience 4 | | | Forgetting | | |
|---|---|---|---|---|---|---|---|---|---|---|---|---|---|---|---|
| | M | F | D | M | F | D | M | F | D | M | F | D | M | F | D |
| Trained(Exp 1) | 25.6 | 37.6 | 80.1 | – | – | – | – | – | – | – | – | – | – | – | – |
| Trained(Exp 1-2) | 27.1 | 38.4 | 79.6 | 67.3 | 69.1 | 88.4 | – | – | – | – | – | – | 0.0 | 0.0 | 0.5 |
| Trained(Exp 1-3) | 25.9 | 36.5 | 79.1 | 66.0 | 67.7 | 87.2 | 44.8 | 59.5 | 81.4 | – | – | – | 0.5 | 1.3 | 1.1 |
| Trained(Exp 1-4) | 25.8 | 37.4 | 79.6 | 66.6 | 68.7 | 88.0 | 41.0 | 52.3 | 70.9 | 40.5 | 50.9 | 78.7 | 1.4 | 2.6 | 3.8 |
| Mean | 26.1 | 37.5 | 79.6 | 66.7 | 68.5 | 87.9 | 42.9 | 55.9 | 76.1 | 40.5 | 50.9 | 78.7 | 1.0 | 2.0 | 1.8 |
| *Joint Training* | | | | | | | | | | | | | | | |
| Zero-shot(Exp 1-4) | 26.3 | 36.6 | 79.9 | 66.5 | 68.6 | 88.2 | 39.5 | 51.6 | 75.3 | 40.2 | 50.0 | 78.5 | – | – | – |
| RAT(Exp 1-4) | 19.7 | 29.5 | 75.7 | 63.7 | 65.4 | 86.0 | 39.8 | 51.7 | 78.6 | 36.8 | 46.8 | 76.5 | – | – | – |

*Table 19.* Extended results for Experiments 1–4 **without label snapping**. For each experience, we report accuracy for Model-ID (M), Model-family (F), and Domain (D) (%). The final block reports forgetting (%) for the same metrics, computed across sequential experiences. Unless otherwise specified, all methods use LLaMA2-7B as the backbone model.

| Setting | Experience 1 | | | Experience 2 | | | Experience 3 | | | Experience 4 | | | Forgetting | | |
|---|---|---|---|---|---|---|---|---|---|---|---|---|---|---|---|
| | M | F | D | M | F | D | M | F | D | M | F | D | M | F | D |
| *Replay (5%)* | | | | | | | | | | | | | | | |
| Trained(Exp 1) | 25.8 | 37.5 | 79.7 | – | – | – | – | – | – | – | – | – | – | – | – |
| Trained(Exp 1-2) | 16.7 | 24.0 | 64.3 | 69.6 | 71.1 | 88.7 | – | – | – | – | – | – | 9.0 | 13.5 | 15.4 |
| Trained(Exp 1-3) | 10.7 | 17.4 | 51.2 | 40.3 | 42.1 | 65.6 | 47.0 | 61.6 | 82.2 | – | – | – | 22.2 | 24.6 | 25.8 |
| Trained(Exp 1-4) | 9.9 | 16.2 | 47.9 | 35.7 | 36.8 | 56.8 | 15.1 | 22.3 | 54.0 | 50.6 | 61.0 | 81.4 | 27.2 | 31.6 | 30.7 |
| Mean | 15.8 | 23.8 | 60.8 | 48.5 | 50.0 | 70.4 | 31.0 | 42.0 | 68.1 | 50.6 | 61.0 | 81.4 | 19.5 | 23.2 | 24.0 |
| *Replay (10%)* | | | | | | | | | | | | | | | |
| Trained(Exp 1) | 25.4 | 37.2 | 79.9 | – | – | – | – | – | – | – | – | – | – | – | – |
| Trained(Exp 1-2) | 19.5 | 27.8 | 72.3 | 68.8 | 70.4 | 88.4 | – | – | – | – | – | – | 5.9 | 9.5 | 7.7 |
| Trained(Exp 1-3) | 15.0 | 22.9 | 64.1 | 48.6 | 50.6 | 74.3 | 47.3 | 61.1 | 82.0 | – | – | – | 15.3 | 17.0 | 15.0 |
| Trained(Exp 1-4) | 14.5 | 22.2 | 64.1 | 44.9 | 46.6 | 68.8 | 19.7 | 28.3 | 67.9 | 50.0 | 61.3 | 81.3 | 20.8 | 23.9 | 16.5 |
| Mean | 18.6 | 27.5 | 70.1 | 54.1 | 55.9 | 77.2 | 33.5 | 44.7 | 75.0 | 50.0 | 61.3 | 81.3 | 14.0 | 16.8 | 13.1 |
| *Replay (20%)* | | | | | | | | | | | | | | | |
| Trained(Exp 1) | 25.7 | 38.1 | 79.8 | – | – | – | – | – | – | – | – | – | – | – | – |
| Trained(Exp 1-2) | 22.5 | 31.1 | 75.0 | 68.7 | 70.0 | 88.7 | – | – | – | – | – | – | 3.2 | 7.0 | 4.8 |
| Trained(Exp 1-3) | 20.1 | 28.9 | 71.8 | 56.3 | 58.4 | 79.7 | 46.3 | 60.1 | 81.4 | – | – | – | 9.0 | 10.4 | 8.5 |
| Trained(Exp 1-4) | 19.7 | 27.7 | 69.7 | 54.4 | 55.9 | 77.3 | 26.0 | 36.6 | 72.5 | 47.3 | 58.0 | 79.6 | 13.5 | 16.0 | 10.1 |
| Mean | 22.0 | 31.4 | 74.1 | 59.8 | 61.4 | 81.9 | 36.1 | 48.4 | 76.9 | 47.3 | 58.0 | 79.6 | 8.6 | 11.1 | 7.8 |
| *Replay (10%) Qwen2.5-7B* | | | | | | | | | | | | | | | |
| Trained(Exp 1) | 26.2 | 38.0 | 79.8 | – | – | – | – | – | – | – | – | – | – | – | – |
| Trained(Exp 1-2) | 20.5 | 28.9 | 75.1 | 71.2 | 72.7 | 89.5 | – | – | – | – | – | – | 5.7 | 9.1 | 4.7 |
| Trained(Exp 1-3) | 14.6 | 23.3 | 68.0 | 50.8 | 52.6 | 75.8 | 49.7 | 63.8 | 83.5 | – | – | – | 16.0 | 17.4 | 12.7 |
| Trained(Exp 1-4) | 15.3 | 24.1 | 69.2 | 47.1 | 48.9 | 72.3 | 22.3 | 31.9 | 70.8 | 51.5 | 62.5 | 83.3 | 20.8 | 23.2 | 13.5 |
| Mean | 19.2 | 28.6 | 73.0 | 56.4 | 58.0 | 79.2 | 36.0 | 47.8 | 77.1 | 51.5 | 62.5 | 83.3 | 14.2 | 16.6 | 10.3 |
| *Replay (10%) Qwen3-4B* | | | | | | | | | | | | | | | |
| Trained(Exp 1) | 26.4 | 38.9 | 79.5 | – | – | – | – | – | – | – | – | – | – | – | – |
| Trained(Exp 1-2) | 18.7 | 26.7 | 73.4 | 69.3 | 70.7 | 88.5 | – | – | – | – | – | – | 7.7 | 12.2 | 6.2 |
| Trained(Exp 1-3) | 14.9 | 21.8 | 65.5 | 48.8 | 50.7 | 75.5 | 47.9 | 62.5 | 83.1 | – | – | – | 16.0 | 18.6 | 13.5 |
| Trained(Exp 1-4) | 15.0 | 23.2 | 67.7 | 45.2 | 47.3 | 72.1 | 20.7 | 29.7 | 69.6 | 51.0 | 61.5 | 82.1 | 20.9 | 24.0 | 13.9 |
| Mean | 18.8 | 27.7 | 71.5 | 54.5 | 56.2 | 78.7 | 34.3 | 46.1 | 76.4 | 51.0 | 61.5 | 82.1 | 14.9 | 18.3 | 11.2 |
| *LwF* | | | | | | | | | | | | | | | |
| Trained(Exp 1) | 25.2 | 37.5 | 80.0 | – | – | – | – | – | – | – | – | – | – | – | – |
| Trained(Exp 1-2) | 20.2 | 28.0 | 62.7 | 61.8 | 63.2 | 83.6 | – | – | – | – | – | – | 5.0 | 9.5 | 17.3 |

**Table 19** (continued)

| Setting | Experience 1 | | | Experience 2 | | | Experience 3 | | | Experience 4 | | | Forgetting | | |
|---|---|---|---|---|---|---|---|---|---|---|---|---|---|---|---|
| | M | F | D | M | F | D | M | F | D | M | F | D | M | F | D |
| Trained(Exp 1-3) | 10.1 | 14.7 | 32.8 | 29.4 | 30.9 | 47.0 | 39.6 | 53.5 | 76.8 | — | — | — | 23.8 | 27.6 | 41.9 |
| Trained(Exp 1-4) | 4.3 | 7.9 | 16.5 | 9.4 | 10.0 | 16.1 | 11.8 | 16.6 | 29.9 | 41.2 | 52.1 | 75.4 | 33.7 | 39.9 | 59.3 |
| Mean | 15.0 | 22.0 | 48.0 | 33.5 | 34.7 | 48.9 | 25.7 | 35.1 | 53.4 | 41.2 | 52.1 | 75.4 | 20.8 | 25.7 | 39.5 |
| *EWC* | | | | | | | | | | | | | | | |
| Trained(Exp 1) | 26.0 | 38.0 | 80.4 | — | — | — | — | — | — | — | — | — | — | — | — |
| Trained(Exp 1-2) | 14.0 | 18.3 | 59.1 | 69.2 | 70.8 | 88.3 | — | — | — | — | — | — | 12.1 | 19.7 | 21.3 |
| Trained(Exp 1-3) | 4.4 | 7.2 | 42.4 | 29.6 | 31.1 | 60.0 | 45.4 | 59.6 | 80.3 | — | — | — | 30.6 | 35.2 | 33.1 |
| Trained(Exp 1-4) | 1.4 | 3.4 | 29.1 | 14.6 | 15.1 | 39.5 | 12.6 | 20.0 | 60.5 | 46.9 | 58.2 | 79.2 | 37.3 | 43.3 | 39.9 |
| Mean | 11.5 | 16.7 | 52.8 | 37.8 | 39.0 | 62.6 | 29.0 | 39.8 | 70.4 | 46.9 | 58.2 | 79.2 | 26.7 | 32.7 | 31.4 |
| *Sequential Finetuning* | | | | | | | | | | | | | | | |
| Trained(Exp 1) | 25.4 | 37.9 | 79.6 | — | — | — | — | — | — | — | — | — | — | — | — |
| Trained(Exp 1-2) | 9.2 | 12.4 | 52.3 | 70.2 | 71.9 | 88.7 | — | — | — | — | — | — | 16.1 | 25.5 | 27.3 |
| Trained(Exp 1-3) | 2.2 | 4.6 | 39.9 | 8.4 | 8.8 | 45.8 | 47.1 | 60.9 | 82.3 | — | — | — | 42.5 | 48.1 | 41.3 |
| Trained(Exp 1-4) | 0.7 | 2.1 | 28.6 | 1.3 | 1.6 | 36.3 | 5.1 | 11.4 | 55.9 | 50.0 | 61.2 | 81.2 | 45.2 | 51.9 | 43.2 |
| Mean | 9.4 | 14.3 | 50.1 | 26.6 | 27.4 | 57.0 | 26.2 | 36.1 | 69.1 | 50.0 | 61.2 | 81.2 | 34.6 | 41.8 | 37.3 |
| *Merging (TIES)* | | | | | | | | | | | | | | | |
| Trained(Exp 1) | 25.5 | 37.2 | 80.3 | — | — | — | — | — | — | — | — | — | — | — | — |
| Merged(Exp 1-2) | 12.0 | 17.4 | 49.7 | 22.7 | 24.4 | 47.6 | — | — | — | — | — | — | 13.5 | 19.8 | 30.6 |
| Merged(Exp 1-3) | 5.0 | 8.1 | 28.8 | 9.5 | 10.3 | 28.2 | 5.9 | 10.6 | 23.6 | — | — | — | 16.8 | 21.6 | 35.5 |
| Merged(Exp 1-4) | 2.4 | 4.9 | 18.6 | 4.8 | 5.4 | 15.7 | 3.1 | 8.7 | 22.0 | 2.4 | 3.8 | 16.8 | 14.6 | 17.7 | 31.7 |
| Mean | 11.2 | 16.9 | 44.3 | 12.3 | 13.4 | 30.5 | 4.5 | 9.6 | 22.8 | 2.4 | 3.8 | 16.8 | 15.0 | 19.7 | 32.6 |
| *Gorilla with RAG* | | | | | | | | | | | | | | | |
| Trained(Exp 1) | 18.2 | 31.0 | 73.2 | — | — | — | — | — | — | — | — | — | — | — | — |
| Indexed(Exp 1-2) | 17.9 | 29.6 | 73.1 | 7.0 | 9.1 | 43.1 | — | — | — | — | — | — | 0.3 | 1.4 | 0.1 |
| Indexed(Exp 1-3) | 17.4 | 27.8 | 73.2 | 6.8 | 8.5 | 43.4 | 0.9 | 2.2 | 32.4 | — | — | — | 0.5 | 1.9 | 0.0 |
| Indexed(Exp 1-4) | 18.3 | 28.0 | 72.9 | 6.9 | 8.5 | 42.6 | 0.9 | 3.2 | 32.8 | 1.0 | 1.7 | 21.9 | 0.0 | 0.9 | 0.1 |
| Mean | 18.0 | 29.1 | 73.1 | 6.9 | 8.7 | 43.0 | 0.9 | 2.7 | 32.6 | 1.0 | 1.7 | 21.9 | 0.4 | 1.4 | 0.1 |
| *BGE-M3* | | | | | | | | | | | | | | | |
| Indexed(Exp 1) | 22.6 | 27.7 | 60.0 | — | — | — | — | — | — | — | — | — | — | — | — |
| Indexed(Exp 1-2) | 20.8 | 24.0 | 58.2 | 12.4 | 14.2 | 47.2 | — | — | — | — | — | — | 1.8 | 3.7 | 1.8 |
| Indexed(Exp 1-3) | 18.8 | 21.6 | 54.8 | 11.0 | 12.6 | 44.1 | 11.3 | 13.3 | 34.8 | — | — | — | 2.6 | 3.8 | 4.2 |
| Indexed(Exp 1-4) | 17.4 | 20.1 | 52.6 | 10.3 | 11.7 | 43.8 | 9.6 | 10.8 | 34.3 | 13.0 | 16.7 | 39.8 | 3.0 | 4.2 | 3.8 |
| Mean | 19.9 | 23.4 | 56.4 | 11.2 | 12.8 | 45.0 | 10.4 | 12.0 | 34.6 | 13.0 | 16.7 | 39.8 | 2.5 | 3.9 | 3.3 |
| *BGE-M3 with domain Oracle* | | | | | | | | | | | | | | | |
| Indexed(Exp 1) | 29.6 | 36.0 | 100 | — | — | — | — | — | — | — | — | — | — | — | — |
| Indexed(Exp 1-2) | 27.1 | 31.6 | 100 | 20.1 | 21.9 | 100 | — | — | — | — | — | — | 2.5 | 4.4 | — |
| Indexed(Exp 1-3) | 25.1 | 29.0 | 100 | 18.2 | 19.8 | 100 | 19.9 | 24.6 | 100 | — | — | — | 3.2 | 4.5 | |
| Indexed(Exp 1-4) | 24.3 | 28.5 | 100 | 17.2 | 18.8 | 100 | 15.3 | 18.6 | 100 | 19.8 | 26.2 | 100 | 4.3 | 5.5 | |
| Mean | 26.5 | 31.3 | 100 | 18.5 | 20.2 | 100 | 17.6 | 21.6 | 100 | 19.8 | 26.2 | 100 | 3.3 | 4.8 | |
| *Cumulative* | | | | | | | | | | | | | | | |
| Trained(Exp 1) | 25.8 | 38.1 | 80.4 | — | — | — | — | — | — | — | — | — | — | — | — |
| Trained(Exp 1-2) | 27.5 | 37.6 | 80.4 | 68.6 | 70.2 | 89.0 | — | — | — | — | — | — | 0.0 | 0.5 | 0.0 |
| Trained(Exp 1-3) | 26.8 | 36.5 | 79.1 | 67.9 | 69.3 | 87.8 | 46.8 | 61.2 | 82.3 | — | — | — | 0.0 | 1.2 | 1.3 |
| Trained(Exp 1-4) | 27.4 | 37.1 | 79.7 | 69.4 | 70.7 | 88.1 | 44.0 | 56.0 | 80.2 | 42.7 | 52.4 | 78.2 | 0.1 | 1.9 | 1.2 |
| Mean | 26.9 | 37.3 | 79.9 | 68.6 | 70.0 | 88.3 | 45.4 | 58.6 | 81.3 | 42.7 | 52.4 | 78.2 | 0.1 | 1.2 | 1.3 |
| *From scratch* | | | | | | | | | | | | | | | |
| Trained(Exp 1) | 25.6 | 37.5 | 79.7 | — | — | — | — | — | — | — | — | — | — | — | — |
| Trained(Exp 1-2) | 27.1 | 38.3 | 79.2 | 67.1 | 68.9 | 87.9 | — | — | — | — | — | — | 0.0 | 0.0 | 0.5 |
| Trained(Exp 1-3) | 25.9 | 36.5 | 78.7 | 65.9 | 67.6 | 86.7 | 44.8 | 59.4 | 81.3 | — | — | — | 0.4 | 1.2 | 1.1 |

**Table 19** (continued)

| Setting | Experience 1 | | | Experience 2 | | | Experience 3 | | | Experience 4 | | | Forgetting | | |
|---|---|---|---|---|---|---|---|---|---|---|---|---|---|---|---|
| | M | F | D | M | F | D | M | F | D | M | F | D | M | F | D |
| Trained(Exp 1-4) | 25.8 | 37.4 | 79.3 | 66.5 | 68.6 | 87.5 | 40.5 | 51.4 | 69.3 | 40.5 | 50.9 | 78.6 | 1.6 | 2.8 | 4.3 |
| Mean | 26.1 | 37.4 | 79.2 | 66.5 | 68.4 | 87.4 | 42.7 | 55.4 | 75.3 | 40.5 | 50.9 | 78.6 | 1.0 | 2.0 | 2.0 |
| *Joint Training* | | | | | | | | | | | | | | | |
| Zero-shot(Exp 1-4) | 26.2 | 36.5 | 79.5 | 66.5 | 68.5 | 87.8 | 39.4 | 51.3 | 74.7 | 40.1 | 50.0 | 78.4 | – | – | – |
| RAT(Exp 1-4) | 19.3 | 29.2 | 75.0 | 63.6 | 65.3 | 85.4 | 39.7 | 51.6 | 78.4 | 36.8 | 46.8 | 76.5 | – | – | – |
| *HuggingGPT Qwen3-32B* | | | | | | | | | | | | | | | |
| Zero-shot (no training) | – | – | 59.5 | – | – | 60.2 | – | – | 42.9 | – | – | 43.3 | – | – | – |

## D.5. CARvE Experiments Extended Results

Here, we see the extended results for the CARvE, covering random and domain coreset replay, the effects of adding the embedding and projection anchors and three different upper bounds. In cumulative training, the same adapter is trained on the union of all experiences seen so far. In from scratch training, a separate adapter is trained on the union of all experiences so far, resulting in 4 separate adapters (with the final one being the same as joint training).

*Table 20.* Extended Ablations Over CARVE (1). Here we compare random versus domain coreset replay across 4 experiences, along with the average forgetting computed across all four experiences. We report M: model-ID accuracy, F: model family accuracy, D: domain accuracy (%) and forgetting (%) across various setups.

| Setting | Experience 1 | | | Experience 2 | | | Experience 3 | | | Experience 4 | | | Forgetting | | |
|---|---|---|---|---|---|---|---|---|---|---|---|---|---|---|---|
| | M | F | D | M | F | D | M | F | D | M | F | D | M | F | D |
| *Random Replay (5%)* | | | | | | | | | | | | | | | |
| Trained(Exp 1) | 28.8 | 35.2 | 82.7 | – | – | – | – | – | – | – | – | – | – | – | – |
| Trained(Exp 1-2) | 26.4 | 32.4 | 79.2 | 67.9 | 69.2 | 88.3 | – | – | – | – | – | – | 2.4 | 2.8 | 3.5 |
| Trained(Exp 1-3) | 19.5 | 26.0 | 72.1 | 37.9 | 37.2 | 70.6 | 51.6 | 61.8 | 82.8 | – | – | – | 19.7 | 20.6 | 14.2 |
| Trained(Exp 1-4) | 21.7 | 27.1 | 75.8 | 38.2 | 38.9 | 69.2 | 14.9 | 17.1 | 66.9 | 42.3 | 50.9 | 78.5 | 24.5 | 27.7 | 14.0 |
| Mean | 24.1 | 30.2 | 77.5 | 48.0 | 48.4 | 76.0 | 33.2 | 39.5 | 74.8 | 42.3 | 50.9 | 78.5 | 15.5 | 17.0 | 10.6 |
| *Random Replay (10%)* | | | | | | | | | | | | | | | |
| Trained(Exp 1) | 28.8 | 35.2 | 82.7 | – | – | – | – | – | – | – | – | – | – | – | – |
| Trained(Exp 1-2) | 26.0 | 31.8 | 80.2 | 68.4 | 69.4 | 87.8 | – | – | – | – | – | – | 2.8 | 3.4 | 2.5 |
| Trained(Exp 1-3) | 19.7 | 24.8 | 74.2 | 47.6 | 48.8 | 77.3 | 51.0 | 61.2 | 83.6 | – | – | – | 15.0 | 15.5 | 9.5 |
| Trained(Exp 1-4) | 22.0 | 27.8 | 77.5 | 50.1 | 50.9 | 79.5 | 20.7 | 24.2 | 72.2 | 56.5 | 64.9 | 83.6 | 18.5 | 21.0 | 8.3 |
| Mean | 24.1 | 29.9 | 78.7 | 55.4 | 56.4 | 81.5 | 35.9 | 42.7 | 77.9 | 56.5 | 64.9 | 83.6 | 12.1 | 13.3 | 6.8 |
| *Random Replay (20%)* | | | | | | | | | | | | | | | |
| Trained(Exp 1) | 28.8 | 35.2 | 82.7 | – | – | – | – | – | – | – | – | – | – | – | – |
| Trained(Exp 1-2) | 27.9 | 33.0 | 80.5 | 68.6 | 69.1 | 89.3 | – | – | – | – | – | – | 0.9 | 2.2 | 2.2 |
| Trained(Exp 1-3) | 25.4 | 31.3 | 79.9 | 57.5 | 58.7 | 83.2 | 51.1 | 59.8 | 83.6 | – | – | – | 7.2 | 7.1 | 4.4 |
| Trained(Exp 1-4) | 25.0 | 31.3 | 80.0 | 59.8 | 61.6 | 86.3 | 28.0 | 32.6 | 77.0 | 53.3 | 63.2 | 83.5 | 11.9 | 12.9 | 4.1 |
| Mean | 26.8 | 32.7 | 80.8 | 62.0 | 63.1 | 86.3 | 39.5 | 46.2 | 80.3 | 53.3 | 63.2 | 83.5 | 6.7 | 7.4 | 3.6 |
| *Domain Replay (5%)* | | | | | | | | | | | | | | | |
| Trained(Exp 1) | 28.8 | 35.2 | 82.7 | – | – | – | – | – | – | – | – | – | – | – | – |
| Trained(Exp 1-2) | 26.1 | 31.7 | 79.5 | 68.2 | 70.1 | 88.5 | – | – | – | – | – | – | 2.7 | 3.5 | 3.2 |
| Trained(Exp 1-3) | 21.8 | 26.2 | 74.9 | 38.1 | 37.5 | 72.5 | 50.6 | 61.0 | 82.8 | – | – | – | 18.6 | 20.8 | 11.9 |
| Trained(Exp 1-4) | 23.2 | 27.2 | 74.5 | 37.9 | 39.0 | 70.6 | 16.6 | 18.7 | 69.7 | 56.3 | 63.8 | 84.1 | 23.3 | 27.1 | 13.1 |
| Mean | 25.0 | 30.1 | 77.9 | 48.1 | 48.9 | 77.2 | 33.6 | 39.9 | 76.2 | 56.3 | 63.8 | 84.1 | 14.9 | 17.1 | 9.4 |
| *Domain Replay (10%)* | | | | | | | | | | | | | | | |
| Trained(Exp 1) | 28.8 | 35.2 | 82.7 | – | – | – | – | – | – | – | – | – | – | – | – |
| Trained(Exp 1-2) | 26.4 | 31.3 | 81.9 | 67.6 | 68.7 | 88.3 | – | – | – | – | – | – | 2.4 | 3.9 | 0.8 |
| Trained(Exp 1-3) | 24.6 | 29.4 | 76.0 | 43.8 | 45.0 | 76.3 | 52.0 | 60.8 | 83.8 | – | – | – | 14.0 | 14.8 | 9.4 |
| Trained(Exp 1-4) | 25.4 | 31.3 | 79.2 | 47.7 | 49.6 | 77.8 | 22.0 | 25.0 | 73.5 | 55.5 | 64.0 | 83.5 | 17.8 | 19.6 | 8.1 |
| Mean | 26.3 | 31.8 | 80.0 | 53.0 | 54.4 | 80.8 | 37.0 | 42.9 | 78.7 | 55.5 | 64.0 | 83.5 | 11.4 | 12.8 | 6.1 |
| *Domain Replay (20%)* | | | | | | | | | | | | | | | |
| Trained(Exp 1) | 28.8 | 35.2 | 82.7 | – | – | – | – | – | – | – | – | – | – | – | – |
| Trained(Exp 1-2) | 27.6 | 33.3 | 81.5 | 67.9 | 68.6 | 88.8 | – | – | – | – | – | – | 1.2 | 1.9 | 1.2 |
| Trained(Exp 1-3) | 27.2 | 32.4 | 79.4 | 57.4 | 58.4 | 83.3 | 51.4 | 62.2 | 83.4 | – | – | – | 6.1 | 6.5 | 4.4 |
| Trained(Exp 1-4) | 28.0 | 33.8 | 81.7 | 57.9 | 59.7 | 84.7 | 28.8 | 34.2 | 76.9 | 56.3 | 63.3 | 83.8 | 11.1 | 12.8 | 3.9 |
| Mean | 27.9 | 33.7 | 81.3 | 61.1 | 62.2 | 85.6 | 40.1 | 48.2 | 80.2 | 56.3 | 63.3 | 83.8 | 6.1 | 7.1 | 3.2 |
| *Cumulative Training* | | | | | | | | | | | | | | | |
| Trained(Exp 1) | 28.8 | 35.2 | 82.7 | – | – | – | – | – | – | – | – | – | – | – | – |
| Trained(Exp 1-2) | 30.3 | 36.4 | 83.5 | 68.0 | 68.9 | 87.5 | – | – | – | – | – | – | 1.5 | 1.2 | 0.8 |
| Trained(Exp 1-3) | 28.6 | 34.9 | 83.3 | 68.2 | 69.1 | 88.8 | 49.1 | 59.6 | 81.1 | – | – | – | 0.0 | 0.1 | 0.9 |
| Trained(Exp 1-4) | 29.2 | 34.8 | 82.9 | 69.1 | 70.1 | 88.8 | 44.1 | 52.6 | 80.0 | 49.0 | 56.5 | 81.8 | 1.2 | 2.1 | 0.1 |
| Mean | 29.2 | 35.3 | 83.1 | 68.4 | 69.4 | 88.4 | 46.6 | 56.1 | 80.5 | 49.0 | 56.5 | 81.8 | 0.9 | 1.1 | 0.6 |
| *Joint Training* | | | | | | | | | | | | | | | |
| Joint Training | 29.6 | 36.3 | 81.9 | 72.1 | 72.9 | 91.0 | 47.6 | 56.3 | 81.5 | 49.1 | 57.0 | 83.1 | – | – | – |

*Table 21.* Extended Ablations Over CARVE (2). Cumulative training denotes continuing to train the same adapter on all the union of all experiences seen so far. From Scratch Training denotes the training of a separate adapter for each experience using the union of all experiences seen so far, acting as an upper bound if one were to deploy a model in practice which requires retraining as new data becomes available. We report M: model-ID accuracy, F: model family accuracy, D: domain accuracy (%) and forgetting (%) across various setups.

| Setting | Experience 1 | | | Experience 2 | | | Experience 3 | | | Experience 4 | | | Forgetting | | |
|---|---|---|---|---|---|---|---|---|---|---|---|---|---|---|---|
| | M | F | D | M | F | D | M | F | D | M | F | D | M | F | D |
| *From Scratch Training* | | | | | | | | | | | | | | | |
| Trained(Exp 1) | 28.8 | 35.2 | 82.7 | – | – | – | – | – | – | – | – | – | – | – | – |
| Trained(Exp 1-2) | 29.5 | 35.3 | 83.0 | 70.5 | 71.4 | 90.2 | – | – | – | – | – | – | 0.7 | 0.1 | 0.3 |
| Trained(Exp 1-3) | 28.6 | 34.7 | 82.2 | 71.6 | 72.9 | 90.6 | 51.3 | 63.0 | 83.4 | – | – | – | 0.4 | 0.5 | 0.1 |
| Trained(Exp 1-4) | 29.6 | 36.3 | 81.9 | 72.1 | 72.9 | 91.0 | 47.6 | 56.3 | 81.5 | 49.1 | 57.0 | 83.1 | – | – | – |
| Mean | 29.1 | 35.4 | 82.5 | 71.4 | 72.4 | 90.6 | 49.5 | 59.7 | 82.5 | 49.1 | 57.0 | 83.1 | 0.6 | 0.3 | 0.2 |
| *No Embedding Anchor, 10% Domain Replay* | | | | | | | | | | | | | | | |
| Trained(Exp 1) | 28.8 | 35.2 | 82.7 | – | – | – | – | – | – | – | – | – | – | – | – |
| Trained(Exp 1-2) | 25.8 | 31.9 | 80.2 | 67.5 | 69.4 | 88.2 | – | – | – | – | – | – | 3.0 | 3.3 | 2.5 |
| Trained(Exp 1-3) | 23.6 | 27.9 | 75.0 | 43.6 | 45.5 | 76.1 | 51.7 | 62.1 | 83.1 | – | – | – | 14.5 | 15.6 | 9.9 |
| Trained(Exp 1-4) | 24.2 | 30.4 | 78.9 | 47.8 | 50.3 | 79.1 | 20.2 | 22.4 | 72.8 | 55.4 | 63.1 | 84.2 | 18.6 | 21.2 | 7.7 |
| Mean | 25.6 | 31.4 | 79.2 | 53.0 | 55.1 | 81.1 | 36.0 | 42.2 | 77.9 | 55.4 | 63.1 | 84.2 | 12.0 | 13.4 | 6.7 |
| *No Projection Anchor, 10% Domain Replay* | | | | | | | | | | | | | | | |
| Trained(Exp 1) | 28.8 | 35.2 | 82.7 | – | – | – | – | – | – | – | – | – | – | – | – |
| Trained(Exp 1-2) | 23.4 | 28.8 | 78.2 | 68.1 | 68.9 | 88.0 | – | – | – | – | – | – | 5.4 | 6.4 | 4.5 |
| Trained(Exp 1-3) | 19.7 | 24.2 | 71.2 | 46.3 | 46.3 | 75.3 | 44.5 | 56.6 | 81.6 | – | – | – | 15.4 | 16.8 | 12.1 |
| Trained(Exp 1-4) | 17.1 | 22.0 | 69.3 | 44.6 | 46.1 | 74.2 | 20.5 | 24.5 | 72.3 | 48.2 | 57.4 | 80.8 | 19.7 | 22.7 | 12.2 |
| Mean | 22.2 | 27.6 | 75.4 | 53.0 | 53.8 | 79.2 | 32.5 | 40.5 | 76.9 | 48.2 | 57.4 | 80.8 | 13.5 | 15.3 | 9.6 |
| *EWC, 10% Domain Replay* | | | | | | | | | | | | | | | |
| Trained(Exp 1) | 28.3 | 32.8 | 83.2 | – | – | – | – | – | – | – | – | – | – | – | – |
| Trained(Exp 1-2) | 26.1 | 31.7 | 81.8 | 65.4 | 66.5 | 86.1 | – | – | – | – | – | – | 2.2 | 1.1 | 1.4 |
| Trained(Exp 1-3) | 27.3 | 32.9 | 82.1 | 54.5 | 55.9 | 81.9 | 49.0 | 60.2 | 82.9 | – | – | – | 6.0 | 5.2 | 2.6 |
| Trained(Exp 1-4) | 28.3 | 34.0 | 82.4 | 52.3 | 54.0 | 79.5 | 18.8 | 22.9 | 72.8 | 54.0 | 62.9 | 84.6 | 14.4 | 16.2 | 5.8 |
| Mean | 27.5 | 32.9 | 82.4 | 57.4 | 58.8 | 82.5 | 33.9 | 41.5 | 77.8 | 54.0 | 62.9 | 84.6 | 7.5 | 7.5 | 3.3 |
| *Qwen3-4B, 10% Domain Replay* | | | | | | | | | | | | | | | |
| Trained(Exp 1) | 26.9 | 32.3 | 83.4 | – | – | – | – | – | – | – | – | – | – | – | – |
| Trained(Exp 1-2) | 24.8 | 29.4 | 81.2 | 62.5 | 63.9 | 88.7 | – | – | – | – | – | – | 2.1 | 2.9 | 2.2 |
| Trained(Exp 1-3) | 21.7 | 27.2 | 77.6 | 37.1 | 39.1 | 78.2 | 47.6 | 57.4 | 82.6 | – | – | – | 15.3 | 14.9 | 8.2 |
| Trained(Exp 1-4) | 22.4 | 29.2 | 79.5 | 40.7 | 42.7 | 78.2 | 16.1 | 19.8 | 69.7 | 51.1 | 60.1 | 83.6 | 19.3 | 20.6 | 9.1 |
| Mean | 24.0 | 29.5 | 80.4 | 46.8 | 48.6 | 81.7 | 31.9 | 38.6 | 76.2 | 51.1 | 60.1 | 83.6 | 12.2 | 12.8 | 6.5 |
| *Qwen2.5-7B, 10% Domain Replay* | | | | | | | | | | | | | | | |
| Trained(Exp 1) | 28.0 | 34.1 | 84.8 | – | – | – | – | – | – | – | – | – | – | – | – |
| Trained(Exp 1-2) | 25.1 | 31.0 | 83.2 | 67.0 | 67.9 | 90.0 | – | – | – | – | – | – | 2.9 | 3.1 | 1.6 |
| Trained(Exp 1-3) | 23.2 | 28.7 | 77.4 | 45.2 | 46.5 | 78.2 | 51.6 | 62.2 | 82.9 | – | – | – | 13.3 | 13.4 | 9.6 |
| Trained(Exp 1-4) | 24.0 | 29.1 | 79.0 | 48.7 | 50.1 | 79.6 | 20.6 | 24.6 | 73.9 | 54.8 | 63.9 | 83.9 | 17.8 | 20.1 | 8.4 |
| Mean | 25.1 | 30.7 | 81.1 | 53.6 | 54.8 | 82.6 | 36.1 | 43.4 | 78.4 | 54.8 | 63.9 | 83.9 | 11.3 | 12.2 | 6.5 |
| *Label Noise 10%, 10% Domain Replay* | | | | | | | | | | | | | | | |
| Trained(Exp 1) | 27.6 | 32.5 | 71.0 | – | – | – | – | – | – | – | – | – | – | – | – |
| Trained(Exp 1-2) | 25.1 | 30.8 | 71.5 | 63.0 | 64.0 | 84.1 | – | – | – | – | – | – | 2.5 | 1.7 | 0.5 |
| Trained(Exp 1-3) | 21.8 | 27.3 | 65.1 | 38.0 | 40.1 | 67.9 | 48.4 | 57.2 | 74.9 | – | – | – | 15.4 | 14.5 | 11.0 |
| Trained(Exp 1-4) | 25.4 | 31.4 | 69.3 | 39.9 | 42.2 | 69.7 | 18.4 | 21.0 | 61.0 | 52.0 | 60.9 | 77.8 | 18.4 | 19.7 | 10.0 |
| Mean | 25.0 | 30.5 | 69.2 | 47.0 | 48.8 | 73.9 | 33.4 | 39.1 | 68.0 | 52.0 | 60.9 | 77.8 | 12.1 | 12.0 | 7.2 |
| *Label Noise 20%, 10% Domain Replay* | | | | | | | | | | | | | | | |
| Trained(Exp 1) | 26.5 | 30.9 | 57.7 | – | – | – | – | – | – | – | – | – | – | – | – |
| Trained(Exp 1-2) | 23.9 | 28.4 | 57.3 | 56.8 | 59.2 | 72.5 | – | – | – | – | – | – | 2.6 | 2.5 | 0.4 |
| Trained(Exp 1-3) | 20.8 | 24.6 | 51.9 | 31.8 | 32.6 | 54.2 | 43.7 | 54.9 | 66.2 | – | – | – | 15.3 | 16.4 | 12.1 |
| Trained(Exp 1-4) | 21.3 | 25.7 | 54.0 | 35.4 | 37.6 | 56.4 | 15.9 | 18.7 | 52.4 | 48.2 | 58.3 | 69.1 | 18.1 | 21.0 | 11.2 |
| Mean | 23.1 | 27.4 | 55.2 | 41.3 | 43.1 | 61.0 | 29.8 | 36.8 | 59.3 | 48.2 | 58.3 | 69.1 | 12.0 | 13.3 | 7.9 |

*Table 22.* Extended Ablations Over CARvE (3). We compare random versus domain coreset replay across four experiences, together with cumulative and joint training upper bounds, reporting average forgetting across all four experiences. We report M3: top-3 model-ID accuracy and D3: top-3 domain accuracy (%) and forgetting (%) across all setups.

| Setting | Experience 1 | | Experience 2 | | Experience 3 | | Experience 4 | | Forgetting | |
|---|---|---|---|---|---|---|---|---|---|---|
| | M3 | D3 | M3 | D3 | M3 | D3 | M3 | D3 | M3 | D3 |
| *Random Replay (5%)* | | | | | | | | | | |
| Trained(Exp 1) | 55.1 | 96.2 | — | — | — | — | — | — | — | — |
| Trained(Exp 1-2) | 53.3 | 95.8 | 93.5 | 98.6 | — | — | — | — | 1.8 | 0.4 |
| Trained(Exp 1-3) | 46.2 | 90.9 | 76.7 | 92.3 | 84.9 | 97.4 | — | — | 12.8 | 5.8 |
| Trained(Exp 1-4) | 45.4 | 91.3 | 75.9 | 90.0 | 44.5 | 87.4 | 79.5 | 94.2 | 22.6 | 7.8 |
| Mean | 50.0 | 93.5 | 82.0 | 93.6 | 64.7 | 92.4 | 79.5 | 94.2 | 12.4 | 4.7 |
| *Random Replay (10%)* | | | | | | | | | | |
| Trained(Exp 1) | 55.1 | 96.2 | — | — | — | — | — | — | — | — |
| Trained(Exp 1-2) | 52.1 | 95.8 | 94.0 | 99.0 | — | — | — | — | 3.0 | 0.4 |
| Trained(Exp 1-3) | 47.9 | 92.8 | 78.7 | 94.0 | 83.7 | 96.7 | — | — | 11.2 | 4.2 |
| Trained(Exp 1-4) | 48.4 | 92.8 | 80.1 | 92.9 | 54.1 | 90.7 | 89.5 | 97.4 | 16.7 | 5.2 |
| Mean | 50.9 | 94.4 | 84.3 | 95.3 | 68.9 | 93.7 | 89.5 | 97.4 | 10.3 | 3.3 |
| *Random Replay (20%)* | | | | | | | | | | |
| Trained(Exp 1) | 55.1 | 96.2 | — | — | — | — | — | — | — | — |
| Trained(Exp 1-2) | 55.6 | 95.5 | 95.1 | 98.9 | — | — | — | — | 0.5 | 0.7 |
| Trained(Exp 1-3) | 50.5 | 93.7 | 86.5 | 95.8 | 84.2 | 97.0 | — | — | 6.6 | 2.8 |
| Trained(Exp 1-4) | 51.2 | 94.9 | 86.1 | 95.5 | 62.8 | 94.8 | 88.2 | 97.8 | 11.4 | 2.3 |
| Mean | 53.1 | 95.1 | 89.2 | 96.7 | 73.5 | 95.9 | 88.2 | 97.8 | 6.2 | 1.9 |
| *Domain Replay (5%)* | | | | | | | | | | |
| Trained(Exp 1) | 55.1 | 96.2 | — | — | — | — | — | — | — | — |
| Trained(Exp 1-2) | 51.5 | 95.7 | 94.4 | 98.4 | — | — | — | — | 3.6 | 0.5 |
| Trained(Exp 1-3) | 45.0 | 90.7 | 71.6 | 92.3 | 84.3 | 96.7 | — | — | 16.5 | 5.8 |
| Trained(Exp 1-4) | 48.1 | 91.8 | 73.7 | 88.4 | 43.9 | 88.5 | 88.1 | 96.9 | 22.7 | 7.5 |
| Mean | 49.9 | 93.6 | 79.9 | 93.0 | 64.1 | 92.6 | 88.1 | 96.9 | 14.3 | 4.6 |
| *Domain Replay (10%)* | | | | | | | | | | |
| Trained(Exp 1) | 55.1 | 96.2 | — | — | — | — | — | — | — | — |
| Trained(Exp 1-2) | 54.1 | 95.8 | 93.5 | 98.6 | — | — | — | — | 1.0 | 0.4 |
| Trained(Exp 1-3) | 49.3 | 92.6 | 74.7 | 93.9 | 83.6 | 97.3 | — | — | 12.3 | 4.1 |
| Trained(Exp 1-4) | 51.1 | 94.5 | 77.3 | 91.8 | 53.4 | 91.3 | 87.9 | 97.5 | 16.8 | 4.8 |
| Mean | 52.4 | 94.8 | 81.8 | 94.8 | 68.5 | 94.3 | 87.9 | 97.5 | 10.0 | 3.1 |
| *Domain Replay (20%)* | | | | | | | | | | |
| Trained(Exp 1) | 55.1 | 96.2 | — | — | — | — | — | — | — | — |
| Trained(Exp 1-2) | 54.2 | 96.1 | 94.1 | 98.8 | — | — | — | — | 0.9 | 0.1 |
| Trained(Exp 1-3) | 54.4 | 94.9 | 85.1 | 95.9 | 83.4 | 97.4 | — | — | 4.9 | 2.1 |
| Trained(Exp 1-4) | 53.8 | 94.8 | 84.0 | 96.5 | 62.9 | 93.8 | 88.3 | 97.8 | 10.6 | 2.4 |
| Mean | 54.4 | 95.5 | 87.7 | 97.1 | 73.2 | 95.6 | 88.3 | 97.8 | 5.5 | 1.5 |
| *Cumulative Training* | | | | | | | | | | |
| Trained(Exp 1) | 55.1 | 96.2 | — | — | — | — | — | — | — | — |
| Trained(Exp 1-2) | 55.4 | 96.2 | 93.5 | 98.8 | — | — | — | — | 0.3 | 0.0 |
| Trained(Exp 1-3) | 54.2 | 95.3 | 92.1 | 98.6 | 84.0 | 96.6 | — | — | 1.2 | 0.6 |
| Trained(Exp 1-4) | 54.8 | 95.8 | 92.4 | 98.1 | 78.4 | 96.5 | 83.4 | 96.9 | 2.3 | 0.4 |
| Mean | 54.9 | 95.9 | 92.7 | 98.5 | 81.2 | 96.5 | 83.4 | 96.9 | 1.3 | 0.3 |
| *Joint Training* | | | | | | | | | | |
| Joint Training | 56.2 | 96.1 | 94.0 | 99.8 | 83.2 | 96.7 | 83.1 | 97.1 | — | — |

*Table 23.* Extended Ablations Over CARvE (4). We report top-3 results corresponding to the ablations in Table 21, covering from-scratch upper bound, anchoring and regularisation ablations, backbone variants, and label noise conditions. From-scratch training denotes training a separate adapter on the union of all experiences seen so far, acting as a practical upper bound for deployment with full retraining. We report M3: top-3 model-ID accuracy and D3: top-3 domain accuracy (%) and forgetting (%) across all setups.

| Setting | Experience 1 | | Experience 2 | | Experience 3 | | Experience 4 | | Forgetting | |
|---|---|---|---|---|---|---|---|---|---|---|
| | M3 | D3 | M3 | D3 | M3 | D3 | M3 | D3 | M3 | D3 |
| *From Scratch Training* | | | | | | | | | | |
| Trained(Exp 1) | 55.1 | 96.2 | — | — | — | — | — | — | — | — |
| Trained(Exp 1-2) | 53.1 | 96.3 | 93.6 | 99.8 | — | — | — | — | 2.0 | 0.1 |
| Trained(Exp 1-3) | 53.4 | 96.7 | 94.1 | 99.6 | 84.3 | 96.4 | — | — | 0.6 | 0.1 |
| Trained(Exp 1-4) | 56.2 | 96.1 | 94.0 | 99.8 | 83.2 | 96.7 | 83.1 | 97.1 | — | — |
| Mean | 54.5 | 96.3 | 93.9 | 99.7 | 83.8 | 96.6 | 83.1 | 97.1 | 0.9 | 0.1 |
| *No Embedding Anchor, 10% Domain Replay* | | | | | | | | | | |
| Trained(Exp 1) | 55.1 | 96.2 | — | — | — | — | — | — | — | — |
| Trained(Exp 1-2) | 51.8 | 95.6 | 94.4 | 98.3 | — | — | — | — | 3.3 | 0.6 |
| Trained(Exp 1-3) | 49.5 | 92.6 | 75.9 | 93.6 | 84.7 | 96.4 | — | — | 12.1 | 4.2 |
| Trained(Exp 1-4) | 48.7 | 94.1 | 75.4 | 95.1 | 48.5 | 91.1 | 87.2 | 97.6 | 20.5 | 3.5 |
| Mean | 51.3 | 94.6 | 81.9 | 95.7 | 66.6 | 93.8 | 87.2 | 97.6 | 12.0 | 2.8 |
| *No Projection Anchor, 10% Domain Replay* | | | | | | | | | | |
| Trained(Exp 1) | 55.1 | 96.2 | — | — | — | — | — | — | — | — |
| Trained(Exp 1-2) | 51.8 | 94.5 | 94.5 | 99.0 | — | — | — | — | 3.3 | 1.7 |
| Trained(Exp 1-3) | 45.1 | 89.3 | 80.7 | 92.9 | 79.7 | 95.2 | — | — | 11.9 | 6.5 |
| Trained(Exp 1-4) | 44.6 | 85.7 | 78.4 | 89.2 | 49.5 | 90.9 | 82.2 | 95.2 | 18.9 | 8.2 |
| Mean | 49.1 | 91.4 | 84.5 | 93.7 | 64.6 | 93.1 | 82.2 | 95.2 | 11.4 | 5.5 |
| *EWC, 10% Domain Replay* | | | | | | | | | | |
| Trained(Exp 1) | 54.0 | 96.2 | — | — | — | — | — | — | — | — |
| Trained(Exp 1-2) | 53.5 | 95.8 | 93.6 | 98.5 | — | — | — | — | 0.5 | 0.4 |
| Trained(Exp 1-3) | 51.7 | 95.5 | 85.6 | 96.8 | 82.7 | 96.3 | — | — | 5.1 | 1.2 |
| Trained(Exp 1-4) | 51.5 | 95.3 | 82.4 | 95.4 | 50.0 | 91.1 | 86.8 | 97.7 | 15.5 | 3.1 |
| Mean | 52.7 | 95.7 | 87.2 | 96.9 | 66.3 | 93.7 | 86.8 | 97.7 | 7.0 | 1.6 |
| *Qwen3-4B, 10% Domain Replay* | | | | | | | | | | |
| Trained(Exp 1) | 51.2 | 97.8 | — | — | — | — | — | — | — | — |
| Trained(Exp 1-2) | 50.7 | 97.5 | 91.3 | 99.1 | — | — | — | — | 0.5 | 0.3 |
| Trained(Exp 1-3) | 45.5 | 94.9 | 67.2 | 94.0 | 81.3 | 97.0 | — | — | 14.9 | 4.0 |
| Trained(Exp 1-4) | 47.5 | 96.5 | 70.5 | 95.1 | 45.6 | 91.1 | 85.5 | 96.6 | 20.1 | 3.7 |
| Mean | 48.7 | 96.7 | 76.3 | 96.1 | 63.5 | 94.0 | 85.5 | 96.6 | 11.8 | 2.7 |
| *Qwen2.5-7B, 10% Domain Replay* | | | | | | | | | | |
| Trained(Exp 1) | 53.2 | 97.8 | — | — | — | — | — | — | — | — |
| Trained(Exp 1-2) | 50.5 | 97.0 | 93.8 | 99.1 | | | | | 2.7 | 0.8 |
| Trained(Exp 1-3) | 49.0 | 94.1 | 75.3 | 95.3 | 85.0 | 97.3 | — | — | 11.4 | 3.8 |
| Trained(Exp 1-4) | 49.2 | 95.0 | 76.1 | 93.8 | 49.4 | 92.9 | 88.8 | 97.4 | 19.1 | 4.2 |
| Mean | 50.5 | 96.0 | 81.7 | 96.1 | 67.2 | 95.1 | 88.8 | 97.4 | 11.1 | 2.9 |
| *Label Noise 10%, 10% Domain Replay* | | | | | | | | | | |
| Trained(Exp 1) | 58.8 | 91.9 | — | — | — | — | — | — | — | — |
| Trained(Exp 1-2) | 55.0 | 91.3 | 91.9 | 97.8 | — | — | — | — | 3.8 | 0.6 |
| Trained(Exp 1-3) | 53.9 | 87.2 | 78.3 | 90.0 | 84.8 | 94.5 | — | — | 9.3 | 6.2 |
| Trained(Exp 1-4) | 54.2 | 88.7 | 76.6 | 87.7 | 52.2 | 86.5 | 90.4 | 96.1 | 17.5 | 7.1 |
| Mean | 55.5 | 89.8 | 82.3 | 91.8 | 68.5 | 90.5 | 90.4 | 96.1 | 10.2 | 4.6 |
| *Label Noise 20%, 10% Domain Replay* | | | | | | | | | | |
| Trained(Exp 1) | 65.4 | 82.7 | — | — | — | — | — | — | — | — |
| Trained(Exp 1-2) | 63.6 | 82.1 | 94.0 | 92.9 | — | — | — | — | 1.8 | 0.6 |
| Trained(Exp 1-3) | 58.7 | 77.3 | 77.0 | 80.0 | 85.7 | 88.0 | — | — | 11.9 | 9.2 |
| Trained(Exp 1-4) | 57.3 | 77.7 | 78.4 | 80.1 | 48.1 | 75.2 | 87.8 | 91.9 | 20.4 | 10.2 |
| Mean | 61.2 | 80.0 | 83.1 | 84.3 | 66.9 | 81.6 | 87.8 | 91.9 | 11.4 | 6.7 |

*Table 24.* Comparison of model-embedding initialisation strategies, all using domain coreset replay at 10%. M-Acc, D-Acc, and D-Fgt (%) are averaged over four experiences. Values are mean $\pm$ SD over three seeds. $\tau$ and $k$ denote the temperature and number of neighbours used to construct soft contrastive targets from card embeddings.

| Initialisation | $\tau$ | $k$ | M-Acc | D-Acc | D-Fgt |
|---|---|---|---|---|---|
| Random (ours) | — | — | **43.1**$\pm$0.2 | **80.7**$\pm$0.0 | **5.9**$\pm$0.3 |
| Global card | 0.05 | 20 | 38.0$\pm$0.4 | 77.4$\pm$0.6 | 13.4$\pm$0.7 |
| Global card | 0.10 | 50 | 38.1$\pm$0.5 | 77.5$\pm$0.5 | 13.3$\pm$0.7 |
| Within-domain card | 0.05 | 20 | 37.1$\pm$0.6 | 78.8$\pm$0.3 | 10.2$\pm$0.5 |
| Within-domain card | 0.10 | 50 | 37.6$\pm$0.5 | 78.5$\pm$0.2 | 10.6$\pm$0.7 |

