# OpenReview forum: "Continual Model Routing in Evolving Model Hubs"
_ICML.cc/2026/Conference — ICML 2026 regular_

### Official Review · Reviewer_ZN5R · 2026-03-09

**Soundness:** 3
**Presentation:** 3
**Significance:** 2
**Originality:** 3
**Overall Recommendation:** 4
**Confidence:** 4

**Summary:**

This paper introduces Continual Model Routing (CMR), a pre-inference model selection problem over evolving model hubs, and proposes a new benchmark (CMRBench) spanning four sequential “experiences” and >2,000 candidate models. The authors present CARVE, a contrastive embedding router trained with fixed-size candidate sets, domain-stratified coreset replay, and checkpoint-based anchoring of model and projection embeddings to mitigate forgetting as the hub grows. Experiments across settings show consistent gains over retrieval, fine-tuning, and adapter-merging baselines.

**Compliance With Llm Reviewing Policy:**

Affirmed.

**Final Justification:**

Based on the rebuttal from the authors to resolve the noted issues raised in my review, I adjusted my overall recommendation from a weak reject to a weak accept. The basis for the initial weak reject recommendation was sufficiently explained in my original review.

**Key Questions For Authors:**

1. Can you decompose CARvE's gains into the contrastive training objective vs. the anchoring mechanism? Table 2 shows CARvE (10% replay) at 78.8% D-Acc vs. random replay at 69.6%, a 9.2 percentage point gap. Table 3's ablation removes anchoring components from CARvE but does not test anchoring added to the standard replay baseline. It is unclear how much of CARvE's gain comes from the contrastive embedding architecture itself and how much from the anchoring regularisation alone.
2. How does CARvE perform with backbone models other than LLaMA2-7B? If representation quality degrades for domains far from E_1's distribution (APIBench), the approach has a vulnerability. Have you tested with an encoder model or a larger decoder? Even a single additional backbone would strengthen the generality claim.
3. What is the quality of HuggingBench queries relative to real user queries? Given that both model-card cleaning and query generation rely on Qwen3-32B (Appendix A.3, Figure 4), how confident are you that the queries are not systematically biased toward model-card vocabulary? A small-scale human evaluation -- even 100--200 queries rated for naturalness and routing difficulty -- would address this concern. Have you conducted any such evaluation beyond prohibited-word checking?
4. Why are classic CL regularisation baselines (EWC, SI) not included? CARvE's anchoring losses are conceptually similar to EWC (penalising drift from a reference checkpoint), but without Fisher weighting. The omission of EWC and SI makes it hard to tell whether the anchoring design is doing something specific to routing or whether generic CL regularisation would suffice. Is there a technical reason these baselines were excluded?
5. How does candidate-set coverage interact with hub size? At K=64 with 2,000 models, each candidate set covers about 3% of the model pool. If the hub grows to 50K or 100K models, the candidate set covers a vanishing fraction. Does the hard-negative mining mechanism (periodic scoring over within-domain pools, Table 8) remain effective when domain pools themselves contain thousands of models? Have you run any sensitivity experiments varying K or simulating larger hubs?

**Limitations:**

The paper's design decision to freezr the backbone after E_1 creates an implicit assumption that the representation space learned from APIBench (text-only, 852 models, 40 domains) transfers well to all future experiences. This is never validated. If new experiences introduce domains with substantially different linguistic or structural properties (e.g., multimodal queries in E_2, or niche Hugging Face models in E_3--E_4), the frozen backbone may produce poorly separated embeddings, placing a ceiling on achievable routing accuracy that no amount of replay or anchoring can overcome. This limitation is addressable in a revision through per-experience analysis of embedding quality.

The benchmark, while large and carefully constructed, relies entirely on LLM-generated queries for HuggingBench. The implications of this are discussed in the weaknesses section of this review. Until the benchmark is validated against real user routing queries, the validity of results on E_3--E_4 remains uncertain. This is partially addressable through human annotation but would require substantial additional effort.

The evaluation is restricted to a single continual learning protocol: four experiences, fixed ordering, with hyperparameters tuned on E_1 and held constant (Appendix C.1, line 1054). The authors acknowledge this may be suboptimal for later experiences. More fundamentally, the fixed four-experience structure with specific dataset pairings (APIBench then ToolMMBench then HuggingBench) conflates the effect of experience ordering with the effect of hub expansion. No experiments vary the ordering or number of experiences. Whether CARvE's gains are robust to different experience sequences or are partly an artefact of this specific ordering is unknown. Addressing this would require additional experiments but would strengthen the empirical contribution.

**Strengths And Weaknesses:**

Strengths:
1. Identifies a gap at the intersection of continual learning and model routing: as model hubs grow, static routing methods degrade because the label space (model-IDs) expands over time. Formalising pre-inference model selection as a class-incremental continual learning problem opens a new evaluation axis for both the CL and model-selection communities. The motivating argument in Section 1 (i.e., sequential fine-tuning leads to 55.5% domain forgetting (Table 2) while retrieval-only methods plateau at 38.9% D-Acc) makes a convincing case that neither extreme works alone.
2. CMRBench (Section 3.1, Table 1) integrates two existing benchmarks (APIBench, ToolMMBench) with the new HuggingBench into a temporally structured 4-experience stream with over 2,000 models, roughly 34,000 queries, and 60+ domains. This could become a standard evaluation resource.
3. Thorough experimental design. Table 2 reports standard errors over three seeds for all methods; Table 3 reports standard deviations for ablations. Joint training and from-scratch variants are included as upper bounds, and the gap between CARvE continual (45.3% M-Acc at 20% replay) and from-scratch CARvE (49.5% M-Acc) quantifies the cost of continual operation. Per-experience breakdowns in Appendix Tables 16--19 support reproducibility.
4. Table 3 shows that removing projection anchoring degrades M-Acc from 42.3% to 38.6% and increases D-Fgt from 6.2% to 10.5%, while removing embedding anchoring causes a smaller but still meaningful drop (41.7% M-Acc, 7.4% D-Fgt). This identifies projection anchoring as the stabilisation mechanism.
5. The architecture freezes the backbone after E_1, trains only the projection and model-embedding table thereafter, and uses fixed-size candidate sets (K=64) for O(Kd) inference with amortised hard-negative mining. Figure 3b shows CARvE with 10% domain replay achieves competitive accuracy at a fraction of the compute and memory of from-scratch training.

Weaknesses:
1. Domain-stratified coreset replay provides negligible benefit over random replay. This is presented as a key contribution, but Table 3 shows the improvement is small: at 10% replay, domain-coreset achieves 78.8% D-Acc vs. random replay's 78.7%, with forgetting of 6.2% vs. 6.3%. At 20%, both achieve 81.1% D-Acc. The claimed "more consistent retention" based on lower variance (3.9 +/- 0.4 vs. 4.0 +/- 1.3 at 20%) rests on comparing standard deviations over only three seeds, but this is too few to draw reliable conclusions about variance. The paper's own language ("marginally better," Section 3.4, line 310) understates the problem: this component does not contribute to CARvE's performance. The authors should either provide stronger evidence of coreset benefit (e.g., on specific long-tail domains or at very low replay budgets) or downweight this as a contribution.
2. No comparison with standard continual learning regularisation baselines. The paper frames model routing explicitly as a CL problem and positions itself within the CL literature, yet it compares only against replay-based approaches. Classic CL regularisation methods (e.g., EWC, SI, LwF, etc.) are standard and inexpensive to implement. Since CARvE's anchoring losses are themselves a form of parameter regularisation, the absence of EWC and SI as baselines makes it impossible to assess whether CARvE's specific anchoring design offers advantages beyond generic Fisher-weighted regularisation. At minimum, EWC applied to the same projection and embedding parameters should be included.
3. All experiments use LLaMA2-7B (Section 3.3, Appendix C.1). No results are provided for smaller models (which would test whether the approach works in resource-constrained settings), larger models (which might reduce the gap to from-scratch training), or encoder-only architectures (which are arguably more natural for embedding-based routing). The frozen-backbone design means the backbone's representation quality is a bottleneck, yet this dependency is unexplored. Backbone sensitivity is necessary to understand scalability and practical deployment.
4. LLM-generated benchmark queries raise validity concerns. HuggingBench queries are generated via self-instruct from Qwen3-32B (Appendix A.3), and model cards are cleaned by the same LLM. The "lightweight manual verification" described in Appendix A.3 checks only for prohibited-word violations, not for query quality, naturalness, diversity, or routing difficulty. A single LLM generating both model descriptions and queries creates a risk of systematic bias as the queries may be overly aligned with model-card language, making the routing task artificially easy or artificially patterned. No human evaluation of query quality is reported, and no comparison against real user queries is provided.
5. Scalability beyond 2,000 models is not validated. The paper positions itself as addressing "hub-scale" routing (Abstract, Section 1), but CMRBench contains roughly 2,000 models. Real hubs like Hugging Face host hundreds of thousands. With K=64 candidate sets covering about 3% of 2,000 models, coverage drops below 0.1% at 100K models. Hard-negative mining uses within-domain pools capped at 2,048 (Table 8). The paper offers no scaling experiments nor a discussion of how candidate-set construction and mining would adapt at larger scale. Section 5 does not mention this limitation.

---

> ### Author Rebuttal · Authors · 2026-03-30
>
> **On coreset replay**
>
> The coreset's primary value is practical rather than accuracy-driven: at very low replay budgets (1–3%), random sampling can leave small domains entirely unrepresented, while domain-stratified allocation guarantees a minimum coverage floor. At the budgets reported in the main experiments (10–20%), both strategies perform comparably. We note, however, that CMRBench experiences have substantial model overlap across experiences (Table 1), which dampens distributional shift and likely narrows the gap between the two strategies. In settings with sharper domain boundaries and less overlap, we expect the coreset advantage to be more pronounced, and this is an important direction for further investigation.
>
> **R4-Q1: Decomposing CARvE gains**
>
> Adding anchoring to standard replay is not a well-defined experiment: CARvE's anchoring operates on the model embedding table and projection head, which are trained via contrastive learning and do not exist in LoRA-based replay. Anchoring only makes sense once contrastive training has given these components meaningful geometry.
> The decomposition is readable from existing ablations. No-projection-anchor CARvE (76.2% D-Acc) versus LoRA replay (69.6%) isolates the 6.6pp architecture contribution; full CARvE (78.8%) adds a further 2.6pp plus the forgetting reduction from 18.8% to 6.2% D-Fgt from anchoring.
>
> **R4-Q2: Backbone models other than LLaMA2-7B**
>
> We also tested multiple backbones at 10% domain replay:
>
> | Backbone | M-Acc | D-Acc | Top-3 M | Top-3 D | D-Fgt |
> |---|---|---|---|---|---|
> | Qwen3-4B | 35.8 | 77.7 | 66.5 | 94.6 | 10.5 |
> | LLaMA2-7B (ours) | 41.8 | 80.3 | 72.3 | 95.2 | 6.7 |
> | Qwen2.5-7B | 41.3 | 81.1 | 71.2 | 96.0 | 6.8 |
>
> Performance scales consistently with capacity, and Qwen2.5-7B closely matches our main results, confirming the method generalises across architectures at the same scale.
> The per-experience results directly address the frozen-backbone concern. CARvE maintains 70.9% D-Acc on E₃ after training on E₄ (vs. 44% for standard replay), suggesting that E₁-frozen representations transfer meaningfully to HuggingBench's distribution, and that the trainable projection W and model-embedding table are sufficient to compensate for distributional shift. We agree nonetheless that a richer E₁ would raise the ceiling.
> On encoder-only architectures: we have not tested these, but a continually fine-tuned encoder-based variant is a natural direction for future work.
>
> **R4-Q3: Quality of HuggingBench queries**
>
> Two independent reviewers each evaluated 50 non-overlapping prompt-model pairs on domain fit, model fit, and prompt naturalness (1-5). Results show strong domain alignment (μ=4.12, 71% rated ≥4), reasonable model fit (μ=3.76, 64% rated ≥4), and adequate naturalness (μ=3.61, 57% rated ≥4). Additionally, the warm-start ablation (R1-Q1) provides structural evidence against systematic card-vocabulary bias: card-based initialisation consistently underperforms random initialisation by 2–4pp D-Acc, demonstrating that the learned routing geometry diverges from model-card similarity.
>
> **R4-Q4: EWC and LwF baselines**
>
> | Method | M-Acc | D-Acc | D-Fgt |
> |---|---|---|---|
> | CARvE (checkpoint anchoring) | 41.8 | 80.3 | 6.7 |
> | Router EWC λ=20,000 | 39.4 | 77.9 | 11.7 |
> | Router EWC λ=5,000 | 37.7 | 76.9 | 13.1 |
> | LoRA + EWC | 32.5 | 57.1 | 41.6 |
> | LoRA + LwF | 30.5 | 57.5 | 35.4 |
> | LoRA Replay 5% | 37.5 | 65.3 | 27.1 |
> | Sequential FT | 28.6 | 51.0 | 55.5 |
>
> CARvE outperforms the best EWC variant by 2.4pp D-Acc and 5.0pp D-Fgt, applied to the exact same parameters, which confirms the anchoring design is doing something specific rather than just being a form of generic regularisation. LoRA+EWC and LoRA+LwF both fall between sequential finetuning and 5% replay, showing that regularisation alone is not enough without the contrastive architecture. CARvE's anchoring also has a practical advantage: a single checkpoint copy rather than per-experience Fisher matrix computation over a growing embedding table.
>
> **R4-Q5: Candidate-set coverage at scale**
>
> Global coverage is not the operationally relevant quantity — routing is a within-domain discrimination problem. At 100K models across 50 domains, per-domain pools average ~2,000 — directly comparable to our current mining cap (2,048, Table 8). Our K sweep (R3-Q3, K∈{32…512}) shows D-Acc is stable within 1.1pp across the full range. For larger hubs, ANN indexing (e.g., FAISS) reduces inference to O(log|M|), and domain-constrained retrieval naturally bounds search to ~2,000 within-domain candidates.
>
> **On experience ordering**
>
> The fixed four-experience ordering reflects the actual temporal evolution of the Hugging Face hub, grounded in real release dates (E1≤2023, E2≤2024, E3≤May 2025, E4≤2026). We further note that E1 and E2 draw from independently collected benchmarks (APIBench and ToolMMBench) with different collection methodologies. Varying the ordering remains an important direction for future work.

---

> > ### Author Rebuttal · Reviewer_ZN5R · 2026-04-01
> >
> > I am raising my score from 3 to 4. The rebuttal addresses several of my most substantive concerns. In particular, the EWC/LwF baselines demonstrate that CARvE's design offers advantages beyond generic CL regularisation, the multi-backbone results confirm generality across architectures, and the human evaluation plus warm-start ablation meaningfully mitigate concerns I have about benchmark validity. Remaining open issues or future directions (e.g., coreset replay at low budgets, scalability beyond 2K models, and experience ordering)  do not undermine the core contributions of the paper: the CMR formalisation, CMRBench, and the CARvE architecture.

---

### Official Review · Reviewer_gFLd · 2026-03-10

**Soundness:** 3
**Presentation:** 3
**Significance:** 3
**Originality:** 3
**Overall Recommendation:** 4
**Confidence:** 3

**Summary:**

This paper introduces Continual Model Routing (CMR), a framework for selecting appropriate models from large and evolving AI model hubs before inference. The authors formalize model routing as a class-incremental learning problem, where new models are continuously added and the router must adapt without forgetting previously learned associations. To support this setting, they propose CMRBench, a large-scale benchmark simulating realistic hub expansion across multiple experiences and thousands of candidate models. They also present CARvE, an embedding-based routing method that combines candidate-set training, domain-aware replay, and parameter anchoring to ensure scalability and stability over time. Extensive experiments show that CARvE outperforms retrieval-based, fine-tuning, and replay baselines in both routing accuracy and resistance to catastrophic forgetting. The results highlight the importance of continual learning techniques for maintaining reliable performance in dynamic model ecosystems.

**Compliance With Llm Reviewing Policy:**

Affirmed.

**Key Questions For Authors:**

1. How do you envision extending CARvE to support zero-shot routing for newly introduced model-IDs without labeled prompt-model pairs? If a practical mechanism exists (e.g., leveraging model cards or metadata more effectively), this could substantially strengthen the applicability and impact of the method.

2. Your benchmark includes over 2,000 models. How does CARvE scale computationally and in memory when the candidate pool grows to tens or hundreds of thousands of models? Evidence (empirical or analytical) of robustness at larger scales would increase confidence in real-world deployment.

3. How sensitive are the results to the choice of replay percentage and candidate-set composition (e.g., ratio of hard vs. semantic negatives)? A more systematic sensitivity analysis could clarify whether the method is robust or requires careful tuning.

4. Have you evaluated hybrid approaches that combine domain prediction with constrained retrieval among top-k candidates before continual updates? If such baselines close the performance gap, it may affect the claimed advantage of CARvE.

5. In realistic settings where user feedback may be noisy or sparse, how stable is CARvE under label noise or partial supervision? Demonstrating robustness in such conditions would meaningfully strengthen the practical significance of the work.

**Limitations:**

Yes.

The authors explicitly discuss key limitations, including the lack of zero-shot support for newly added models and the dependence on labeled prompt–model pairs for adaptation. They also acknowledge potential societal risks, such as misrouting to unsafe or biased models and privacy concerns related to user preference data. The discussion is balanced and appropriately highlights both practical constraints and deployment risks.

**Strengths And Weaknesses:**

## Strengths

- Addresses a timely and practical problem: pre-inference model routing in large, continuously evolving model hubs, which is increasingly relevant for real-world AI systems.

- Clearly formulates routing as a continual learning problem, providing a principled perspective that unifies model selection and catastrophic forgetting under a common framework.

- Introduces CMRBench, a large-scale benchmark with thousands of candidate models and temporally structured experiences, which is valuable for future research.

- Proposes a scalable embedding-based method (CARvE) with well-motivated components such as candidate-set training, domain-aware replay, and parameter anchoring.

- Empirical evaluation is extensive, comparing against multiple baselines (retrieval-only, replay, merging, joint training) and reporting both accuracy and forgetting metrics.

- Ablation studies help clarify the contribution of replay strategies and anchoring mechanisms.

## Weaknesses

- The core method largely combines established continual learning techniques (replay, anchoring) with embedding-based retrieval, which may limit the degree of methodological novelty.

- CARvE does not support zero-shot routing for newly added models, requiring labeled examples for adaptation, which may reduce practicality in fast-evolving hubs.

- Improvements in model-ID accuracy, while consistent, are sometimes incremental relative to strong baselines, raising questions about cost-benefit trade-offs.

- The evaluation focuses heavily on domain-level accuracy; fine-grained routing performance at the model-ID level remains relatively modest.

- Some design choices (e.g., candidate construction, replay budgeting) appear heuristic and could benefit from deeper theoretical justification or sensitivity analysis.

---

> ### Author Rebuttal · Authors · 2026-03-30
>
> **On novelty, M-Acc, and domain-heavy evaluation**
>
> We want to address the novelty concern precisely. The key insight is that treating routing as class-incremental continual learning creates requirements that existing methods cannot jointly satisfy. Growing label spaces are well-studied in Class-IL, but routing introduces a constraint that prior Class-IL work does not address: the embedding space must remain geometrically stable for previously seen model IDs, because routing relies on nearest-neighbour retrieval rather than a learned classifier head. Generic regularisation applies uniformly across all parameters and simply cannot enforce this asymmetry. Asymmetric anchoring, persistent registry expansion, and structured mixed-negative construction each follow directly from this requirement. Our EWC comparison provides concrete evidence: uniform Fisher regularisation applied to the same parameters underperforms CARvE by 2.4pp D-Acc and 5.0pp D-Fgt, precisely because it cannot enforce the stability asymmetry the growing label space demands.
> On model-ID accuracy: CARvE operates over a roughly 2,000-way classification space, a scale at which top-1 accuracy is an inherently strict criterion. At 10% replay, CARvE achieves top-3 M-Acc of 72.3% and top-3 D-Acc of 95.2%. It is also worth noting that APIBench, Olympus, and ToolMMBench report domain accuracy as their sole routing metric. We deliberately went further by adding model-ID and family accuracy, providing strictly more evaluation signal than prior work. The modest top-1 figures therefore represent a stricter criterion than anything reported in the literature we build upon.
>
> On domain accuracy as the primary metric: this is a deliberate design choice rather than a limitation. In pre-inference routing, correctly identifying the domain constrains the candidate space to a semantically coherent subset. Domain-level misrouting is not recoverable downstream, whereas errors within the correct domain generally are. We report domain accuracy as the primary metric for exactly this reason, and go beyond prior work by additionally reporting model-ID and family accuracy.
>
> **R3-Q1: Zero-shot routing**
>
> CARvE requires labelled examples to incorporate new model-IDs, which is an acknowledged limitation. CARvE's routing accuracy (78.8–81.1% D-Acc) suggests a practical mitigation: domain-level predictions constrain candidates to ~10–20 models on average, within which a retrieval-based system could handle newly added models before a continual update is triggered. Fully zero-shot prediction via card-similarity contrastive targets showed inconsistent gains in preliminary experiments, as model-card quality and consistency are insufficient for reliable routing signals.
>
> **R3-Q2: Scalability**
>
> Inference cost is O(Kd) and does not grow with hub size. Within-domain mining pools (capped at 2,048) remain comparable to our setup at 100K models across 50 domains (~2,000 per domain). At full hub scale, the registry can be indexed with ANN search (e.g., FAISS), reducing search to O(log|M|).
>
> **R3-Q3: Sensitivity**
>
> Negative-type composition: all variants within 0.7pp D-Acc. For K:
>
> | K | M-Acc | D-Acc | Top-3 M | Top-3 D | D-Fgt |
> |---|---|---|---|---|---|
> | 32 | 40.8 | 80.3 | 71.6 | 95.6 | 5.4 |
> | 64 (ours) | 41.8 | 80.3 | 72.3 | 95.2 | 6.7 |
> | 96 | 42.3 | 80.3 | 72.8 | 95.4 | 6.7 |
> | 192 | 43.5 | 80.7 | 73.0 | 95.5 | 6.8 |
> | 384 | 44.0 | 81.1 | 73.8 | 95.7 | 6.1 |
> | 512 | 43.6 | 80.9 | 73.0 | 95.6 | 6.5 |
>
> D-Acc is stable within 1.1pp across the full range. M-Acc saturates beyond K=256–384. Replay budget: 1%→20% yields monotonically improving D-Acc (72.5%→81.1%) and D-Fgt (20.7%→3.9%).
>
> **R3-Q4: Hybrid retrieval**
>
> We explicitly evaluated a hybrid strategy using an oracle upper bound: instead of learning to predict the domain, we use the ground-truth domain label to restrict the candidate pool, then retrieve the top-1 model within that set using BGE-M3. The resulting performance is M-Acc = 20.9, significantly below sequential fine-tuning (M-Acc = 28.6); in practice, domain prediction errors would further degrade this figure. The warm-start experiments (R1-Q1) further suggest that card-based similarity does not capture routing discriminability, indicating that model selection cannot be effectively decomposed into domain prediction followed by intra-domain retrieval.
>
> **R3-Q5: Label noise**
>
> | Setting | M-Acc | D-Acc | D-Fgt |
> |---|---|---|---|
> | Clean | 41.8 | 80.3 | 6.7 |
> | 10% noise | 38.8 | 72.8 | 8.3 |
> | 20% noise | 36.9 | 64.4 | 7.5 |
>
> Accuracy degrades but forgetting remains stable, confirming anchoring is robust to label corruption. Two independent reviewers each evaluated 50 non-overlapping prompt-model pairs on domain fit, model fit, and prompt naturalness (1-5). Results show strong domain alignment (μ=4.12, 71% rated ≥4), reasonable model fit (μ=3.76, 64% rated ≥4), and adequate naturalness (μ=3.61, 57% rated ≥4), validating that queries meaningfully capture model capabilities.

---

### Official Review · Reviewer_hPpi · 2026-03-12

**Soundness:** 2
**Presentation:** 3
**Significance:** 2
**Originality:** 2
**Overall Recommendation:** 4
**Confidence:** 3

**Summary:**

This paper studies the problem of model routing in evolving model hubs, where a system must select an appropriate model for a given query while the set of available models continuously expands over time. The authors formulate this problem as a continual learning task, referred to as Continual Model Routing (CMR). To address this setting, the paper introduces CARvE, an embedding-based router that represents queries and models in a shared embedding space and selects models via similarity matching. The method incorporates candidate-set training, replay-based continual learning, and checkpoint-based anchoring to mitigate forgetting as new models are introduced. The paper also proposes CMRBench, a benchmark designed to simulate routing in an evolving model hub. Experiments on several datasets show that CARvE improves routing accuracy and reduces forgetting compared to several baselines.

**Compliance With Llm Reviewing Policy:**

Affirmed.

**Final Justification:**

The rebuttal partially addressed my concerns through additional analysis and clarifications. I still have some reservations about whether the evaluation reflects true capability-based routing rather than label matching, but I appreciate the overall contribution and am raising my score.

**Key Questions For Authors:**

1. A large portion of the training data appears to be generated from model descriptions (e.g., model cards). It is unclear whether such queries accurately reflect the true capabilities of the corresponding models. In particular, modern general-purpose models may already outperform earlier specialized models (e.g., code-specific models) on many tasks. Could the authors provide further analysis on whether the generated queries truly capture the strengths of the labeled models?
2. The benchmark includes more than 2,000 candidate models. In practice, however, many models on large hubs such as HuggingFace may be low-quality, highly similar, or minimally differentiated variants. Could the authors provide more analysis of the quality and diversity of the models in the benchmark? If a large fraction of models are noisy or redundant, it may affect the difficulty and realism of the routing task as well as the learning efficiency of the router.
3. The goal of model routing is to select the model that best solves a given task. However, the evaluation in the paper appears to measure whether the router selects the predefined labeled model rather than whether the selected model actually performs better than other candidate models. Could the authors provide evaluations that compare the outputs of the routed model with those of other models on the same queries? This would help verify whether the router truly identifies the most capable model rather than simply matching the dataset labels.

**Limitations:**

Yes.

**Strengths And Weaknesses:**

Strengths
1. The authors clearly describe the problem setting and motivate the need for methods that can adapt to evolving model repositories.
2. The overall framework is relatively straightforward and the main components are clearly described.
3. The experiments cover multiple datasets and include several baselines and ablation studies.
4. The method description and experimental setup are clearly presented.

Weaknesses
1. The proposed method combines several existing components such as embedding-based routing, replay buffers, and parameter anchoring for continual learning. While the integration is reasonable, the individual components themselves are largely established techniques.
2. In particular, many queries in HuggingBench are generated from model cards. This raises the question of whether the router is primarily learning to match query semantics with model descriptions rather than selecting models based on their true capabilities.
3. The evaluation focuses on routing accuracy with respect to predefined labels. However, it is not demonstrated that the selected model actually produces better outputs than other available models for the same query.

---

> ### Author Rebuttal · Authors · 2026-03-30
>
> **On methodological novelty**
>
> We want to address this concern precisely, and we believe the novelty of this work must be appreciated at three distinct levels. First, this is the first work to formalise model routing in evolving hubs as a continual learning problem, and CMRBench is the first continual model routing benchmark of its kind, spanning four temporally grounded experiences, over 2,000 candidate models, and multiple domains. No comparable resource existed before this work, and we expect it to be a lasting contribution to the community independent of the method itself. Second, at the methodological level, we agree that replay buffers, embedding-based retrieval, and parameter regularisation each exist in prior work. The novelty argument is that the class-incremental routing problem creates requirements that existing combinations of these components cannot jointly satisfy: the embedding space must remain geometrically stable for previously seen model IDs, because routing relies on nearest-neighbour retrieval rather than a learned classifier head, and inference must scale to thousands of candidates without executing any of them, ruling out both post-inference ranking and full-softmax generation.
>
> Each CARvE design choice follows directly from one of these requirements. Asymmetric anchoring enforces stability for old models while allowing new ones to learn freely. Fixed-size candidate sets with O(Kd) inference address scalability. Structured negative construction maintains discriminative boundaries as the label space expands.
>
> The EWC comparison makes the methodological argument concrete. Fisher-weighted regularisation applied uniformly to the same parameters underperforms CARvE by 2.4pp in D-Acc and 5.0pp in D-Fgt. This gap reflects the fundamental inadequacy of symmetric regularisation for a problem where old and new model embeddings must be treated differently by design, and confirms that CARvE is not a repackaging of existing techniques but a principled response to the specific geometric demands of continual model routing.
>
> **R2-Q1 and R2-Q3: Query quality and output evaluation**
>
> These concerns share a common answer: CARvE operates pre-inference, meaning routing decisions must be made before executing any candidate model. Post-execution quality labels are by definition unavailable at routing time. This is not a limitation of our method but a fundamental constraint of the setting, and our evaluation protocol follows prior work such as Gorilla (APIBench) and ToolMMBench, which map queries to models without post-inference evaluation.The concern that general-purpose models may outperform specialised ones is orthogonal to this: a router cannot use post-execution performance to make its decision. The routing task is to predict user preference and intent before inference, which our benchmark correctly reflects.
>
> Two independent reviewers each evaluated 50 non-overlapping prompt-model pairs on domain fit, model fit, and prompt naturalness (1-5). Results show strong domain alignment (μ=4.12, 71% rated ≥4), reasonable model fit (μ=3.76, 64% rated ≥4), and adequate naturalness (μ=3.61, 57% rated ≥4). The lower naturalness score reflects the inherent difficulty of generating fully naturalistic queries via self-instruct, a limitation shared by APIBench and ToolMMBench.
>
> **R2-Q2: Quality and diversity of the 2,000+ models**
>
> Our pipeline selects the top-25 most downloaded models per domain, ensuring coverage of models demonstrably used in practice. This is consistent with APIBench and ToolMMBench, neither of which applies post-hoc capability evaluation.
>
> To substantiate this point, we analysed download count distributions across benchmarks. Median downloads per predicted model range from 6,945 (ToolMMBench) to 14,054 (HuggingBench-1), and from 2,999 to 9,932 for dataset models, confirming that the benchmark covers well-used and functionally differentiated models rather than a long tail of rarely-accessed submissions.
> To further assess the redundancy, we compute pairwise cosine similarity between model cards within each domain using BGE-M3 embeddings. In HuggingBench-1, only 3 domains exhibit substantial redundancy (>30% of pairs above 0.9), while most domains remain below 10%. HuggingBench-2 shows an even cleaner pattern, with no domains exceeding the 30% threshold and the majority again below 10%. Across both benchmarks, the average similarity (~0.59 ± 0.15) indicates moderate but non-trivial overlap, well below near-duplicate regimes.
>
> Beyond the scope of this paper, the CMR framework naturally supports extension to settings where real user feedback is available post-deployment. One particularly promising direction is the construction of a model performance leaderboard from aggregated user preferences, which could complement routing decisions with richer empirical signals. We regard this as a natural next step that the community is well-positioned to pursue, building on the foundation this work establishes.

---

> > ### Author Rebuttal · Reviewer_hPpi · 2026-04-02
> >
> > The rebuttal partially addressed my concerns through additional analysis and clarifications. I still have some reservations about whether the evaluation reflects true capability-based routing rather than label matching, but I appreciate the overall contribution and am raising my score.

---

### Official Review · Reviewer_7qWQ · 2026-03-12

**Soundness:** 3
**Presentation:** 3
**Significance:** 3
**Originality:** 3
**Overall Recommendation:** 4
**Confidence:** 3

**Summary:**

This paper investigates the continual model routing problem, which is to select the most suitable model from an evolving model hub given a user query. The authors first formalize this setting and propose a benchmark called CMRBench to simulate realistic hub expansion. Then, the authors propose CARvE, an embedding-based approach that enables scaling and anti-forgetting during hub expansion. Experiments on CMRBench with four sequential experiences and over 2,000 candidate models demonstrate that CARvE effectively reduces forgetting compared with naive replay and sequential training.

**Compliance With Llm Reviewing Policy:**

Affirmed.

**Key Questions For Authors:**

- Why is the initialization of model embeddings performed randomly rather than leveraging model card information for warm-starting?

- The LLM backbone is kept fixed after the first experience. In scenarios involving significantly longer sequences of expansion, will this fixed backbone possess sufficient representational capacity?

**Limitations:**

Yes.

**Strengths And Weaknesses:**

**Strengths**
- The formulation of model routing within an evolving hub as a continual learning task is well-motivated and addresses a previously overlooked paradigm. The identification of scaling and catastrophic forgetting as the primary challenges in this setting is both intuitive and practically relevant.

- CMRBench represents a non-trivial contribution to the research community. By including four sequential experiences and over 2,000 candidate models across diverse domains, the authors provide a large-scale testing ground that is valuable independent of the proposed method.

- The CARvE framework is technically sound, with complementary design choices. Specifically, the coreset replay mechanism is well-suited for efficient scaling, while the dual anchoring objectives effectively mitigate the forgetting problem. The empirical results convincingly demonstrate the effectiveness of both components.

**Weaknesses**

- Based on the comparison between the "from-scratch" and "continual" variants of CARvE (Table 2 and Figure 3b), the continual version exhibits a performance gap while achieving at most a 50% reduction in computational time and resources. Given that model routing often prioritizes accuracy over marginal efficiency gains, the authors should further justify whether these savings are significant enough to warrant the observed performance degradation.

- The current benchmark is restricted to four sequential experiences with relatively "clean" domain boundaries. In realistic continual learning scenarios, systems often encounter non-stationary distributions characterized by blurred task boundaries and gradual concept drift. CMRBench appears limited in its ability to simulate these longer, more complex, and non-stationary sequences, which may not fully reveal the challenges of real-world hub expansion.

---

> ### Author Rebuttal · Authors · 2026-03-30
>
> We thank Reviewer 7qWQ for the careful and constructive review. We address the two questions and two weaknesses below.
>
> **R1-Q1: Random initialisation vs model-card warm-starting**
>
> We ran four warm-starting variants, covering global and within-domain card initialisation at two hyperparameter settings each, all evaluated at 10% domain replay:
>
> | Initialisation | M-Acc | D-Acc | D-Fgt |
> |---|---|---|---|
> | Random (ours)* | 41.8 | 80.3 | 6.7 |
> | Global card τ=0.05, k=20 | 37.5 | 77.0 | 12.7 |
> | Global card τ=0.10, k=50 | 37.6 | 77.5 | 12.9 |
> | Within-domain card τ=0.05, k=20 | 36.4 | 78.4 | 9.7 |
> | Within-domain card τ=0.10, k=50 | 37.1 | 78.4 | 10.1 |
>
> *Single representative run. The three-seed mean reported in Table 2 of the paper is 78.8% D-Acc (±0.3).
>
> Random initialisation outperforms all variants by 2–4pp D-Acc with half the forgetting. We conjecture that card embeddings impose a geometry reflecting linguistic similarity between descriptions rather than routing discriminability, making it harder for the contrastive signal to push similar models apart. Random initialisation lets the contrastive objective organise the space purely for discriminability from the outset.
>
> **R1-Q2: Fixed backbone and representational capacity**
>
> We ran CARvE with a Qwen3-4B backbone: M-Acc=32.8%, D-Acc=75.0%, D-Fgt=14.9% — substantially below the 7B baseline (41.8%/80.3%/6.7%), confirming backbone capacity is a genuine bottleneck.  Qwen-2.5-7B achieves 41.3%/81.1%/6.8%, closely matching our main results and confirming cross-architecture generalisability at the same scale. Yes, you are correct - for longer experience sequences, periodic backbone updates would be necessary; we will add this to the limitations section.
>
> **R1-W1: Compute savings vs performance gap**
>
> CARvE at 10% domain replay trains in ~2 hours on an H100, whereas from-scratch requires ~4 hours. At ~$3/hour cloud pricing:
>
> | Update cadence | CARvE | From scratch | Annual saving |
> |---|---|---|---|
> | Daily | $2,190/year | $4,380/year | $2,190/year |
> | Weekly | $312/year | $624/year | $312/year |
>
> These estimates are for our benchmark scale (~34K examples, ~2K models). A production deployment 50× larger would yield savings of ~$100,000/year with daily updates, making continual operation an operational necessity rather than a marginal efficiency choice.
> It is worth contextualising the absolute accuracy figures with care. CARvE operates as an approximately 2,000-way classification problem, a scale at which top-1 accuracy is inherently a strict criterion. Drawing a parallel to ImageNet evaluation, where top-5 accuracy is the standard reporting convention precisely because of label-space size, CARvE at 10% domain replay achieves top-3 Model-ID accuracy of 72.3% and top-3 Domain accuracy of 95.2%. In practice, this implies that a lightweight re-ranking step over three candidates would yield near-perfect routing, which is a practically relevant operating point for real deployments.
> The relevant practical comparison is not from-scratch retraining (infeasible at production cadence) but the next-best continual alternative, against which CARvE's 9.2pp D-Acc improvement over standard replay (80.3% vs 69.6%) is the operationally meaningful gain.
>
> **R1-W2: Benchmark temporal realism and experience overlap**
>
> The experience boundaries in CMRBench reflect the actual evolution of the Hugging Face hub: E1 covers models released up to 2023, E2 up to 2024, E3 up to May 2025, and E4 up to 2026. Importantly, the experiences are not cleanly disjoint. Table 1 shows that E2 contains 120 previously seen models, E3 contains 70, and E4 contains 83. Table 6 documents even broader overlap at the family level, with 107, 31, and 46 model families reappearing across E2, E3, and E4 respectively, capturing realistic patterns of model versioning and derivative releases. CMRBench therefore already embeds a degree of partial stationarity that is characteristic of real hub evolution. We fully agree that scenarios with more gradually blurred boundaries are not yet captured by the current benchmark design, and will leave this as future work.

---

> > ### Author Rebuttal · Reviewer_7qWQ · 2026-04-04
> >
> > I thank the authors for addressing my concerns. Please ensure that the relevant discussions and additional experiments are incorporated into the revised version of the manuscript. I maintain my initial score of Weak Accept.

---

### Decision · Program_Chairs · 2026-04-30

**Decision:**

Accept (regular)

**Comment:**

This paper studies Continual Model Routing, the problem of selecting appropriate models from an evolving model hub before inference, and frames it as a class-incremental continual learning problem. The reviewers generally agreed that this is a timely and practically relevant setting, and viewed both CMRBench and CARvE as meaningful contributions: CMRBench provides a large-scale temporally structured benchmark over thousands of models and many domains, while CARvE offers a technically coherent embedding-based routing approach with replay and anchoring mechanisms tailored to growing label spaces. I also carefully read the authors’ rebuttal and follow-up comments, and took them into account in my assessment.

The main weaknesses are also clear. Several reviewers noted that the core method combines established ingredients, so the methodological novelty lies more in problem formulation, benchmark design, and the task-specific integration of these components than in fundamentally new continual-learning primitives. There were also concerns about whether the benchmark and evaluation fully capture true capability-based routing rather than label matching, especially because part of HuggingBench relies on LLM-generated queries and because evaluation is necessarily pre-inference rather than based on post-execution output quality. Additional questions were raised about the modest top-1 model-ID accuracy, the lack of zero-shot support for unseen model IDs, scalability beyond the current benchmark size, the limited benefit of domain-stratified replay at standard replay budgets, and sensitivity to backbone capacity and benchmark ordering. That said, the rebuttal substantially strengthened the paper by adding EWC/LwF comparisons, multi-backbone results, human evaluation of prompt-model pairs, and additional sensitivity and robustness evidence, which addressed a significant portion of these concerns.

Overall, after considering the full set of reviews and the rebuttal, I find that the strengths outweigh the remaining weaknesses. In particular, the formulation of continual model routing, the release of CMRBench, and the solid empirical evidence for CARvE together make this a useful contribution that is likely to stimulate further work at the intersection of continual learning and model routing. While some limitations remain and should be clearly discussed in the final version, I believe the paper is technically sound, well motivated, and sufficiently novel at the level of problem framing, benchmark contribution, and method design to merit acceptance.